# Mapping cells through time and space with moscot

Dominik Klein[1,2,14], Giovanni Palla[1,3,14], Marius Lange[1,2,4,14], Michal Klein[5,14], Zoe Piran[6,14], Manuel Gander[1], Laetitia Meng-Papaxanthos[7], Michael Sterr[8,9], Lama Saber[8,9,10], Changying Jing[8,9,11], Aimée Bastidas-Ponce[8,9], Perla Cota[8,9,10], Marta Tarquis-Medina[8,9], Shrey Parikh[1], Ilan Gold[1], Heiko Lickert[8,9,10 ✉], Mostafa Bakhti[8,9], Mor Nitzan[6,12,13], Marco Cuturi[5] & Fabian J. Theis[1,2,3 ✉]

Single-cell genomic technologies enable the multimodal profiling of millions of cells across temporal and spatial dimensions. However, experimental limitations hinder the comprehensive measurement of cells under native temporal dynamics and in their native spatial tissue niche. Optimal transport has emerged as a powerful tool to address these constraints and has facilitated the recovery of the original cellular context[1–4]. Yet, most optimal transport applications are unable to incorporate multimodal information or scale to single-cell atlases. Here we introduce multi-omics single-cell optimal transport (moscot), a scalable framework for optimal transport in single-cell genomics that supports multimodality across all applications. We demonstrate the capability of moscot to efficiently reconstruct developmental trajectories of 1.7 million cells from mouse embryos across 20 time points. To illustrate the capability of moscot in space, we enrich spatial transcriptomic datasets by mapping multimodal information from single-cell profiles in a mouse liver sample and align multiple coronal sections of the mouse brain. We present moscot.spatiotemporal, an approach that leverages gene-expression data across both spatial and temporal dimensions to uncover the spatiotemporal dynamics of mouse embryogenesis. We also resolve endocrine-lineage relationships of delta and epsilon cells in a previously unpublished mouse, time-resolved pancreas development dataset using paired measurements of gene expression and chromatin accessibility. Our findings are confirmed through experimental validation of NEUROD2 as a regulator of epsilon progenitor cells in a model of human induced pluripotent stem cell islet cell differentiation. Moscot is available as open-source software, accompanied by extensive documentation.

Single-cell genomic technologies have increased our understanding of the dynamics of cellular differentiation and tissue organization. Single-cell assays such as single-cell RNA sequencing (scRNA-seq) profile the molecular state of individual cells at high resolution, whereas spatial assays recover their spatial organization. However, these experiments involve destruction of the cell and capture only a subset of molecular information. As a result, cellular profiles have to be realigned.

Previous work addressed such problems by using optimal transport (OT), a field concerned with mapping and comparing probability distributions[1]. OT has been instrumental in delineating cellular reprogramming processes[2], reconstructing tissue architecture by enhancing spatial data with single-cell references[3] and building common coordinate frameworks (CCFs) of a biological system by aligning spatial transcriptomic data[4].

Despite the potential of OT-based methods to address mapping problems in single-cell genomics, their use faces three key challenges. First, implementations of OT-based tools are geared to unimodal data. Second, current OT methods used in single-cell genomics are computationally expensive. That is, time complexity scales quadratically[5] (or cubically for Gromov–Wasserstein extensions[1,6]) in the number of cells. Similarly, memory scales quadratically, which prevents their application to atlas-scale datasets[7]. Third, existing tools build on heterogeneous implementations[2–4], which make it difficult to adapt or combine approaches to new problems.

[1]Institute of Computational Biology, Helmholtz Center, Munich, Germany. [2]Department of Mathematics, Technical University of Munich, Garching, Germany. [3]TUM School of Life Sciences Weihenstephan, Technical University of Munich, Freising, Germany. [4]Department of Biosystems Science and Engineering, ETH Zürich, Basel, Switzerland. [5]Apple, Paris, France. [6]School of Computer Science and Engineering, The Hebrew University of Jerusalem, Jerusalem, Israel. [7]Google DeepMind, Zurich, Switzerland. [8]Institute of Diabetes and Regeneration Research, Helmholtz Center, Munich, Germany. [9]German Center for Diabetes Research, Neuherberg, Germany. [10]School of Medicine, Technical University of Munich, Munich, Germany. [11]Munich Medical Research School (MMRS), Ludwig Maximilian University (LMU), Munich, Germany. [12]Racah Institute of Physics, The Hebrew University of Jerusalem, Jerusalem, Israel. [13]Faculty of Medicine, The Hebrew University of Jerusalem, Jerusalem, Israel. [14]These authors contributed equally: Dominik Klein, Giovanni Palla, Marius Lange, Michal Klein, Zoe Piran. ✉e-mail: heiko.lickert@helmholtz-munich.de; fabian.theis@helmholtz-munich.de

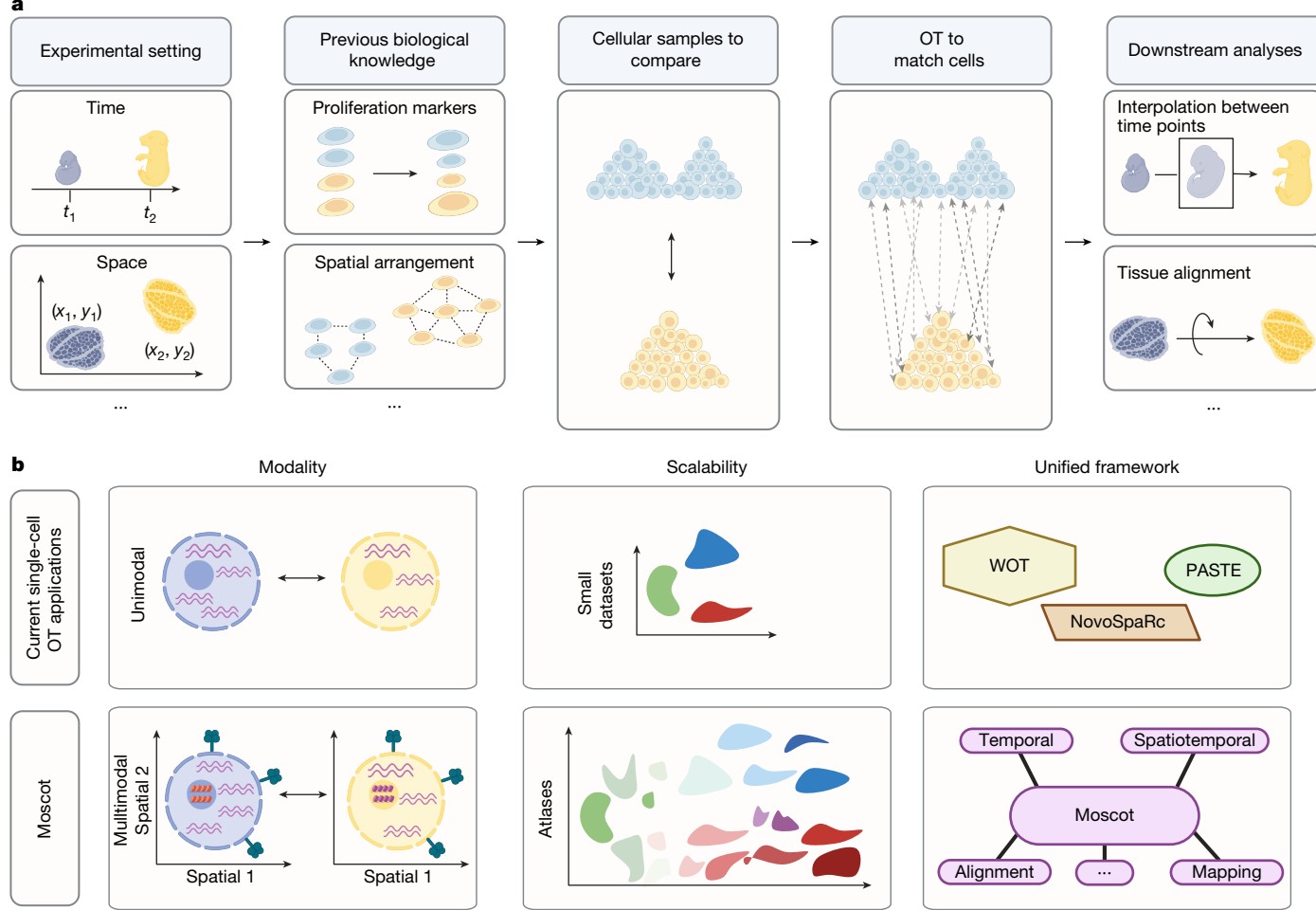

**Fig. 1 | Moscot enables efficient multimodal OT across single-cell applications. a**, Schematic of a generic OT pipeline for single-cell genomic analyses (from left to right): experimental shifts (for example, time points and different spatial slides) lead to disparate cell populations. Previous biological knowledge (for example, proliferation rates and spatial arrangement) is often available and should be used to guide the mapping process. OT aligns cellular distributions by minimizing the displacement cost. The learnt mapping facilitates various downstream analysis opportunities. **b**, Moscot introduces three key innovations that unlock the full power of OT. First, it supports multimodal data across all models. Second, it overcomes previous scalability limitations to enable atlas-scale applications. Third, moscot is a unified framework with a consistent API across biological problems, which will facilitate usability and enable extensions to new problems in a straightforward manner. Panels **a** and **b** were created using BioRender (https://www.biorender.com).

Here we present moscot, a computational framework to solve mapping and alignment problems, and we demonstrate its capabilities for temporal, spatial and spatiotemporal applications. Moscot is based on three design principles to overcome current limitations. Moscot supports multimodal data, improves scalability and unifies previous single-cell applications of OT in the temporal and spatial domain. We also introduce a previously undescribed spatiotemporal application. An intuitive application programming interface (API) that interacts with the broader scverse[8] ecosystem makes these features accessible.

We demonstrate the capabilities of moscot by studying the development of 1.7 million cells during mouse embryogenesis. Furthermore, we map information from multimodal cellular indexing of transcriptomes and epitopes by sequencing (CITE-seq) to high-resolution spatial readouts in the mouse liver and align large spatial transcriptomic sections of mouse brain samples. Concurrently to SPATEO[9], we introduce the concept of spatiotemporal mapping and demonstrate its benefits using a spatiotemporal atlas of mouse embryogenesis[10]. Finally, we jointly profile gene expression and chromatin accessibility during mouse pancreas development and apply moscot to better delineate cell trajectories of delta and epsilon cells. We identify potential transcription factors (TFs) that drive lineage formation and experimentally verify NEUROD2 as a TF that regulates epsilon-cell formation during human

endocrinogenesis in vitro. Moscot unlocks OT for multiview atlas-scale single-cell applications and it is accessible, together with extensive documentation, at https://moscot-tools.org.

## Moscot is an OT framework for mapping cells

Moscot translates biological mapping and alignment tasks into OT problems and solves them using a consistent set of algorithms. Moscot takes unpaired datasets as input; for example, measurements taken at different time points or corresponding to different spatial transcriptomic slides, each containing one or more molecular modalities. Moscot also accepts previous biological knowledge, such as cellular growth rates, to guide the mapping process. Moscot solves an OT problem and generates a coupling matrix that probabilistically relates samples in each of the datasets. Equipped with that coupling matrix, moscot offers various application-specific downstream analysis functions (Fig. 1a and Methods).

Moscot builds on three notions of OT to accommodate various biological problems. These differ in how samples are related across cellular distributions: Wasserstein-type (W-type)[5] OT compares two sets of cells with the same cellular features; Gromov–Wasserstein-type (GW-type)[6] OT compares cellular distributions living in different spaces;

and fused Gromov–Wasserstein-type (FGW-type)[11] OT compares cells with partially shared features (Methods and Supplementary Note 1). We built on previous OT-based method assumptions to map cells across temporal and spatial domains (Methods).

To support multimodality throughout the framework, we leveraged shared latent representations (Fig. 1b and Methods). We made moscot applicable to atlas-scale datasets by reducing the computation time and the memory consumption of W-type, GW-type and FGW-type notions by orders of magnitude compared with previous OT-based tools (Fig. 1b, Methods and Supplementary Note 2). Specifically, we based moscot on optimal transport tools (OTT)[12], a scalable JAX implementation of OT algorithms that supports just-in-time compilation, on-the-fly evaluation of the cost function and GPU acceleration (Methods). When required by the size of the dataset, we used recent methodological innovations[13–15] that constrain the coupling matrix to be low-rank, which enabled linear time and memory complexity for W-type, GW-type and FGW-type notions (Supplementary Note 3). A unified API makes moscot easy to use and extend (Supplementary Fig. 1). In particular, modular implementation enables the use of similar infrastructure for solving different biological problems.

## Reconstructing mouse embryogenesis

Modelling cell-state trajectories for biological systems that are not in steady state requires time-course single-cell studies combined with computational analysis to infer cellular differentiation across time points. Waddington OT[2] (WOT) solves the problem using W-type OT. However, WOT remains limited to unimodal gene-expression data and does not scale to large datasets. Thus, we created moscot.time. Our model inherits the popular cell-growth-rate modelling of WOT and is applicable to multimodal data. Moreover, it scales to millions of cells and, like all trajectory inference methods in moscot, can be interfaced with tools such as CellRank 2 (refs. 16,17) for downstream analyses (Methods).

We asked whether the improved scalability of moscot.time translates into a more faithful description of biological systems. Thus, we applied our model to a published atlas[7] of early mouse development that contains almost 1.7 million cells across 20 time points spanning embryonic day 3.5 (E.3.5) to E13.5 (Fig. 2a and Methods). We first assessed whether we could use WOT[2] to analyse this dataset. We selected the E11.5–E12.5 time point pair, which contained more than half a million cells, and benchmarked memory and computation time on subsets of increasing cell number (Fig. 2b, Methods and Supplementary Table 1). Moscot.time computed a coupling for all 275,000 cells at both time points, whereas WOT ran out of memory as soon as 75,000 cells was exceeded. When we included a low-rank OT approximation in moscot[13–15], this addition computed coupling faster than default moscot.time once 75,000 cells per time point was exceeded. The linear memory complexity of moscot.time enables it to process developmental atlases on a laptop, whereas WOT failed on a server (Fig. 2b, Methods and Extended Data Fig. 1).

As WOT did not scale to a dataset of this size, the authors of the developmental atlas[7] devised a deterministic approach based on $k$-nearest neighbour ($k$NN) matching called trajectories of mammalian embryogenesis (TOME). We formulated two metrics that operated on the level of germ layers and cell types (Methods and Supplementary Table 2). For both metrics, moscot.time achieved comparable performance to TOME across all time points and developmental stages, even though TOME was specifically designed for this dataset (Fig. 2c). For the low-rank approximation, the accuracy for both metrics converged to default moscot.time for sufficiently large ranks while being faster (Extended Data Fig. 1b,c). Moreover, the performance of moscot mappings was robust with respect to rank and embedding (Supplementary Figs. 2–4). We further compared TOME and moscot.time using cellular growth rates and death rates. As TOME only provides cluster-level mappings, we extended the original approach to produce cell-level output with

cell-level TOME (clTOME) (Methods). Using the E8.0–E8.25 pair of time points, we mapped cells using moscot.time and clTOME (Fig. 2d). clTOME frequently assigned growth rates much smaller than one and predicted that more than 19% of the population at this stage is apoptotic (Fig. 2e and Supplementary Table 3). Such a high death rate represents an unrealistic scenario for embryonic development, whereby beyond E7.0, the fraction of cells going through apoptosis is typically <10%[18]. By contrast, we were able to tune the growth rates predicted by moscot.time to be more realistic and cell-type specific (Fig. 2e and Methods). These results generalized to all other time points that contained sufficient cell numbers (Supplementary Figs. 5–7). We also compared predicted growth rates with scanpy-computed cell-cycle scores[19] on an in vitro reprogramming dataset[2], for which we expected predictions to be less affected by cell-sampling stochasticity. The predictions generated using moscot.time correlated better with averaged growth rates for each cell set than when using clTOME (Pearson's $r$ of 0.48 compared with 0.13, respectively; Supplementary Fig. 8).

Next, we considered the reliability of the models for cell-fate prediction. We considered E8.25 first heart field cells, a population that emerges from the splanchnic mesoderm[20]. We used moscot.time and clTOME to compute ancestor probabilities, which quantify the likelihood of E8.0 cells to differentiate to E8.25 first heart field cells. We compared ancestor probabilities with the expression of known driver genes for the formation of first heart field cells at E8.0 (Fig. 2f, Methods and Supplementary Table 3). Using moscot.time, we consistently achieved higher absolute Spearman's correlations (Fig. 2g), a result that generalized to three other cell types we investigated across early development (Fig. 2h and Supplementary Table 4). Finally, we showed that mapping metacells[21] instead of single cells yielded comparable results in terms of germ layer and cell type scores, but failed to resolve rare primordial germ cells at E9.5 and gave lower driver gene correlations for the pancreatic epithelium (Methods and Extended Data Fig. 2).

## Mapping and aligning spatial samples

Spatial omic technologies enable the profiling of thousands of cells in their native tissue environment. The analysis of such data requires methods that are able to integrate datasets across molecular layers and spatial coordinate systems. OT has proven useful to tackle these problems, particularly novoSpaRc[3] for gene-expression mapping and PASTE[4] for the alignment of spatial transcriptomic datasets. Moscot implements both applications and leverages scalable implementations and more performant algorithms (Methods).

Image-based spatial transcriptomic data are often limited in the number of genes measured (hundreds to a few thousands)[22]. The mapping problem of moscot learns a coupling between dissociated single-cell profiles and their spatial organization using an FGW-type problem. This enabled us to incorporate cellular similarities in molecular features and physical distances of cells (Methods). The OT solution facilitated the transfer of gene-expression or additional multimodal profiles to spatial coordinates (Fig. 3a).

We benchmarked moscot against two state-of-the-art methods, Tangram[23] and gimVI[24], on a recent benchmark[25]. We assessed the quality of the mapping process by computing correlations of held-out genes in spatial coordinates (Methods). Moscot consistently outperformed the other methods across 14 datasets generated using various technologies. Furthermore, for each dataset, we quantified spatial correspondence, a measure of correlation between gene-expression similarity and distances in physical coordinates, as originally proposed[3] (Methods). A spatial transcriptomic dataset has high spatial correspondence if nearby cells have similar gene-expression profiles (Supplementary Fig. 9). Moscot showed a positive correlation between spatial correspondence and accuracy (Fig. 3b), which indicated that it is able to leverage spatial associations between

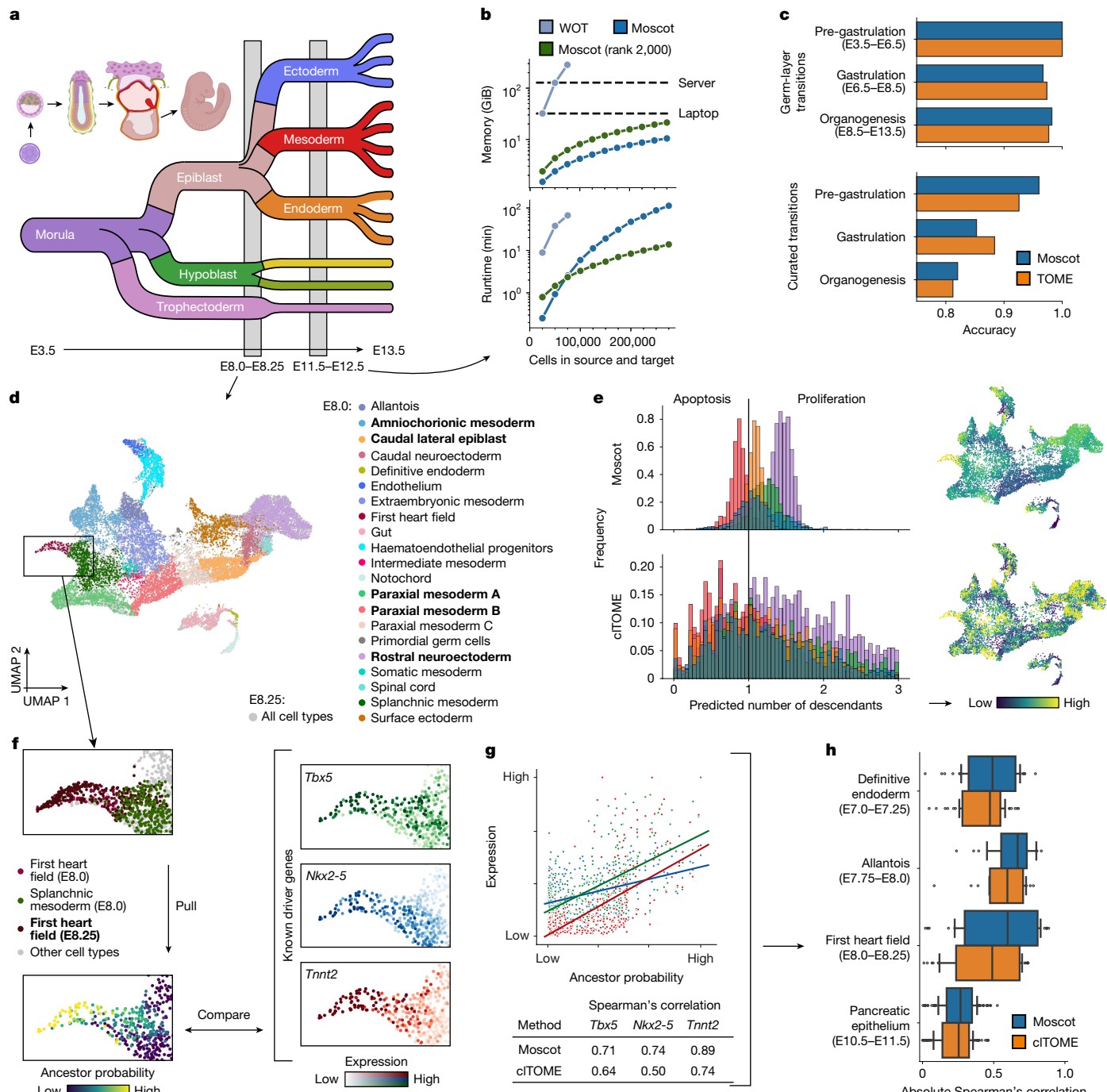

**Fig. 2 | Moscot faithfully reconstructs atlas-scale developmental trajectories.** **a**, Schematic of an example mouse embryogenesis atlas[7], which includes 20 time points and 1.7 million cells. **b**, Benchmark of peak memory consumption (top, on CPU) and computation time (bottom, on GPU) for increasing numbers of cells, subsampled from the E11.5–E12.5 time point pair (Methods and Supplementary Table 1). We compared WOT[2] with default moscot.time and low-rank[13–15] moscot.time (rank 2,000) (Supplementary Note 3; WOT was run on CPU as it does not support GPU acceleration). **c**, Accuracy comparison between TOME[7] and moscot.time in terms of germ-layer and cell-type transition scores by developmental stage (Methods and Supplementary Table 2). **d**, Uniform manifold approximation and projection (UMAP) projection of the E8.0–E8.25 time point pair, coloured by original cluster annotations. **e**, Growth-rate estimates of moscot.time (top) and clTOME (bottom) for the five most prevalent E8.0 cell types in **d** (highlighted in bold) as histograms (left) and on UMAP projections (right). The black vertical bar denotes a growth rate of one. **f**, The ancestor probability for E8.25 first heart field cells (left) versus gene-expression levels of known driver genes *Tbx5*, *Nkx2-5* and *Tnnt2* (right; Methods and Supplementary Table 7) calculated using moscot.time. **g**, Quantification of the comparison in **f** using Spearman's correlation. Genes are coloured as in **f**, and each dot denotes a cell and lines indicate a linear data fit. **h**, Distribution (*n* = 36 genes (definitive endoderm), *n* = 18 (allantois), *n* = 39 (heart field), *n* = 106 (pancreatic epithelium); vertical lines correspond to quarters, whiskers are outliers) of absolute Spearman's correlation values between ancestor probabilities and known driver-gene expression for moscot.time and clTOME (Methods and Supplementary Table 4). Panel **a** was created using BioRender (https://www.biorender.com).

distances in gene expression and physical space. Nevertheless, even when spatial correspondence was low, moscot outperformed the other methods.

We then set out to map multimodal single-cell profiles to their spatial context. This method is of particular interest as spatial transcriptomic technologies are mostly limited to gene-expression measurements[22].

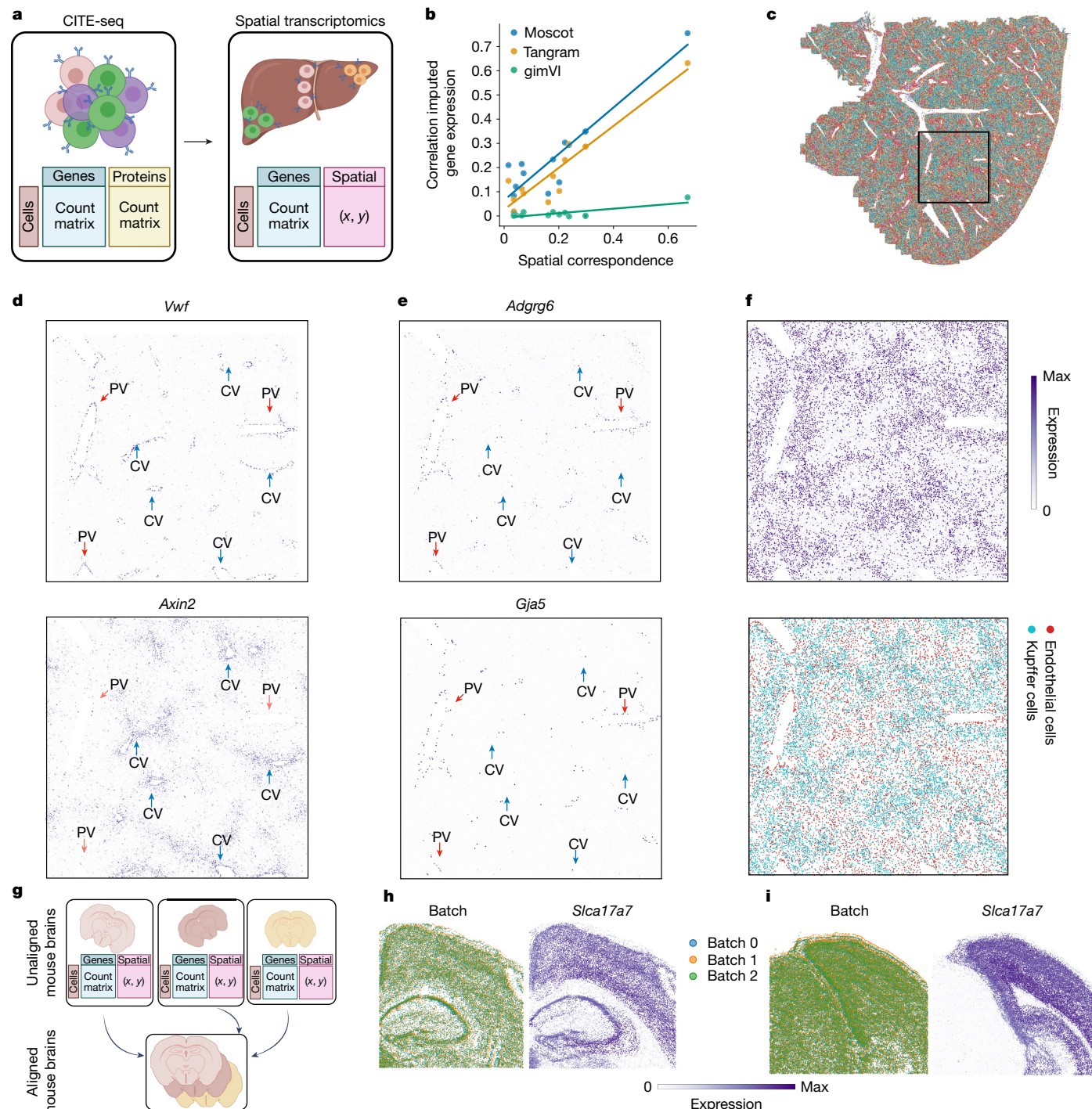

**Fig. 3 | Moscot enables multimodal mapping and alignment of spatial transcriptomic data. a**, Schematic of a multimodal single-cell reference dataset being mapped onto a spatial dataset. **b**, Spatial correspondence is associated with prediction accuracy in moscot. Linear fit of the median Spearman's correlation between true and imputed gene expression with respect to the spatial correspondence (Methods) of 12 datasets. **c**, Liver sections with annotations mapped from the CITE-seq dataset (Extended Data Fig. 3). The square marks the cropped tiles in **d**–**f**. **d**, Measured gene expression for *Vwf* (endothelial cell marker) and *Axin2* (hepatocytes and endothelial marker). *Vwf* is used to identify all epithelial cells that define the boundaries of CVs and PVs. *Axin2* is a positive marker for CVs. **e**, Predicted gene expression for *Adgrg6* and *Gja5*, known endothelial cells markers for PVs. **f**, Predicted protein expression of folate receptor β, a marker for Kupffer cells (top) and imputed cell types for Kupffer cells and endothelial cells (bottom). **g**, Schematic of the process of aligning sections from multiple slides to a common reference sample. **h**, Visualization of a tile of the spatial sections of the mouse brain for section 1 coloured by batch (left) and by expression of *Slc17a7* (right). **i**, Visualization of a tile of the spatial sections of the mouse brain for section 2 coloured by batch (left) and by expression of *Slc17a7* (right). Panels a and g were created using BioRender (https://www.biorender.com).

We considered a CITE-seq dataset of around 91,000 cells of the mouse liver[26] and a spatial transcriptomic section consisting of about 367,000 cells measured using the Vizgen MERSCOPE platform (Fig. 3c). We incorporated gene-expression, protein and spatial information to recover the spatial organization of the proteins (Methods). We then mapped annotations from the CITE-seq dataset as no cell-type

annotation was provided in the original data (Extended Data Fig. 3). Use of any of the other methods was not feasible owing to prohibitive time or memory complexity.

A central problem in liver physiology is the identification of central veins (CVs) and portal veins (PVs) to characterize liver zonation[27]. This problem can be solved by considering expression patterns of marker genes, cell-type localization and protein abundance. CVs can be identified using *Axin2*, a CV-associated endothelial cell marker[28] (Fig. 3d). Similarly, *Vwf*, a known marker for endothelial cells in blood vessels, indicates the presence of both CVs and PVs[29]. However, owing to the limited number of genes measured in the spatial transcriptomic data, it proved challenging to identify PVs on the basis of marker gene expression. Leveraging moscot, we overcame this constraint by mapping the expression of the PV-specific markers *Adgrg6* and *Gja5* (ref. 26) (Fig. 3e and Supplementary Fig. 10). Another limitation of characterizing cellular niches of liver zonation was the lack of detailed cell-type annotation and protein expression. Hence, we used moscot to transfer the cell-type annotation provided by the single-cell dataset. Focusing on resident liver macrophages called Kupffer cells, we confirmed their enriched presence in areas around PVs where liver sinusoids are more prevalent[26]. We corroborated our findings by mapping the folate receptor β protein to its spatial organization (Fig. 3f). By integrating results from cell-type annotation, measured and imputed marker genes, and transferred protein expression, we could characterize in detail the tissue niche of liver zonation in a mouse liver sample. We quantitatively confirmed the benefits of incorporating multiple modalities by imputing assay for transposase-accessible chromatin with sequencing (ATAC–seq) data on a spatial multiome dataset of human tonsils. This analysis consisted of the joint profiling of spatially resolved ATAC–seq and RNA-seq data (Supplementary Fig. 11).

A different prevalent task in spatial transcriptomics is building a consensus view of the tissue of interest. This requires the alignment of several spatial measurements from contiguous sections or from the same section from different biological replicates. The alignment problem of moscot facilitates the alignment of several sections and the building of such a consensus view from multiple spatial transcriptomic slides (Fig. 3g). This is an important step towards building a CCF of biological systems. First, we evaluated the capability of moscot to spatially align synthetic datasets adapted from previous benchmark studies[4,30] and with other registration methods not specific to spatial omic data. The benchmark results showed that moscot performed on par or better than the method PASTE[4] (Methods and Supplementary Figs. 12 and 13).

Next, we set out to investigate the scalability of the methods to larger datasets. To that end, we used the brain coronal sections from MERSCOPE (Methods). This dataset is prohibitively large for methods such as PASTE (around 250,000 cells for section 1 and about 300,000 cells for section 2; Methods). Moscot accurately aligned two samples to the reference slide for both coronal sections of the mouse brain. We observed that for most genes, there was a strong correspondence of gene-expression densities across cellular neighbourhoods both quantitatively and visually (Fig. 3h,i, Extended Data Fig. 4 and Supplementary Figs. 14 and 15).

## Charting spatiotemporal mouse development

The advent of spatially resolved single-cell datasets of developmental systems presents the challenge of developing methods that are able to delineate cellular trajectories and leverage both intrinsic and extrinsic effects on cellular phenotypes. Here we introduce a trajectory inference method that incorporates similarities in gene-expression profiles and physical distances to infer more accurate trajectories. It is based on a FGW-type problem that merges moscot.time and moscot.space into a spatiotemporal method (Methods).

We assessed the capabilities of moscot to perform trajectory inference on the mouse embryogenesis spatiotemporal transcriptomic atlas (MOSTA)[10], which consists of eight time points from E9.5 to E16.5. We analysed a single slide for each time point, which resulted in a total of about 500,000 spatial array locations (hereafter denoted as bins, per a previously described notation[10]; Fig. 4a, and Methods). We used annotations to major tissue regions and organs as provided by the authors[10] and evaluated the annotation-transition score over computed trajectories (Methods and Supplementary Table 5). We compared the performance of moscot.spatiotemporal to trajectories computed from only gene-expression information across time points using either moscot.time (a W-type problem) or TOME[7] (Fig. 2). Accounting for spatial similarity in the trajectory inference resulted in an improved prediction of annotation-transition scores, with an average improvement across time points of 5% and 13% with respect to moscot.time and TOME, respectively (Fig. 4b and Methods). Moreover, moscot.spatiotemporal outperformed PASTE2 (ref. 31) and was robust with respect to hyperparameters (Supplementary Fig. 16). Next, we used moscot to identify driver and target genes of liver development (Methods), which revealed known hepatic genes *Afp*, *Alb* and *Apoa2* and established driver genes that encode the TF HNF4A (Supplementary Table 6).

Subsequently, we focused on the fates of heart and brain regions of the developing mouse embryo. For each pair of consecutive time points, we visualized heart bins at the earlier time point and where these bins mapped to at the later time point (Fig. 4c). To further characterize cellular dynamics, we interfaced moscot with CellRank 2 (refs. 16,17) (Methods), which enabled the identification of cellular fates on the basis of the coupling matrix provided by moscot. The predicted fates corresponded to the known differentiation lineages of the mouse embryo[10] (Extended Data Fig. 5). We also identified known driver genes of heart development, such as *Gata4* and *Tbx20* (which encode TFs) and genes related to metabolism and heart regeneration, such as *Myl7* and *Myh6* (Fig. 4d and Supplementary Table 7).

A study by Chen et al.[10] provided a cell-type annotation of the brain tissue at E16.5, but not for earlier time points. To investigate developmental trajectories in the brain, we utilized moscot to transfer cell-type annotation from the E16.5 data to preceding time points. Visually, predicted annotations retained the spatial distribution of the manual annotation (Fig. 4e), and quantitatively, they showed strong correspondence with reported marker genes (Methods and Supplementary Fig. 17).

The interplay between moscot and CellRank 2 enabled us to identify terminal states of brain development in the mouse embryo, with fate probabilities that were in accordance with the predicted annotation (Supplementary Fig. 18). Analogous to the heart, we predicted driver genes of neuron and fibroblast development (Fig. 4f,g and Methods). For neuronal fate, identified TF-encoding genes such as *Tcf7l2*, *Sox11*, *Myt1l* and *Zfhx* have previously been reported as relevant for neuronal development (Supplementary Table 8). Notably, our results included known spatially localized drivers, such as *Neurod2*, which is associated with forebrain glutamatergic neurons[32], and non-regional drivers, such as *Sox11* (Fig. 4g). For fibroblasts, we identified the TF-encoding genes *Prrx1*, *Runx2* and *Msx1*, and known key genes such as *Dcn*, *Col1a2* and *Col1a1* (Supplementary Table 9). Finally, we demonstrated the capabilities of moscot to recover trajectories in three-dimensional (3D) spatiotemporal data by identifying key TFs in the embryonic development of *Drosophila*[33] (Methods and Supplementary Fig. 19).

## Delineating mouse pancreas development

To highlight the potential of moscot for studying complex lineage relationships, we focused on the poorly understood process of delta cell and epsilon cell formation during mouse pancreas development[16,34,35] (Supplementary Note 4). Hypotheses of lineage specification range

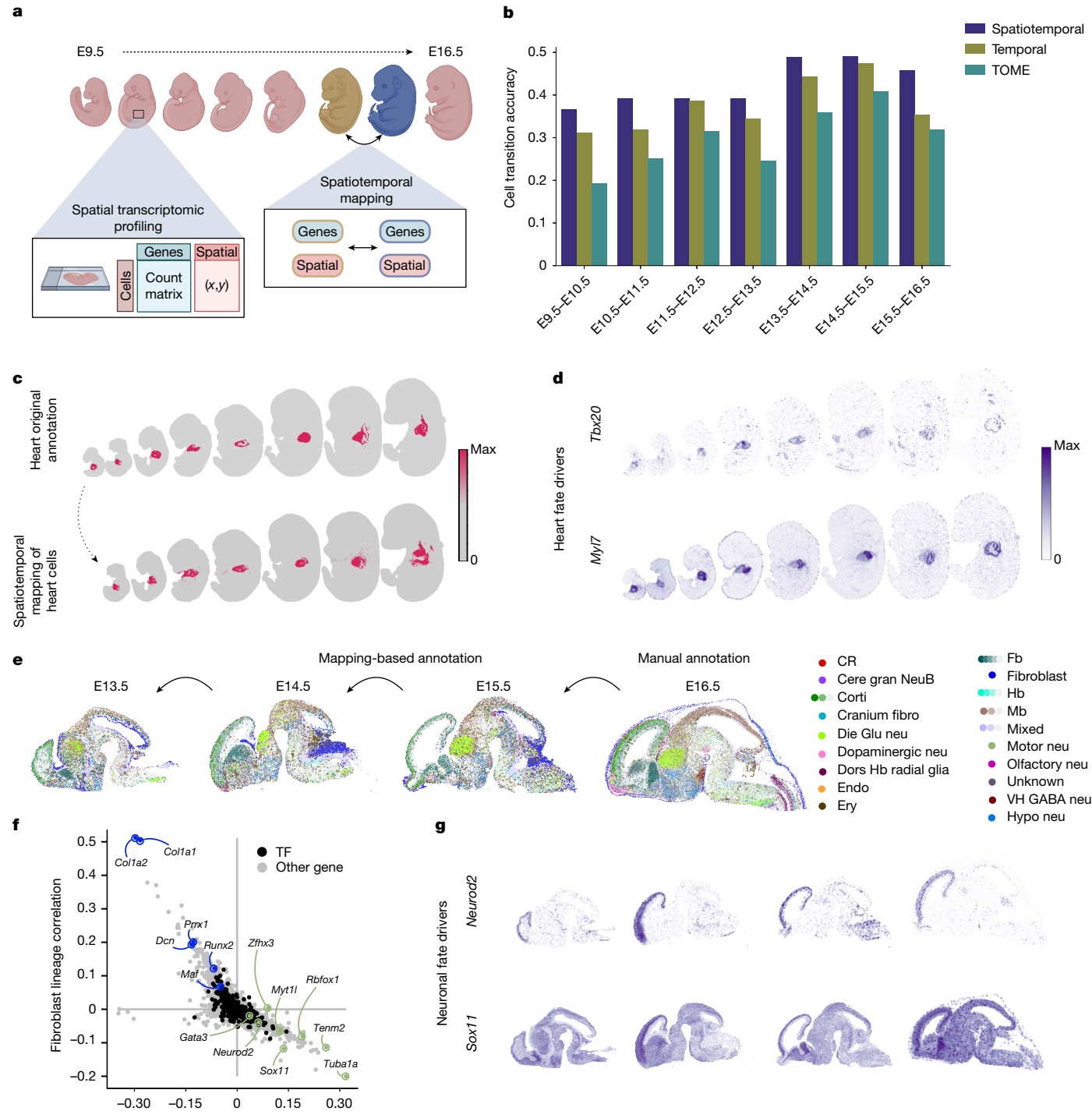

**Fig. 4 | Inference of spatiotemporal dynamics with moscot. a**, Schematic of spatiotemporal trajectory inference of mouse embryogenesis. **b**, Accuracy of curated transitions across developmental stages (Methods and Supplementary Table 5) for the temporal and spatiotemporal application of moscot compared with TOME[7]. **c**, Mapping heart cells across time points (bottom) and ground-truth annotation of the heart lineage (top). **d**, Heart-lineage driver genes found by interfacing moscot with CellRank 2 (refs. 16,17). Top, *Tbx20* encodes a TF known to have various fundamental roles in cardiovascular development. Bottom, *Myl7* encodes a protein related to metabolism and heart regeneration (Supplementary Table 7). **e**, Transferring high-resolution cell-type annotations only provided in the latest time point (E16.5) to earlier time points. **f**, Pearson's correlations of gene expression with neuronal (*x* axis) and fibroblast (*y* axis) fate probabilities. Annotated genes are among the top 20 driver genes and were previously associated with fibroblasts and neuronal lineage (Supplementary Tables 7 and 8). **g**, Spatial visualization of sample neuronal-driver genes, *Neurod2* and *Sox11* (Supplementary Table 8). Cere gran NeuB, cerebellar granule neuroblast; corti, cortical; CR, Cajal–Retzius cell; fibro, fibroblast; die, diencephalon; dors, dorsal; endo, endothelial; ery, erythrocyte; Fb, forebrain; Glu, glutamatergic; neu, neuron; Hb, hindbrain; hypo, hypothalamus; Mb, midbrain; VH, ventromedial hypothalamus. Panel **a** was created using BioRender (https://www.biorender.com).

from delta cells splitting simultaneously with alpha and beta cells after going through a common *Fev*[+] cell state[36] to delta cells being derived from the same progenitor population as beta cells[37]. In previous work[16,34], we had proposed that delta cells differentiate from a *Fev*[+] population, but we could not resolve their precise lineage hierarchy. Similarly, our previous analysis[16] had indicated that epsilon cells develop from both *Ngn3*[+] progenitors and glucagon-producing alpha cells. However, lineage-tracing experiments confirmed that epsilon cells that produce ghrelin (encoded by *Ghrl*) are not in a terminal state and can give rise to alpha and PP cells and rare beta cells[38].

We wanted to better understand the cellular fates of pancreas cells. Therefore, we used the NGN3–Venus fusion (NVF) reporter mouse line[34] to generate a single-nucleus (snRNA) and ATAC multiome dataset of E14.5 (about 9,000 nuclei), E15.5 (10,000 nuclei) and E16.5 (3,000 nuclei) of the pancreatic epithelium enriched for endocrine progenitors (Fig. 5a and Methods). *Ngn3* encodes a master regulatory TF necessary and sufficient for endocrine-cell formation in the pancreas. Hence, enrichment of *Ngn3*[+] progenitors enabled a detailed study of endocrine lineage induction and segregation into glucagon-producing alpha cells, insulin-producing beta cells, somatostatin-producing delta cells, pancreatic polypeptide-producing PP cells and ghrelin-producing epsilon cells. Compared with previous scRNA and ATAC–seq studies that relied on bulk ATAC measurements[35] or on a low number of cells for scRNA-seq[36], our dataset enabled a comprehensive multimodal analysis of endocrine-cell differentiation.

We observed a distributional shift in cell-type abundance between time points (Fig. 5b and Extended Data Fig. 6). Clustering based on both modalities revealed the expected cell-type heterogeneity in the endocrine branch, ranging from *Ngn3*[Low] to heterogeneous progenitors of endocrine-cell states (Fig. 5c, Methods, Supplementary Table 10 and Supplementary Fig. 20). We linked the cells across the three time points with moscot.time by leveraging information from both gene-expression and ATAC data (Supplementary Note 5). To validate the couplings, we aggregated the transport matrix to the cell-type level and found that the majority of recovered transitions were supported by the literature[16,34,36,39,40] (Fig. 5d, Methods and Supplementary Fig. 21). We also studied the influence of cost and embeddings. The results revealed the necessity of using geodesic costs while being robust with respect to the embedding (Methods and Supplementary Fig. 22). Moreover, we recovered the correct cell-cycle direction using moscot (Supplementary Note 6 and Supplementary Fig. 23).

Subsequently, we explored the lineage segregation of delta and epsilon cells. Therefore, we restricted our analysis to the endocrine branch and further subclustered the poorly understood *Fev*[+] delta cell population. To emphasize the developmental axes of variation, we computed an embedding using PHATE[41] (Fig. 5e). We used moscot to compute putative ancestry and descendancy relationships and found that alpha, beta and delta cells are predicted to mostly remain in their cellular identity as expected (Methods and Supplementary Figs. 24 and 25). We predicted both epsilon and delta cells to follow a similar trajectory (Fig. 5d,e). In particular, moscot modelled that progenitors of epsilon cells and a large proportion of progenitor cells of delta cells branch off the *Ngn3*[High] population at a similar cellular state.

Next, we quantified the predicted descendancy relationships between cell types and confirmed that the cell transitions computed from E14.5 and E15.5 data are in line with results obtained for E15.5 and E16.5 (Fig. 5f). In particular, epsilon cells partially mature into alpha cells (Fig. 5f and Supplementary Fig. 26), as previously reported[37,38]. Moreover, most of the epsilon cell population was derived from a population that we refer to as epsilon progenitors, which themselves we predict to originate from the *Ngn3*[High] endocrine progenitors (Supplementary Fig. 27). Contrary to our recent hypothesis[34], the epsilon progenitor population showed a low mean expression of *Fev*, which implied that these cells have a relatively immediate expression of *Ghrl* following *Fev* (Supplementary Fig. 28). We corroborated this hypothesis using

independent computational methods (Extended Data Fig. 7); however, experimental validation of this claim is necessary.

Based on the results of moscot.time, delta cells are mainly derived from *Fev*[+] delta cells. Although our data did not reveal a single source of origin of *Fev*[+] delta cells, moscot predicted that a considerable proportion of *Fev*[+] delta cells have a similar origin as epsilon cells (Fig. 5f). We computationally confirm our findings using a published dataset covering E12.5 and E13.5 (ref. 34) (Supplementary Fig. 29). Next, we investigated the similarity of chromatin accessibility (Fig. 5g and Extended Data Fig. 8). The similarity between the ATAC profiles of epsilon progenitors, *Fev*[+] delta 0 cells, *Fev*[+] delta 1 cells, epsilon cells and delta cells corroborated the hypothesis that delta and epsilon cells have similar ancestries. Moreover, we observed notable similarities in chromatin accessibility in the promoter regions of both *Ghrl* (epsilon) and *Hhex*, a key regulatory TF of delta-cell formation[42] (Supplementary Fig. 30). To identify additional relevant chromatin regions, we performed differentially accessible peak analysis of the epsilon progenitor population (Supplementary Table 11) and the *Fev*[+] delta population (Supplementary Table 12). The findings showed that the peaks are co-accessible among the proposed ancestors of delta and epsilon cells (Supplementary Note 7 and Supplementary Fig. 31). Moreover, the expression of *Arx* as an alpha-cell determinant and the expression of *Pax4* as a beta-cell determinant supported the hypothesis of the high plasticity of *Fev*[+] delta cells (Fig. 5f and Supplementary Fig. 32).

To learn more about the regulatory mechanisms that drive delta and epsilon cell fate, we used moscot.time to find potential driver genes (Methods, Supplementary Tables 13–22 and Supplementary Fig. 33). The recovery of known driver genes such as *Arx* and *Mafa*[43] of the well-studied alpha and beta cells, respectively, validated the utility of this method (Supplementary Tables 13 and 14 and Supplementary Fig. 34). Notably, we identified NEUROD2 as the second most relevant TF for both the *Fev*[+] delta and the epsilon progenitor populations (Supplementary Tables 19 and 20). The expression of *Neurod2* was prominent in the epsilon progenitor and *Fev*[+] delta populations across developmental stages (Supplementary Fig. 35). Leveraging information from both RNA and ATAC datasets, we identified potential target genes of NEUROD2 (Supplementary Tables 23–29 and Supplementary Fig. 36). Several of these genes were also expressed in the epsilon lineage, such as *Lurap1l* and *Fam107b*, thereby implicating a potential regulatory function of NEUROD2 for epsilon cell-fate decisions. Although NEUROD1 can regulate islet-cell differentiation[44], the expression patterns of *Neurod1* and *Neurod2* are distinct during mouse endocrinogenesis (Supplementary Fig. 35) and in human induced pluripotent stem (iPS) cell differentiation[45], which indicated non-redundant and specific functions of these TFs. To experimentally validate our hypothesis, we used a human iPS cell differentiation system to generate endocrine islet cell types[45]. The differentiation of *NEUROD2* knockout (KO) iPS cells to stem-cell-derived islets resulted in a significant decrease in the number of ghrelin-expressing cells and reduced levels of *GHRL* mRNA when compared with an wild-type control iPS cell line (Fig. 5h,i and Extended Data Fig. 9). This result suggests that NEUROD2 has a role in directing epsilon-cell differentiation. At the same time, our previous[45] and current data indicated that NEUROD2 has no function in the specification of alpha, beta and delta cells, results in line with what has been reported in mice[46].

We leveraged orthogonal approaches to support the hypotheses of regulatory mechanisms using feature-sparse OT[47], differential feature analysis and motif analysis (Methods and Supplementary Figs. 37 and 38). Similarities in motif profiles indicate a similar cell state, as related TFs govern developmental trajectories. Owing to a temporal shift between the gene expression of a TF and its activity, profiling of motif activity and gene expression within the same sample might fail to recover regulatory mechanisms[48]. Moscot links gene expression at an earlier time point with motif activity in cells corresponding to the later

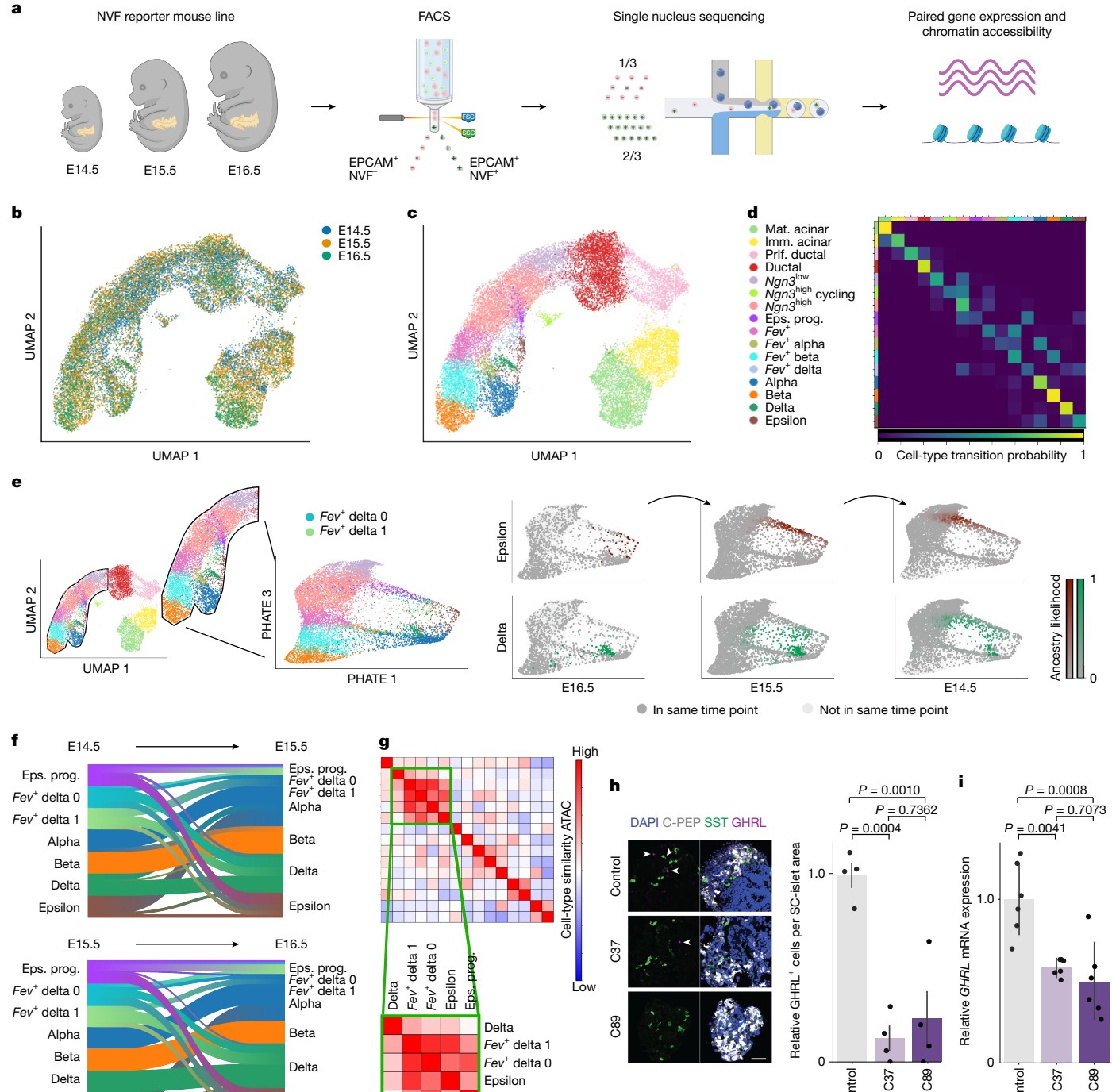

**Fig. 5 | Moscot reveals lineage ancestries of delta and epsilon cells.**
**a**, Schematic of the experimental protocol to generate paired gene expression
and ATAC data that capture the development of the mouse pancreas.
**b**,**c**, Multimodal UMAP join embedding, coloured by time (**b**) and cell-type
annotation (**c**) (Methods). **d**, Heatmap visualizing descendancy probabilities
of cell types in E14.5 as obtained using moscot.time. **e**, UMAP embedding
coloured as in **c**, including the refined *Fev*⁺ delta populations. The inset
highlights the cells that a PHATE embedding[41] is computed for. The top row
shows epsilon cells at E16.5 (left) as well as the progenitor population at E15.5
(middle) and E14.5 (right) as predicted by moscot. The bottom row shows the
corresponding plots for delta cells. **f**, Sankey diagram of the cell-type transitions
between E14.5 and E15.5 (top) and E15.5 and E16.5 (bottom). **g**, Similarity in ATAC
profile between different cell types (Methods). The green boxes highlight the

cell types for which ancestry was focused on. **h**, Representative confocal
microscopy images (left) and quantification (right) of ghrelin-expressing cells
in control and *NEUROD2* KO (C37 and C89) stem-cell-derived islets (SC islets)
at stage 6, day 14 (Methods). White arrowheads indicate GHRL⁺ cells. Scale
bar, 50 μm. *n* = 4 independent experiments, mean and s.e.m. reported.
**i**, Quantitative PCR analysis of expression levels of *GHRL* at stage 6, day 14
(*n* = 6 biologically independent samples). Data are represented as the mean
and s.d. (Methods). *P* values (**h**,**i**) were calculated using one-sided analysis
of variance test with Tukey's multiple comparison correction. Eps. prog.,
epsilon progenitors; FSC, forward scatter; imm., immature; mat., mature;
prlf., proliferating; SSC, side scatter. Panel **a** was created using BioRender
(https://www.biorender.com).

time point (Methods, Extended Data Fig. 10, Supplementary Note 8 and Supplementary Fig. 39). *Isl1* and *Tead1* had high motif activity in delta cells and epsilon cells, respectively, which was complemented by high gene expression in their progenitors (Supplementary Tables 30 and 31). The hypothesis of similar developmental trajectories of delta and epsilon cells was corroborated by the similarity of motif activity in their progenitors. We further supported this finding using established trajectory inference methods (Methods and Supplementary Figs. 40–43).

## Discussion

We presented moscot, a computational framework for mapping cellular states across time and space using OT. Unlike previous applications of OT, moscot incorporates multimodal information, scales to atlas-sized datasets and provides an intuitive and consistent interface. We accurately recovered mouse differentiation trajectories during embryogenesis[7,10], enriched spatial liver samples with multimodal information[26] and aligned brain tissue slides in datasets that were previously inaccessible with state-of-the-art techniques. Moreover, we presented an analysis approach for spatiotemporal data. Finally, we generated a multimodal developmental pancreas dataset that enabled us to predict that epsilon and delta cells have a similar trajectory in the pancreas. Using moscot, we identified candidates for lineage-specific TFs and confirmed the role of NEUROD2 as an epsilon-cell regulator in islet cells derived from human iPS cells.

Moscot will simplify future OT applications in single-cell genomics. With our unified API, incorporating other OT applications such as cross-modality data integration[49] becomes easier. The current approach of using discrete OT is well-suited for the applications described in this study and for the extensions outlined above. However, discrete OT is in general not applicable to out-of-sample data points. To overcome this limitation, neural OT has proved useful for modelling development[50–52] and perturbation responses[51–53] as well as translating modalities[52].

Given the widespread need to align cellular measurements in single-cell genomics, we anticipate that moscot will accelerate and simplify the analyses of large-scale multimodal datasets.

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

## Methods

### The moscot algorithm

**OT for single-cell genomics.** OT is an area of mathematics that is concerned with comparing probability distributions in a geometry-aware manner[1]. OT-based tools have been successfully applied to various problems that arise in single-cell genomics, including mapping cells across time points[2,50–52,54–57], mapping cells from molecular to physical space[3], aligning spatial transcriptomic samples[4], integrating data across molecular modalities[49,52], learning patient manifolds[58] or mapping cells across different experimental perturbations[53,59]. Despite such success, the widespread adaptation of OT-based tools in single-cell genomics faces three key challenges.

First, most current OT-based tools are geared towards a single modality and cannot use the added information provided by multimodal assays. Second, computation time and memory consumption quadratically scale in cell number for vanilla OT and cubically for Gromov–Wasserstein extensions[6]. Such poor scalability limits the application of these tools to datasets that contain millions of cells. Third, the landscape of OT-based tools is split across programming languages and softwares that provide OT algorithms, which results in a fractured landscape of incompatible APIs. This makes it difficult for users to adapt and for developers to create new tools. By contrast, user-friendly and extensible APIs accelerate and facilitate research, as demonstrated through the scVI-tools framework[60].

**Moscot unlocks the full power of OT for spatiotemporal applications.** Our method is built on three key design principles to overcome limitations and unlock the full potential of OT for single-cell applications: multimodality, scalability and consistency. For multimodality, all moscot models extend to multimodal data, including CITE-seq and multiome (RNA and chromatin accessibility) data. For scalability, we use both engineering and methodological innovations to overcome scalability limitations; in particular, we reduce computation time and memory consumption so that they are linear in the number of cells. For consistency, our implementation unifies temporal, spatial and spatiotemporal problems through a consistent API that interacts with the wider scverse[8] ecosystem and is easy to use. Solving any of these problems in moscot follows a common pattern that translates the biological problem into an OT problem that is solved by the OTT backend[12].

In the sections below, we describe how we realize these principles for temporal, spatial and spatiotemporal applications.

### Moscot.time for mapping cells across time

**Model rationale, inputs and outputs.** Biologists frequently use time-series experiments to study biological processes such as development or regeneration that are not in a steady state. As current single-cell assays usually involve the destruction of cells, such experiments result in disparate molecular profiles measured at different time points. As previously suggested[2], OT can be used to probabilistically link cells from early to late time points. We follow the WOT model in assuming that cells collectively minimize the distance they travel in phenotypic space and that cellular fate decisions are Markov; that is, cellular fate depends only on the current state and not on earlier history. Previous methods had limited scalability and were only applied to gene expression. We outline below how moscot.time overcomes these limitations.

Let $X \in R^{N \times D}$ and $Y \in R^{M \times D}$ represent pairs of state matrices for $N$ and $M$ cells observed at early ($t_1$) and late ($t_2$) time points, respectively. State matrices $X$ and $Y$ may represent, for example, gene expression (scRNA-seq) or chromatin accessibility (scATAC-seq) across $D$ features (for example, genes or peaks). Optionally, the user may provide marginal distributions $a \in \Delta_N$ and $b \in \Delta_M$ over cells at $t_1$ and $t_2$ for probability simplex $\Delta_N := \{a \in R_+^N \mid \sum_{i=1}^N a_i = 1\}$. Any previous cell-level information may be represented through the marginals, including cellular growth rates and death rates.

The key output of moscot.time is a coupling matrix $P \in U(a, b)$, where $U(a,b)$ is the set of feasible coupling matrices, defined by

$$U(a, b) := \{P \in R_+^{N \times M} \mid P1_M = a, P^\top 1_N = b\} \tag{1}$$

for constant one vector $1_N = [1, \ldots, 1]^\top \in R^N$. We link $t_1$ cells to $t_2$ cells through the coupling matrix $P$; the $i$th row $P_{i:}$ represents the amount of probability mass transported from cell $i$ at $t_1$ to any $t_2$ cell. The set $U(a,b)$ contains the coupling matrices $P$ that are compatible with the user-provided marginal distributions $a$ and $b$ at $t_1$ and $t_2$, respectively.

These definitions enabled us to formalize the aim of moscot.time: we sought to find a coupling matrix $P \in U(a, b)$ that couples $t_1$ cells to $t_2$ cells such that their overall travelled distance in phenotypic space is minimized.

**Model description.** To quantify the distance that cells travel in phenotypic space between time points, let $c(x_i, y_i)$ be a cost function for early ($x_i$) and late ($y_j$) molecular profiles, representing, for example, gene expression or chromatin-accessibility state. Moscot enables the use of various cost functions (Supplementary Note 5). We use the cost function $c$ to measure cellular distances in a modality-specific, shared latent space, for example, principal component analysis (PCA) for gene-expression data, latent semantic indexing (LSI) for ATAC data or corresponding models of scVI-tools[60].

We evaluated the cost function $c$ for all pairs of cells $(i,j) \in \{1, \ldots, N\} \times \{1, \ldots, M\}$ to form the cost matrix $C \in R_+^{N \times M}$. Given the cost matrix $C$, which quantifies distances along the phenotypic manifold, we solved the optimization problem

$$P^* = \mathrm{argmin}_{P \in U(a,b)} \langle C, P \rangle = \mathrm{argmin}_{P \in U(a,b)} \sum_{ij} C_{ij} P_{ij} \tag{2}$$

known as the Kantorovich relaxation of OT[1], where $P^*$ is the optimal coupling matrix. When using $P^*$ to transport $t_1$ cells to $t_2$ cells, we accumulated the lowest cost according to $C$. Subsequently, we refer to this type of OT problem as a W-type OT problem.

**Introducing entropic regularization.** In practice, the OT problem of equation (2) is usually not solved directly because it is computationally expensive, and the solution has statistically unfavourable properties. Instead, it is more common to consider a regularized version of the problem[5],

$$P = \mathrm{argmin}_{P \in U(a,b)} \langle C, P \rangle - \epsilon H(P) \tag{3}$$

for entropy regularizer

$$H(P) = - \sum_{ij} P_{ij} (\log P_{ij} - 1) \tag{4}$$

The parameter $\epsilon > 0$ controls the regularization strength. Intuitively, entropic regularization introduces uncertainty to the solution in that it has a 'blurring' effect on $P^*$. Mathematically, it renders the problem $\epsilon$ strongly convex, differentiable and less prone to the issue of dimensionality.

**The Sinkhorn algorithm for optimization.** It can be shown that the solution to the regularized W-type problem of equation (3) has the form $P_{ij} = u_i K_{ij} v_j$ for Gibbs kernel

$$K := \exp(-C_{ij}/\epsilon) \tag{5}$$

and unknown scaling variables $(u, v) \in R_+^N \times R_+^M$. Using this formulation, we rewrote the constraints $P1_M = a$ and $P1_N = b$ of equation (1) to produce

$$u \odot (Kv) = a, v \odot (K^\top u) = b,$$

where $\odot$ denotes element-wise multiplication. Iteratively solving these equations gave rise to Sinkhorn's algorithm:

$$u^{(l+1)} := \frac{a}{Kv^{(l)}}, v^{(l+1)} := \frac{b}{K^\top u^{(l+1)}}, \tag{6}$$

where the division is applied element-wise, and $l$ is the iteration counter. Using this algorithm, the (unique) solution to the regularized W-type problem of equation (3), corresponding to the optimal coupling of $t_1$ cells to $t_2$ cells, was computed in time and memory quadratic in cell number[5].

**Adjusting the marginals for growth and death.** Cells differentiate, proliferate and die as the biological process unfolds between time points $t_1$ and $t_2$. The coupling matrix $P^*$, computed by solving equation (3), reflects a mixture of these effects. To disentangle proliferation and apoptosis from differentiation, we adjusted the left marginal $a$ for cellular growth and death. Specifically, we followed WOT[2] in defining

$$a_i = \frac{g(x_i)^{t_2-t_1}}{\sum_{j=1}^N g(x_j)^{t_2-t_1}} \forall i \in \{1, \dots, N\} \tag{7}$$

where $g: R^D \to R_+$ corresponds to the expected value of a birth–death process $g(x) = e^{\beta(x)-\delta(x)}$ with proliferation at rate $\beta(x)$ and death at rate $\delta(x)$. We estimated growth rates and death rates from curated marker gene sets. Note that moscot comes with predefined gene sets for mice and humans. Intuitively, our adjustment enabled $t_1$ cells that are likely to proliferate or die to distribute more or less probability mass, respectively, to $t_2$ cells. In the absence of cellular growth and death, every $t_1$ cell would be allowed to distribute $1/N$ probability mass; thus, values greater or smaller than $1/N$ indicate proliferation or apoptosis, respectively. For the right marginal $b$, we assigned uniform weights $b_j = 1/M, \forall j \in \{1, \dots, M\}$. Such an adjustment encouraged the optimal coupling matrix $P^*$ to reflect differentiation rather than proliferation and apoptosis.

As it is difficult to adjust the hyperparameters of the birth–death problem, we also implemented a more intuitive and more easily adjustable estimation of the growth rates using

$$a_i = \exp\frac{p_i - q_i}{c} \tag{8}$$

where $p_i$ denotes a proliferation score and $q_i$ an apoptosis score, obtained using scanpy.tl.score_genes. $c$ denotes a scaling parameter.

**Unbalancedness to account for biased sampling.** Our formulation of equation (3) enforced the prespecified marginals $a$ and $b$ to be exactly met by the solution $P^*$. This is problematic from two perspectives.

First, the cells profiled at each time point usually correspond to a sample from the overall population. That is, small variations in cell-type frequencies across time points do not necessarily reflect underlying differentiation but might result from stochastic cell sampling. Exactly enforcing the marginals therefore implies that we encode the sampling effect in the coupling, which confounds the actual differentiation signal.

Second, our growth-adjusted and death-adjusted marginals of equation (7) are unlikely to reflect ground-truth proliferation or apoptosis rates, as they are estimated using noisy gene expression data and do not include any post-transcriptional effects. Thus, exactly enforcing these marginals propagates noise into the coupling matrix $P^*$.

To avoid both pitfalls, we followed WOT[2] to allow small deviations from the exact marginals in an unbalanced OT framework[61]. Specifically, we replaced the hard constraint $P \in U(a, b)$ with soft Kullback–Leibler (KL)-divergence penalties,

$$P^* := \text{argmin}_{P \in R_+^{N \times M}} \langle C, P \rangle + \frac{\epsilon\tau_a}{1-\tau_a}\text{KL}[P1_M || a]$$
$$+ \frac{\epsilon\tau_b}{1-\tau_b}\text{KL}[(P^\top 1_N || b)] - \epsilon H(P) \tag{9}$$

which may be solved at the same computational complexity level using a generalization of Sinkhorn's algorithm. The parameters $\tau_a, \tau_b \in (0,1)$ are hyperparameters that determine the weight we gave to complying with the left and right marginals $a$ and $b$, respectively. Values near one or zero correspond to strict or weak marginal penalties, respectively.

**Multimodal data and scalability.** The model we presented in the previous section is similar to the WOT[2] model. However, WOT is only applied to unimodal data and has quadratic time and memory complexity in the number of cells, which largely prevented its application to atlas-scale temporal datasets that contain multiple modalities. This section presents how we extended the moscot.time model to overcome these limitations.

**Application to multimodal data.** We incorporated multimodal data in moscot.time through an adjusted definition of the cost function. Intuitively, we used a joint representation to render the computed distances more faithful to the phenotypic manifold. Specifically, given bimodal representations $(X^{(1)}, X^{(2)})$ and $(Y^{(1)}, Y^{(2)})$ at $t_1$ and $t_2$, respectively, we scaled these to have the same variance and measured distances in a concatenated space. In this example, (1) and (2) can represent any pair of modalities, for example, gene expression and ATAC data. This strategy naturally extends beyond two modalities to any number of jointly measured modalities, which makes moscot.time truly multimodal. Alternatively, moscot.time can be applied to representations computed using shared latent-space-learning techniques, for example, from variational autoencoders (VAEs)[60,62,63].

**Scalability through engineering-type innovations.** Moscot.time builds on OTT[12] in the backend, which offers three key engineering-type improvements: online evaluation of the cost function; GPU execution; and just-in-time compilation (jitting).

Although memory complexity of the Sinkhorn algorithm is quadratic, it can be reduced to linear through online-cost matrix evaluation with minor assumptions on the cost function. The key observation is that the Sinkhorn algorithm only accesses the cost matrix $C$ through the matrix–vector products $Kv$ and $K^\top u$ (equation (6)), which are evaluated row by row. Thus, the cost function $c$ can be queried on the fly for those cell–cell distances that are required to evaluate the current row of the matrix–vector product. Online evaluation reduces the memory complexity so that it is linear in cell number (first improvement)[12].

Second, although the Sinkhorn algorithm can, in principle, be run on GPUs to accelerate optimization, the quadratic memory complexity prevents this in practice. Although CPUs can handle large memory consumption, GPUs are usually more limited (typically around 40 GB). Online memory evaluation (first improvement) renders GPU acceleration possible, and OTT implements it in practice. Performing computations on GPUs accelerates the computation of cell–cell couplings in moscot.time (second improvement).

Third, jitting compiles Python code before it is executed for the first time, which further reduces computation time (third improvement).

Combining these three engineering-type innovations enabled moscot.time to run datasets that contain a few hundred thousand cells per time point with linear memory and quadratic time complexity on modern GPUs. However, if millions of cells per time point are involved, the quadratic time complexity becomes prohibitive.

**Scalability through methodological innovations.** To enable the application of moscot.time to datasets that contain millions of cells per time point, we must overcome the quadratic time complexity in

the number of cells. Following previous work[13–15], we achieved this by imposing low-rank constraints on the set of feasible couplings. That is, requiring $P \in U(a, b, r)$ for non-negative coupling matrix rank $r$ (Supplementary Note 3). Such regularization led to linear time and memory complexity in the number of cells. Low-rank Sinkhorn was implemented in OTT and available through moscot.time, which enables the application to future atlas-scale developmental studies.

**Downstream applications.** The coupling matrix $P^*$ optimally links $t_1$ cells to $t_2$ cells for the cost function $c$. Moscot.time uses coupling to relate cellular states and to derive insights about putative driver genes. Thus, consider a $t_1$ cell state $P$ of interest, where $P$ is the set of corresponding cell indices. This state may represent, for example, a rare or transient cell population. Define the corresponding normalized indicator vector $p \in \{0, 1\}^N$ through

$$p_i := \frac{1}{|P|} i \in P, \text{ else } 0 \tag{10}$$

for $t_1$ cell $i$ and $|P|$, the number of cells in state $P$. Following the original suggestion in WOT, we computed $t_2$ descendants of cell state $P$ by a push-forward operation of $P$,

$$q = P^\top p \tag{11}$$

where $q \in R_+^M$ describes the probability mass that cell state $P$ distributes to $t_2$ cells. Using $P$ rather than its transpose, we analogously computed ancestors of a cell state $Q$ at $t_2$. For a global view of cell-state transitions, we aggregated pull and push operations over all states into transition matrices, which we visualized using heatmaps or Sankey diagrams. We also correlated pull and push distributions with gene expression to uncover putative driver genes.

In summary, we used pull and push operations based on our computed transport matrix $P$ to recover putative ancestors and descendants, respectively, of cell states of interest. In biological terms, for a given $t_1$ cell state $P$, we interpreted its push distribution over $t_2$ cells as the likelihood of $P$ giving rise to these cells. Analogously, for a given $t_2$ cell state $Q$, we interpreted its pull-back distribution over $t_1$ cells as the likelihood of these cells to give rise to $Q$. Accordingly, we correlated gene expression with the density of the pull-back distribution to pinpoint putative driver genes of transitioning into state $Q$. Using positive and negative correlations, such a strategy will reveal consistently upregulated or downregulated genes, respectively, in cells that are likely to transition to state $Q$.

**Coupling more than two time points.** Following the WOT model, we coupled several time points by assuming that the state of $t_{r+1}$ cells depends only on the state of $t_r$ cells and not on any other earlier or later states. The index $r$ runs over time points, $r \in \{1, \ldots, R\}$, for $R$ time points. This Markov assumption enabled us to chain together time points by matrix multiplication. For time points $\{t_1, \ldots, t_R\}$ and corresponding sequential coupling matrices $\{P^{(1)}, \ldots, P^{(R-1)}\}$, we linked $t_1 t_1$-cells to $t_r$ cells by matrix multiplication, $P^{(1)} P^{(2)} \ldots P^{(R-1)}$.

**Feature-sparse OT maps using Sparse Monge.** Sparse Monge[47] is a method to perform (linear) OT in high-dimensional spaces while selecting only the most relevant genes per single cell. The concept builds on entropic maps, which enabled the estimation of deterministic Monge maps from discrete entropy-regularized OT couplings. Given the dual potential $g_\varepsilon$ corresponding to the target cells (obtained using the output of the Sinkhorn algorithm), the entropic map $T_\varepsilon$ for the squared Euclidean cost[64] is defined as

$$T_\varepsilon(x) = \frac{\frac{1}{n}\sum_{i=1}^n Y_i e^{\frac{1}{\varepsilon}\left(g_\varepsilon(Y_i) - \frac{1}{2}\|x - Y_i\|^2\right)}}{\frac{1}{n}\sum_{i=1}^n e^{\frac{1}{\varepsilon}\left(g_\varepsilon(Y_i) - \frac{1}{2}\|x - Y_i\|^2\right)}} \tag{12}$$

Sparse Monge extends the entropic map estimators to more general costs. That is, translation-invariant costs of the form $c(x, y) = h(x - y)$ with $h : R^d \to R$. In particular, this enabled us to choose sparsity-inducing costs. While we refer to the original publication[47] for a more comprehensive list of such costs, we here restrict it to the elastic L1 cost given as

$$h(z) = \frac{1}{2}\|z\|_2^2 + \gamma\|z\|_1 \tag{13}$$

with $\gamma$ denoting the scaling regularizer. Thus, the entropic map estimator is given as

$$T_\varepsilon(x) = x - \text{ST}_\gamma\left(x - \sum_{j=1}^m p^j(x)(y^j + \gamma\text{sign}(x - y^j))\right)$$

where the soft threshold operator is defined as

$$\text{ST}_\gamma(z) = (1 - \gamma/|z|)_+ \odot z$$

and the weights are the factors as given in equation (12)

$$p^j(x) := \frac{\exp(-(h(x - y^j) - g_j)/\varepsilon)}{\sum_{k=1}^m \exp(-(h(x - y^k) - g_k)/\varepsilon)}.$$

## Moscot.space.mapping for spatial reference mapping

**Model rationale, inputs and outputs.** Techniques to simultaneously measure the spatial context of a cell and its transcriptional state have matured in recent years. In particular, spatial resolution, the field of view and the number of profiled transcripts have increased[22,65]. However, current approaches still fall short of measuring the full transcriptome at true single-cell resolution. This experimental difficulty has fuelled the development of a range of computational tools that map dissociated single-cell reference datasets onto spatial coordinates, a problem known as spatial mapping[23,66–68]. Solving a spatial-mapping problem can provide two types of information.

The first is an annotation-centric perspective, whereby spatial mapping annotates cell types using single-cell-resolved spatial transcriptomic technologies (for example, MERFISH[69] and Seqfish[70]). The second is a feature-centric perspective, whereby spatial mapping imputes unmeasured gene expression in the spatial domain for techniques that do not achieve full transcriptome coverage (for example, MERFISH[71] and seqFISH+).

As previously suggested in NovoSpaRc[3], a variant of OT can be used to probabilistically map reference cells into the spatial domain. We followed the NovoSpaRc model in assuming that cells in physical proximity tend to have similar gene-expression profiles. In other words, we assumed that there exists a (possibly noisy and imperfect) correspondence between physical and expression distances. Previous approaches faced several limitations, including scalability, applicability beyond gene-expression reference data and incorporation of spatial information in the mapping problem. With moscot.space.mapping, we produced a model that applies to both the sample-centric and feature-centric perspectives, scales to large datasets and incorporates multimodal information. Moreover, moscot.space.mapping explicitly makes use of spatial information when solving the mapping problem.

Let $X \in R^{N \times D_x}$ and $Y \in R^{M \times D_y}$ represent a pair of state matrices for $N$ cells and $M$ samples (for example, cells, spots, among others) observed in the dissociated reference and the spatial dataset, respectively. We assumed state matrices to represent gene expression for different numbers of genes, $D_x$ for the dissociated reference and $D_y$ for the spatial dataset. We allowed further multimodal information in $X$, for example, from joint RNA and ATAC readouts[72–74]. In addition, let $C^Y \in R_+^{M \times M}$ encode spatial similarity among the $M$ samples in $Y$ (we define $C^X$ below).

Depending on the spatial technology, $C^Y$ contained either Euclidean distances among spatial locations or similarities in spatial graphs. Optionally, as in moscot.time, the user may provide marginal distributions $a \in \Delta_N$ and $b \in \Delta_M$ over cells in the dissociated reference and samples in the spatial dataset. In the context of moscot.space.mapping, these may represent sample-level uncertainties or estimated cell numbers per spot in the spatial dataset for barcoding-based spatial technologies.

The key output of moscot.space.mapping is a coupling matrix $P \in U(a, b)$ that links cells in the dissociated reference with samples in the spatial dataset. In particular, the $i$th row $P_{i,:}$ represents the amount of probability mass transported from cell $i$ in the reference to any spatial sample $j$.

These definitions enabled us to formalize the aim of moscot.space.mapping: we sought to find a coupling matrix $P \in U(a, b)$ that related reference cells with spatial samples such that their distance in the shared transcriptome space is minimized while the correspondence between molecular and spatial similarity is maximized.

**Model description.** To quantify the global distance between the reference and spatial datasets in the shared transcriptome space, we followed moscot.time and defined a cost function $c(x_i, y_j)$ and associated cost matrix $C \in R_+^{N \times M}$. The matrix $C$ quantified expression distance in raw gene space or a shared latent space computed using PCA or scVI. Note that the shared latent space was constructed using only those genes that had been measured in both the dissociated reference and the spatial dataset.

**Gromov–Wasserstein for structural correspondence.** In NovoSpaRc[3], the authors showed how introducing a structural correspondence assumption between gene expression and spatial information enhanced their ability to accurately solve the spatial-mapping problem. In particular, they assumed that cell pairs should be coupled such that there is a correspondence between distances in gene expression and distances in physical space. Following their suggestion, we encoded the structural correspondence assumption in a GW-type OT problem,

$$P^* := \text{argmin}_{P \in U(a,b)} \sum_{ijkl} L(C_{ij}^X, C_{kl}^Y) P_{ik} P_{jl} \qquad (14)$$

for spatial distance matrix $C^Y \in R_+^{M \times M}$, defined as above, and reference distance matrix $C^X \in R_+^{N \times N}$, quantifying molecular similarity among cells in the dissociated reference. To compute $C^X$, we measured the expression distance among reference cells in a latent space defined using PCA or scVI. Correspondence between $C^X$ and $C^Y$ was quantified entry-wise using the cost function $L$, which was set to the squared Euclidean cost by default. This cost was evaluated element-wise; that is, $L(C_{ij}^X, C_{kl}^Y) = (C_{ij}^X - C_{kl}^Y)^2$.

Intuitively, the GW-type problem aimed to find a coupling matrix to maximize the structural correspondence between gene expression and spatial information. Note that individual genes may still show sharp gradients in the spatial domain, and the structural correspondence assumption applies to aggregated molecular profiles.

**The moscot.space.mapping model.** The moscot.space.mapping model is a combination of the W term, which quantifies the expression distance between the reference and the spatial dataset, and the GW term, which quantifies the structural correspondence between the reference and the spatial dataset, to create a FGW-type OT problem[11],

$$P^* := \text{argmin}_{P \in U(a,b)} \alpha \sum_{ijkl} L(C_{ij}^X, C_{kl}^Y) P_{ik} P_{jl}$$
$$+ (1 - \alpha) \sum_{ik} C_{ik} P_{ik} - \epsilon H(P), \qquad (15)$$

where we added entropic regularization at strength $\epsilon$ and introduced the weight parameter $\alpha$ to control the relative contribution of the

W term and the GW term. The objective function contained the following cost matrices:
- $C \in R_+^{N \times M}$ compares reference cells with spatial samples in terms of expression in shared genes.
- $C^X \in R_+^{N \times N}$ compares reference cells among each other in terms of gene expression.
- $C^Y \in R_+^{M \times M}$ compares spatial samples among each other in terms of spatial distance.

We optimized the moscot.space.mapping objective function of equation (15) using a mirror descent scheme[6] (Supplementary Note 1). To account for uneven cell-type proportions between the reference and the spatial datasets, we optionally allowed for unbalancedness in the FGW-type problem[75].

**Multimodal data and scalability.** The model presented here is an extension of the NovoSpaRc[3] model, which is restricted to a certain cost function and only supports feature-centric interpretation. Furthermore, NovoSpaRc is only applicable to unimodal data and has cubic time complexity and quadratic memory complexity in the number of cells, which largely prevents its application to atlas-scale spatial datasets and references that contain multiple modalities. This section extends the moscot.space.mapping model to overcome these limitations.

**Multimodal reference datasets.** Multimodal data contains additional information about the molecular state of cell that can guide the mapping process. Although previous methods could apply mapping learnt from gene-expression data to other modalities collected for the same cells[23], moscot.space.mapping is different because it makes use of multimodal information in the actual mapping problem. In other words, our approach uses multimodal information when learning the mapping rather than learning the mapping based on unimodal data and subsequently applying it to jointly captured modalities.

Consider a dissociated reference dataset with multimodal data matrices $X^{(R)}$ and $X^{(O)}$, where $R$ refers to gene expression and $O$ refers to another modality, for example, chromatin accessibility[76] or surface marker expression[77,78]. We constructed the across-space cost matrix $C$ and the spatial cost matrix $C^Y$ as before but modified the construction of the reference cell cost matrix $C^X$. Similar to moscot.time, we concatenated joint representations or used joint latent-space-learning techniques[62,79,80] to obtain a single molecular representation and to measure distances in this representation to define $C^X$. Our multimodal approach enabled the learning a more faithful correspondence between molecular similarity in the dissociated reference dataset and spatial proximity in the spatial dataset.

**Atlas-scale spatial mapping.** For the squared Euclidean loss function $L$ and within-space cost functions $C^X$ and $C^Y$, we implemented moscot.space.mapping to have quadratic time and memory consumption by exploiting low-rank properties of the Euclidean distance (Supplementary Note 2). Similar to moscot.time, solving our FGW-type problem in the backend using OTT granted us GPU execution and jitting. Although this led to good performance on datasets of intermediate size (approximately 10,000 cells in reference and spatial datasets), the quadratic scaling became prohibitive for atlas-scale datasets.

To overcome the quadratic time and memory complexities, we made use of a recently proposed low-rank GW formulation[14,15] (Supplementary Note 2), which extended the original low-rank Sinkhorn formulation (Supplementary Note 3). This enabled moscot.space.mapping to relate hundreds of thousands of dissociated reference cells to spatial locations.

**Downstream applications.** Moscot.space.mapping supports both sample and feature-centric downstream analysis techniques.

**Annotation-centric perspective.** In this perspective, we had cell-type or cell-state labels available in the reference, which we used to map to the spatial dataset. Suppose we are given a set of one-hot encoded reference labels through the matrix $F \in \{0,1\}^{N \times S}$ for $S$ cell types or states. We obtained annotated cell types in the spatial domain using the matrix $G = P^\top F \in R_+^{M \times S}$. For each spatial sample $j$, the row $G_{j,:}$ contained the mapped cell-type likelihood for each of the $S$ cell types or states. We could then assign discrete cell types to the spatial sample by either taking the label of the most likely match in the dissociated samples or by taking the most likely element of the transport matrix aggregated to cell type level.

**Feature-centric perspective.** In this perspective, we had more genes measured in the dissociated reference dataset than in the spatial dataset. We aimed to use the solution of the mapping problem to impute spatial gene expression. This setting is relevant for spatial technologies that do not achieve full transcriptome coverage. Let $\widetilde{Y} \in R^{M \times D_x}$ denote inferred expression in the spatial domain; it holds

$$\widetilde{Y} = P^\top X \tag{16}$$

Analogous definitions hold for additional modalities collected in the dissociated reference; for example, we can use equation (16) to map chromatin accessibility or surface-marker expression into spatial coordinates.

To facilitate further downstream analyses of mapped spatial data, moscot.space.mapping interfaces with squidpy[81], a spatial analysis toolkit that contains various visualization and testing capabilities. For example, squidpy can be used to test for the spatial enrichment of mapped cell-type annotations or to quantify spatial variability of imputed gene expression.

**Moscot.space.alignment for aligning spatial transcriptomic data**
**Model rationale, inputs and outputs.** The rapidly increasing number of spatial datasets poses substantial data-analysis challenges. In particular, faithful integration of spatial data across tissue slides, individuals and laboratories is currently an open problem that limits our ability to study tissue architecture across scales[22,82]. Different terms exist to refer to spatial integration problems; here we prefer to speak of spatial alignment. Solving a spatial alignment problem can serve two principal objectives: joint analysis and 3D construction.

Joint analysis aligns spatial datasets against a CCF[82], which enabled us to gain statistical power by jointly considering multiple samples and enable new types of analysis, such as expression variability in space. Aligning data against CCFs will be a crucial step towards building spatial atlases. For 3D reconstruction, aligning sequential adjacent tissue sections enabled us to build faithful 3D tissue models.

As previously suggested in PASTE[4], FGW-type OT[11] can be used to probabilistically align spatial datasets. However, the previous PASTE method was targeted to small-scale 10x Visium datasets, and the authors considered a maximum of 4,000 spots per sample in their applications[83]. The scalability of PASTE is limited because it cannot run on GPUs and does not make use of entropic regularization, jitting or recent low-rank formulations of FGW-type OT. Furthermore, PASTE is limited to adjacent Visium tissue slides from the same individual because it cannot handle varying cell-type proportions. Moreover, the approach does not make use of multimodal molecular readout.

With moscot.space.alignment, we produced an approach that overcomes these limitations. In particular, moscot.space.alignment scales to large and diverse spatial datasets through GPU acceleration, entropic regularization[6], jitting[84] and low-rank factorizations[14,15]. Our approach can integrate samples from different individuals with varying cell-type proportions through an unbalanced formulation and applies to spatial technologies beyond 10x Visium, including assays that use in situ sequencing or situ hybridization.

Furthermore, our approach makes use of multimodal information if available.

Let $X \in R^{N \times D_x}$ and $Y \in R^{M \times D_y}$ represent a pair of state matrices for $N$ and $M$ spatial samples observed in two spatial datasets. We refer to $X$ and $Y$ as the left and right datasets, respectively. We assumed that state matrices represent gene expression for varying gene numbers $D_x$ and $D_y$. Optionally, we allowed additional multimodal readout at both left and right datasets. In addition, let $C^X \in R_+^{N \times N}$ and $C^Y \in R_+^{M \times M}$ encode spatial similarity among the $N$ samples in $X$ and the $M$ samples in $Y$, defined through, for example, Euclidean distance in space or similarities in spatial graphs[22,81]. Optionally, as in previous moscot models, the user may provide marginal distributions $a \in \Delta_N$ and $b \in \Delta_M$ over spatial samples in left and right datasets. In the context of moscot.space.alignment, these may represent sample-level uncertainties or estimated cell numbers per spot for barcoding-based spatial technologies[85–87].

The key output of moscot.space.alignment is a coupling matrix $P \in U(a, b)$ that links spatial samples across the two datasets. In particular, the $i$th row $P_{i,:}$ represents the amount of probability mass transported from spatial sample $i$ in the left dataset to any spatial sample $j$ in the right dataset.

These definitions enabled us to formalize the aim of moscot.space.alignment: we sought to find a coupling matrix $P \in U(a, b)$ that relates spatial samples across left and right datasets such that their distance in the shared transcriptome space is minimized while the correspondence between spatial arrangements is maximized.

**Model description.** To quantify the global distance between left and right datasets in the shared transcriptome space, we followed previous moscot models and defined a cost function $c(x_i, y_j)$ and associated cost matrix $C \in R_+^{N \times M}$. The matrix $C$ quantifies expression distance in a shared latent space computed using PCA or scVI[60]. Using the transcriptome–cost matrix $C$ in the W term and the spatial–cost matrices $C^X$ and $C^Y$ in the GW term, we defined a FGW-type OT problem as for moscot.space.mapping (equation (15)) and solved it using the mirror descent scheme (Supplementary Note 1). For samples with varying cell-type proportions, we optionally allowed for unbalancedness.

**Multimodal data and scalability.** We included additional multimodal data collected at left and right datasets in the W term. In particular, we followed moscot.time and used concatenated representations or joint latent space learning techniques.

We used the same scalability improvements as for moscot.space.mapping. In particular, we achieved fast runtimes on datasets of intermediate size through GPU acceleration and jitting. For atlas-scale left and right datasets, we used low-rank factorizations to achieve linear time and memory complexity (Supplementary Note 2).

**Downstream applications.** Moscot.space.alignment supports both joint analysis of several spatial datasets in a CCF and 3D reconstruction of adjacent tissue sections through different alignment policies.

For joint analyses of several spatial datasets, we relied on a predefined CCF. To define such a CCF, one may either use a dedicated computational method or manually designate a spatial sample to serve as the CCF. Given a CCF $X \in R^{N \times D_x}$ and $R$ query datasets $Y^{(r)} \in R^{M_r \times D_r} \forall r \in \{1, \ldots, R\}$, moscot.space.alignment solves a star-policy alignment problem whereby each query $Y^{(r)} Y^{(r)}$ is aligned against the central CCF $X$. To enable joint analysis of all query datasets $Y^{(r)}$ in terms of CCF spatial coordinates, we computed the projection

$$\widetilde{Y}^{(r)} = P^{(r)} Y^{(r)},$$

for projected gene expression $\widetilde{Y}^{(r)} \in R^{N \times D_r}$ and corresponding coupling matrix $P^{(r)} \in R_+^{N \times M_r}$. Solving the star-policy alignment problem with

moscot.space.alignment and projecting into CCF coordinates enabled the joint analysis of all spatial query samples $\{Y^{(1)}, \ldots, Y^{(R)}\}$.

For 3D reconstruction of adjacent tissue sections, let $X^{(r)} \in R^{N_r \times D_r}$ represent gene expression of slide $r$ for $N_r$ spatial samples and $D_r$ genes. Furthermore, let $Z^{(r)} \in R^{N_r \times 2}$ represent the corresponding spatial coordinates. We considered $R$ sequential slides, $r \in \{1, \ldots, R\}$. To align their coordinate systems, moscot.space.alignment solves a sequential policy alignment problem, whereby each dataset $X^{(r)}$ is aligned against the next dataset $X^{(r+1)}$ in the sequence. Given the corresponding coupling matrix $P^{(r)} \in R_+^{N_r \times N_{r+1}}$, slide $(r+1)$ coordinates are transformed into slide $r$ coordinates using

$$\widetilde{Z}^{(r+1)} = P^{(r)} Z^{(r+1)} \tag{17}$$

for $\widetilde{Z}^{(r+1)} \in R^{N_r \times 2}$. We refer to this as the warping transformation because it nonlinearly warps $Z^{(r+1)}$ coordinates onto $Z^{(r)}$ coordinates. Alternatively, moscot.space.alignment implements the previously suggested affine-linear transformation. We recommend the warping transformation whenever nonlinear effects between adjacent slides are expected. By designating any reference slide $r^*$, all other coordinate systems can be transformed into $Z^{(r^*)}$ coordinates through sequential application of either the warping or the affine transformation.

In either case of the alignment problem, it is possible to further refine the alignment by solving an additional W-type problem on the spatial coordinates.

We interfaced with squidpy[81] for further joint analyses of several spatial datasets in a CCF. For example, squidpy can be used to study expression heterogeneity at a defined spatial location in the CCF across several spatial datasets.

### Moscot.spatiotemporal to decipher spatiotemporal variation

**Model rationale, inputs and outputs.** Cellular state-change processes, including development, regeneration and reprogramming, do not unfold in isolation in single cells but in constant communication with the surrounding tissue[22]. Recent experimental advancements have enabled spatially resolved gene expression measurements at near single-cell resolution across developmental processes. In particular, the Stereo-seq[10] technology has been applied to various developmental settings[10,33,88–90]. These experiments produce a time series of gene expression measurements (as in moscot.time), with additional spatial readouts at each time point. With moscot.spatiotemporal, we developed a method to map cells across time points while preserving spatial organization, which enabled us to decipher spatiotemporal variation during complex cell-state changes.

Let $X \in R^{N \times D}$ and $Y \in R^{M \times D}$ represent pairs of state matrices for $N$ and $M$ spatial samples observed at early ($t_1$) and late ($t_2$) time points, respectively. In addition, as stated for moscot.space.alignment, let $C^X \in R_+^{N \times N}$ and $C^Y \in R_+^{M \times M}$ encode spatial similarity among the $N$ samples in $X$ and the $M$ samples in $Y$. Optionally, as in previous moscot models, the user may provide marginal distributions $a \in \Delta_N$ and $b \in \Delta_M$ over cells at $t_1$ and $t_2$. In the context of moscot.spatiotemporal, these usually correspond to cellular growth and death rates.

The key output of moscot.spatiotemporal is a coupling matrix $P \in U(a, b)$ that links samples across the two time points. In particular, the $i$th row $P_{i,:}$ represents the amount of probability mass transported from the $t_1$ sample $i$ to any $t_2$ sample $j$.

These definitions enabled us to formalize the aim of moscot.space.mapping: we sought to find a coupling matrix $P \in U(a, b)$ that relates $t_1$ and $t_2$ samples such that their distance in the shared transcriptome space is minimized. At the same time, the correspondence between spatial arrangements is maximized.

**Model description.** We used identical definitions to the moscot.space.alignment model, where $t_1$ samples play the part of the left dataset and $t_2$ samples play the part of the right dataset. We adjusted the marginals to accommodate cellular growth and death rates as in the moscot.time model, and we optionally allowed for unbalancedness to handle noisy estimates.

**Multimodal data and scalability.** We used the same methods as in moscot.space.alignment to include additional multimodal readout at $t_1$ and $t_2$, and we used the same strategy to scale our model to atlas-scale datasets (Supplementary Note 2).

**Downstream applications.** We extended our model to more than two time points using the same method as in moscot.time, and we supported all downstream analysis functions introduced for moscot.time. We extended the computation of ancestor and descendant probabilities to spatial regions. That is, the cell state $P$ of interest in equation (10) may now represent a spatial region. Thus, moscot.space.mapping enables spatial regionalization to be studied throughout cell-state changes.

We interfaced with squidpy[81] for further downstream analyses of spatiotemporal variation. For example, squidpy can be used to study the spatial enrichment of a mapped cell state of interest across the temporal axis.

### Datasets

**Temporal analysis.** Unless stated otherwise, computations were done using SCANPY[19] with default parameters. To obtain driver features for a subset of cells, for example, for a certain cell type, we correlated (Pearson's or Spearman's) the density of the pull-back distribution of the considered cell type with the corresponding feature, for example, (processed) gene expression. To compute target genes of a TF, we correlated the density of the push-forward distribution of the expression of the TF with all genes and identified highly correlated genes as target genes. The code is available at GitHub (https://github.com/theislab/moscot-framework_reproducibility).

**Moscot.time on a mouse embryogenesis atlas.** The mouse embryogenesis atlas is a collection of data from different sources[7,91–94]. These datasets were preprocessed and annotated[7], and we downloaded them as Seurat objects from http://tome.gs.washington.edu/.

The authors of the study showed how their embedding computation successfully handled batch effects; therefore, we followed their pipeline and reproduced these representations by selecting genes using the FindVariableFeatures of Seurat (v.3) and batch-correcting the data using FindIntegrationAnchors[95]. For further analysis using moscot.time in Python, the Seurat objects were transformed into AnnData[96] objects using SeuratData[97]. For the displayed UMAP[98] of the E8.0–E8.25 pair of time points, we used the 30-dimensional Seurat PCA latent space and a $k$NN graph with $k = 15$.

**Comparison of the memory and runtime benchmark between moscot.time and WOT.** To investigate method scalability, we ran a memory and runtime benchmark. For this, we subsampled from the E11.5–E12.5 time point pair, which had the largest number of cells out of all time point pairs: 455,124 cells at E11.5 and 292,726 cells at E12.5. We generated 11 subsets of increasing size, each containing the same number of cells at E11.5 and E12.5, with a step size of 25,000 cells, up to a maximum of 275,00 cells in either time point.

We compared the performance of three different approaches: WOT, moscot.time and low-rank moscot.time. For moscot.time, we evaluated the cost function on the fly (online evaluation) to achieve linear memory complexity. For low-rank, we chose rank 2,000 because it showed the best accuracy scores in the low-rank comparison (see below). For the memory benchmark, we ran all algorithms on CPUs, as GPU memory benchmarking is difficult, and memory consumption is likely to be very similar on CPUs. For the runtime benchmark, we ran the moscot.time variants on GPUs, but had to run WOT on CPUs as it cannot make use of GPU acceleration. For entropic regularization of both WOT

and moscot.time, we chose $\epsilon = 0.005$. For low-rank moscot.time, we chose $\epsilon = 0.0001$. For the left and right unbalancedness parameters $\tau_a$ and $\tau_b$, respectively, we choose $\tau_a = 0.9$ for full rank and $\tau_a = 0.09$ for low rank, and $\tau_b = 0.99995$ for both low rank and full rank. For these unbalancedness parameters, the apoptosis rates fell within the predefined window of 2–4%.

**Accuracy benchmark between moscot.time and TOME.** We compared the accuracy of the cell transitions inferred using moscot.time and TOME[7]. TOME is a $k$NN-based algorithm that was developed specifically for this dataset. For each $t_2$ cell, TOME finds the $k = 5$ nearest neighbours at $t_1$ and treats these as putative ancestors. By aggregating over cell states at both time points, TOME computes weighted ancestor and descendant relationships on the cell-state level. To improve robustness, TOME median-aggregates the inferred edges over 500 randomly subsampled cell sets, each containing 80% of all cells.

Of note, TOME computes neighbourhood relationships in a 3D UMAP space despite the known pitfalls of low-dimensional nonlinear representations[99–101]. In particular, low-dimensional embeddings such as UMAP or t-SNE[102] do not preserve global data topology well[103,104]; trajectories inferred in such spaces are prone to suffering from projection artefacts. Moreover, TOME lacks the concept of probability mass conservation, whereby a considerable number of cells at $t_1$ can remain without descendants. By contrast, moscot.time computes cell–cell distances in a higher dimensional latent space (30-dimensional PCA in this application), a crucial feature to faithfully describe the data topology of complex developmental state changes. Moreover, moscot.time is a probabilistic approach equipped with a notion of mass conservation grounded on OT.

We applied both moscot.time and TOME to all time-point pairs. Again we chose $\tau_b$ fixed at 0.9995 and tuned $\tau_a$ such that the resulting apoptosis rates are biologically plausible (see the section 'Growth-rate comparison'). For moscot.time, we aggregated the cell-level couplings to cell-state transition rates using the pull-back operation of the corresponding cell state as described above. These cell-state transition rates correspond to the weighted cell-state transition edges obtained using TOME, which enabled direct comparison of both approaches.

**Metrics for the accuracy benchmark for germ-layer and cell-type scores.** We developed two metrics to evaluate the accuracy of obtained cell-state transitions: one germ-layer metric and one cell-type metric. First, our germ-layer metric aggregated cell states into germ layers and considered transitions within and across germ layers as correct and incorrect, respectively. This metric was motivated by the observation that cells typically do not cross germ layers[105]. A prominent exception to this rule is the neural crest, for which a transition from neuroectoderm into osteoblast progenitors (mesoderm) was allowed[106]. We followed the original publication[7] in classifying cell types into neuroectoderm, surface ectoderm, endoderm and mesoderm. As in the original study, we excluded transitions between cell types that could not be unambiguously assigned to a germ layer and transitions with edge weights below 0.05.

Second, our cell-type metric compared every predicted transition with a curated set of allowed transitions. To curate the set of allowed transitions during mouse embryogenesis, we conducted an extensive literature search for all 89 cell types present in the data to identify previously reported ancestor and descendant states (Supplementary Table 2).

We computed accuracy scores for the germ-layer and cell-type metrics by dividing the weighted sums over all transitions that satisfied germ-layer boundaries and cell-type restrictions by the weighted sum over all transitions included in the evaluation, whereby weights are given by the edge weight. We mean-aggregated accuracy scores for pre-gastrulation (E3.5–E6.5), gastrulation (E6.5–E8.5) and organogenesis (E8.5–E13.5), for which we combined the accuracy scores of

different time pairs by weighting by the amount of cell types for the starting time point.

**Embedding robustness comparison.** We compared the performance of moscot.time and TOME on two additional embeddings. First, we computed a PCA embedding on highly variable genes, identified using scanpy.pp.highly_variable_genes with default parameters. On these highly variable genes, we then ran scanpy.tl.pca to obtain the 30-dimensional PCA embedding used in moscot.time. TOME requires a 3D UMAP embedding. Hence, we used scanpy.tl.umap with default parameters and supplied a neighbourhood graph computed in the PCA embedding as input.

Second, we computed a scVI embedding. We split the data into the three stages that corresponded to pre-gastrulation, gastrulation and organogenesis. For each of these stages, we combined all time points into a single AnnData object on which we ran scVI. We used scanpy to compute highly variable genes with flavor=seurat_v3 and subsetted to 2,000 highly variable genes for pre-gastrulation, 3,000 for gastrulation and 5,000 for organogensis. For each stage, we trained a model with default parameters apart from using n_layers=2, n_latent=30, gene_likelihood='nb' in scvi.model.SCVI. Notably, we did not perform any type of batch correction. We used the resulting 30-dimensional embedding directly in moscot.time, whereas for TOME, we ran scanpy. tl.umap to obtain the required 3D UMAP embedding.

We then ran moscot.time and TOME as outlined above, using germ-layer and cell-type scores as evaluation criteria. To obtain mappings with realistic apoptosis rates, for moscot.time, we tuned the left unbalancedness parameter $\tau_a$ for each time point pair and latent representation to fall within the previously described windows (see also the section 'Growth-rate comparison').

**Consistency checks with WOT.** We compared moscot.time with WOT on the smaller time point pairs corresponding to pre-gastrulation and gastrulation stages, for which we could run WOT without memory issues. We ran both methods using the same entropic regularization and unbalancedness parameters and provided the same initial growth rates and median-normalized both cost matrices (default in WOT). We ran both methods until convergence and compared the resulting transport maps.

**Comparison of full-rank to low-rank moscot.time for different choices of ranks.** We compared full-rank to low-rank moscot.time, considering ranks 10, 100, 1,000 and 2,000. We ran the full-rank version with the exact same parameters as in the TOME comparison above. For the same entropic regularization parameter $\epsilon$, low-rank transport maps had higher levels of entropy compared with full-rank maps. We used a smaller $\epsilon$ of 0.0001 for low-rank approaches to counteract this effect. To obtain a good choice for the low-rank gradient step size $\gamma$, we performed a grid search and found that $\gamma = 500$ was a suitable value. Full-rank moscot.time was run until convergence, whereas low-rank moscot.time was run with a fixed number of 1,000 iterations. We kept the right unbalancedness parameter $\tau_b$ fixed at 0.99995 and tuned the left unbalancedness parameter $\tau_a$ such that the apoptosis rates fell within the predefined ranges for the different stages of embryonic development[107,108] (Supplementary Table 3).

**Growth-rate comparison.** Beyond comparisons on the germ-layer and cell-type levels, we wanted to evaluate how moscot.time and TOME compared on the single-cell level. However, the TOME method does not output single-cell transitions; it only reports aggregated cell-type transitions. Thus, to still have a baseline, we implemented a variant of the TOME approach, which we call clTOME, whereby we collected the neighbours that TOME identifies and aggregated them into a single-cell transport matrix. Again, following the original approach, we increased robustness by repeating the process over 500 randomly subsampled

datasets, each containing 80% of the original cells. As subsampling also affected the cells of the later time point, we normalized the data such that columns in the early-to-late cell transition matrix summed to one. In other words, each $t_2$ cell received the same unit mass of incoming transition probability. We did this to make the TOME and moscot.time transport matrices comparable, as the column-sum of the moscot.time transport matrix is close to uniformity because of the high unbalancedness parameter $\tau_b$ of 0.99995. This interpretation of the kNN approach enabled us to define cell–cell coupling matrices in clTOME. Analogous to moscot.time, we used pull and push operations (see section 'moscot.time for mapping cells across time') to compute ancestors and descendants.

We computed cell-level couplings across time points using moscot.time and clTOME, excluding extraembryonic tissues to avoid introducing additional variance from the experimental protocol. For moscot.time, we did not initialize the growth rates using marker genes to enable a fair comparison with clTOME, which does not support such initialization. Instead, we ran moscot.time with uniform marginals and used unbalancedness to learn growth rates de novo. As before, we set $\tau_b = 0.99995$ and chose $\tau_a$ such that the resulting predicted fraction of apoptotic cells lies within a biologically reasonable range[107,108] (Supplementary Table 3). For both methods, we calculated growth rates through the left marginal (row sum) of the corresponding coupling matrix, $\sum_j P_{ij}$. To avoid overcrowding our histograms of growth rates per cell type, we only showed the five cell types with most cells per time point.

An important aspect of interpreting trained OT growth rates (marginals) with biological growth rates is adjusting for the number of cells in the embryo. Specifically, we computed the change in population size between two time points, $s = |e_1|/|e_2|$ where $|e_1|$ and $|e_2|$ represent the estimated cell number at early and late embryo stages, respectively. Next, we scaled the mean growth rate of $t_1$ cells by multiplying with the average number of ancestors $s$ to obtain biologically interoperable growth rates. To obtain an estimate of the apoptosis rate, we calculated, for each cell, the difference between 1 and the scaled growth rate $g_i$. If for a particular cell $g_i$ is smaller than 1, on average $(1 - g_i)$ of this cell dies.

By summing over these differences for all $t_1$ cells for which the scaled growth rate was smaller than 1, we calculated the predicted number of dying cells at $t_1$. We divided the sum by the total number of $t_1$ cells in the dataset to obtain estimated apoptosis rates. We ran the above calculations independently for all time points for moscot.time and clTOME. We chose the target apoptotic range by combining information from various publications[18,107–111]. We chose the target apoptotic range for pre-gastrulation to be 10–15% apoptotic cells, 4–6% for gastrulation and 2–4% for organogenesis. For the time pair E8.5a–E8.5b, for which not real time passes but there is a transition in experimental methods, we aimed for a relatively high apoptosis rate of 10–40% to allow for the correction of sampling biases.

**Correlating predicted growth rates with gene-set-based growth rates.** To validate predicted growth rates, we correlated them with cell-cycle scores computed on the basis of marker gene expression using scanpy through scanpy.tl.score_genes. In brief, the scanpy implementation of gene scoring follows the original suggestion in Seurat (v.1)[112]: it averages over genes in the supplied gene set, normalized by the average expression of a reference set of genes. For this comparison, we initialized marginals uniformly so that our algorithm was not aware of growth rates and we could use this information for validation.

We applied this strategy to a different dataset that comprised reprogramming mouse embryonic fibroblasts[2]. This dataset was better suited for the growth-rate comparison for two reasons. First, the gene set we used to score the cell cycle based on scRNA-seq data was tailored to mouse fibroblasts and haematopoietic stem cells[113]. Applying this gene set to the mouse embryogenesis atlas gave haematopoietic lineages

consistently higher scores than other cell types, which contradicted previous biological findings[114]. Second, the mouse embryogenesis atlas represents an in vivo setting, for which each time point corresponds to a different individual, thereby leading to strong variations in cell-type proportions across time points, which are not driven by cellular growth and death but by cellular sampling effects. In particular, extraembryonic tissues were subject to large systematic sampling biases, most likely due to variations in sample handling.

By contrast, the mouse embryonic fibroblast reprogramming dataset was better suited given our gene set, and, as an in vitro setting, contained fewer biases in cell-type frequencies driven by cell sampling. We ran moscot.time using an entropic regularization of 0.0005 and unbalancedness parameters $\tau_a = 0.98$ and $\tau_b = 0.99995$. We then computed clTOME growth rates on the same dataset. We computed the cell-cycle scores using scanpy.tl.score_genes through the implementation of moscot with the gene set determined for mouse fibroblast and haematopoietic stem cells.

**Comparison in terms of driver-gene correlations.** To further assess the cell-level couplings predicted by moscot.time and TOME, we reasoned that high correlations between ancestor probabilities and known driver genes for a cell state are indicative of the success of the method. Thus, for the cell states described in the main text, we computed the ancestor distributions predicted using moscot.time and clTOME (see the section 'moscot.time for mapping cells across time'). To exclude the influence of driver genes involved in unrelated differentiation events, we restricted the correlation computation to known progenitor populations. For each pulled cell state, we curated a list of known driver genes (Supplementary Table 4), filtered the list to contain only highly variable genes at the corresponding time point and imputed their expression using the decoder output of scVI with get_normalized_expression. After filtering to highly variable genes, we retained 36 genes for definitive endoderm, 18 genes for allantois, 39 genes for the first heart field and 106 genes for the pancreatic epithelium. We calculated Spearman's correlation values between these imputed expression values and predicted ancestor distributions using scipy.stats.spearmanr[115].

**Metacell analysis.** Another way to accelerate the computation of mappings is to aggregate cells into metacells. To investigate the performance of this possibility, we computed metacells using the popular Metacell-2 algorithm[21] on E9.5 cells. However, we found that for the rare cell state primordial germ cells (30 cells or 0.03% of the population), no metacell was created, which made the inference of progenitors and ancestors of this cell state impossible.

We also computed metacells on the E10.5–E11.5 pair of time points and used moscot for temporal mapping, both on the single-cell and metacell levels. For this, we adjusted the marginals, making them proportional to the number of cells in the corresponding metacell. We chose $\epsilon = 0.005$, $\tau_b = 0.99995$ and $\tau_a = 0.8$ such that the resulting apoptosis rate was in the range of 2–4%. To enable a fair comparison with moscot ran on single cells, we mapped the metacell coupling matrix back to the level of single cells. We then evaluated mapping accuracy using the curated cell-type and germ-layer transition scores as well as the correlation of E11.5 pancreatic epithelium ancestor probabilities with known driver genes for pancreas development. For the curated cell-type and germ-layer transitions, we set the threshold of transitions to consider to zero because the transition matrix from the metacell analysis contained many small entries.

### Moscot.time on multimodal pancreas development

**Dataset generation.** Embryonic pancreata from NVF homozygous mice were collected and pooled together (8 pancreata from E14.5 and 11 pancreata from E15.5 for the first experiment (exp-1), and 10 pancreata from E15.5 and 10 pancreata from E16.5 for the second experiment (exp-2)). Trypsin (0.25%) was added to the samples for 5 min on ice and then

incubated at 37 °C for 10 min. The single-cell samples were centrifuged at 1700 r.p.m. (290$g$) for 5 min at 4 °C. After removing the supernatant, cells were counted. Next, 5 µl rat IgG2a K isotype control (eBioscience, 12-4321-42) and anti-mouse CD326 (EpCAM) PE (eBioscience, 12-5791-81) was used for $1 \times 10^6$ cells (100 µl total volume). Samples were stained for 30 min at 4 °C following staining with DAPI to detect dead cells. After washing twice and resuspending in FACS buffer (PBS, 1% BSA and 0.5 mM EDTA), the single-cell samples were loaded for FACS analysis. The following gating strategy was used: main population > single-cells > living cells (DAPI⁻) > EpCAM⁺ (PE⁺) and NGN3⁺ (FITC⁺)/NGN3⁻ (FITC⁻) cells. Sorted cells were pooled in a 2:1 (EpCAM⁺NGN3⁺: EpCAM⁺NGN3⁻) ratio and immediately used for isolation of nuclei.

To isolate nuclei, a low-input nucleus isolation protocol adapted from 10x Genomics was performed. In brief, sorted cells were washed once with 1 ml PBS + 1% BSA, counted on the basis of Trypan blue staining and centrifuged. Subsequently, the washed cell pellet was resuspended in chilled lysis buffer (50 µl per sample) and placed on ice for 4 min. Then, wash buffer (500 µl per sample) was added and nuclei were centrifuged. To gradually change from wash to diluted nucleus buffer, cells were washed once in a 1:1 mixture of wash buffer and diluted nucleus buffer and subsequently once with pure diluted nucleus buffer. The washed isolated nuclei were then resuspended in 7–10 µl diluted nucleus buffer and were, after quality control and counting, immediately used for single-cell multiome library preparation with a target recovery of 10,000 cells.

Libraries were prepared using the Chromium Next GEM Single Cell Multiome ATAC + Gene Expression Reagent Bundle (10x Genomics, 1000283) according to the manufacturer's instructions. Libraries were sequenced on an Illumina NovaSeq6000 platform following the recommendations from 10x Genomics. Raw reads from both experiments were jointly aligned to the GRCm38 mouse genome with Ensembl release 102 annotations and pre-processed using the 10x Genomics CellRangerARC pipeline (v.2.0.2) for downstream analyses.

**Preprocessing.** We preprocessed the samples independently for gene expression and chromatin accessibility. Peaks were taken from the CellRanger output independently for each sample and subsequently merged.

With respect to gene expression, all cells with a mitochondrial gene fraction higher than 0.025 in E14.5 or higher than 0.02 in E15.5 were removed in exp-1. For exp-2, all cells in E15.5 with a mitochondrial gene fraction higher than 0.015 were removed, and the threshold for E16.5 was set to 0.02. Moreover, for exp-1, cells with fewer than 4,000 counts or more than 30,000 counts were removed in E14.5, and cells with fewer than 5,000 counts or more than 40,000 counts were removed in E15.5. For exp-2, the lower thresholds for both time points were chosen to be 3,000, whereas cells with a total gene count of at least 60,000 and 70,000 were removed for time point E15.5 and E16.5, respectively. All cells with fewer than 2,300 genes expressed were filtered out for E14.5 (exp-1). The analogous thresholds for E15.5 (exp-1), E15.5 (exp-2) and E16.5 (exp-2) were set to 2,700, 2,000 and 2,000, respectively.

Concerning ATAC modality, all cells in E14.5 (exp-1) with nucleome signals lower than 0.35 or higher than 1.75 were removed. All cells with transcription start site enrichment scores lower than 2.5 or higher than 7.5 were filtered out. Cells were also removed if their total open-chromatin region count was below 4,000 or above 150,000. Analogously, the minimum nucleosome signal was set to 0.3, 0.35 and 0.25 for E15.5 (exp-1), E15.5 (exp-2) and E16.5 (exp-2), respectively, whereas the upper threshold of the nucleosome signal was chosen to be 1.75, 1.5 and 1.4, respectively. Moreover, the lower threshold of the transcription start site enrichment score was set to 2.75, 2.5 and 2.5 for E15.5 (exp-1), E15.5 (exp-2) and E16.5 (exp-2), respectively, whereas the upper one was set to 10.5, 8 and 7.5, respectively. The lower total peak counts threshold was set to 4,000 for E15.5 (exp-1), E15.5 (exp-2) and E16.5 (exp-2), whereas the upper one was set to 100,000, 160,000 and 170,000, respectively.

After concatenation of the two samples, genes that were detected in fewer than 20 cells were filtered, which resulted in 20,244 genes.

Doublets were identified using a mean prediction of multiple doublet-detection methods. We used Scrublet[116], scDblFinder[117], DoubletDetection[118], scds[119], SOLO[120] and DoubletFinder[121]. To identify doublets based on ATAC counts, we used AMULET[122]. Whenever at least three single methods out of the seven methods rated a cell to be a doublet, we considered the cell as doublet. In total, 12.60% doublets were identified in sample E14.5 (exp-1), 10.73% doublets in E15.5 (exp-1), 16.68% doublets in E15.5 (exp-2) and 15.11% doublets in E16.5 (exp-2).

Subsequently, clustering in MultiVI[62] embedding was repeatedly performed, and clusters with a large majority of identified doublets were removed.

**Cell-type annotation.** To construct a weighted nearest neighbour graph, an embedding of both modalities is needed. Therefore, before performing a PCA (50 dimensions) on the $\log_1 P$-tranformed gene expression data, the count data were normalized using SCTransform[123] and cell-cycle genes and ambient genes were discarded. Ambient genes were identified using DropletUtils[124]. The ATAC data were processed by term frequency-inverse document frequency (tf-idf) normalization followed by singular-value decomposition using Signac, computing the first 50 singular components. Owing to a high correlation with the sequencing depth, the first and the fifth components were removed. Having computed respective embeddings for GEX and ATAC, we constructed a weighted nearest neighbour graph using MUON[125] and used it for multimodal, unsupervised clustering. Unless stated otherwise, this is also the graph on which we computed UMAPs.

Annotation was performed on the basis of the expression of marker genes as reported in previous studies[34,42,113,126–128] (Supplementary Table 9) and cell-cycle scores for the proliferating populations computed using scanpy.tl.score_genes_cell_cycle. It is important to mention that we identified a cluster branching off the $Ngn3$[High] population, which we found to express similar genes as a cluster called $Fev$⁺ epsilon as previously described[34]. In fact, neither the cluster reported in that study[34] nor the cluster found in our new dataset has a substantially high expression of $Fev$ (Supplementary Fig. 28). Hence, we labelled this cluster as epsilon progenitors.

To arrive at the finer resolution of cell types as shown in Fig. 3e, subclustering was performed on the same neighbourhood graph (incorporating both modalities).

**The moscot.time model.** We computed the cost matrix defining the OT problem between E14.5 and E15.5 using the 30-nearest neighbours graph (computed with scanpy.pp.nearest_neighbors) on the MultiVI embedding to compute distances based on heat kernel diffusion (Supplementary Note 5). Hence, this graph was constructed on a different embedding than the one we used for unsupervised clustering to reduce the bias to one embedding.

Two moscot models were run on the basis of the weighted nearest neighbour graph for which construction is described above. First, a model was run on the full dataset. The moscot.time model was run with default parameters, but a bit of unbalancedness was introduced by setting $\tau_a = \tau_b = 0.99$. In detail, the regularization parameter ε was set to $10^{-3}$ and the cost matrix was scaled by its mean. To guarantee convergence, the number of iterations was increased to $10^7$. Uniform marginals were chosen because the large abundance of highly proliferating ductal cells would have marginalized the influence of less-abundant cell types. It is also important to note that the dataset is FACS-sorted; therefore, proportions of initially sequenced cells are highly biased and do not reflect the true cell-type distribution. This also causes the final model to not predict descendants (ancestors) across one day (wall clock time). In effect, the directionality of the developmental process is kept, whereas its magnitude does not reflect ground-truth biological progress.

For the analysis of the endocrine branch, the OT solution was computed on a reduced dataset that only contained endocrine cells (alpha, beta, delta and epsilon) and their progenitors (cells labelled as *Fev*+ alpha, *Fev*+ beta, *Fev*+ delta, epsilon progenitors, *Fev*+, *Ngn3*High, *Ngn3*High cycling or *Ngn3*Low). Again, uniform marginals were chosen because the proliferation and apoptosis scores obtained from TemporalProblem.score_genes_for_marginals were almost constant. The OT solution was computed using standard parameters provided by moscot.

**Studying the influence of the embedding and cost function.** To demonstrate the stability of the OT predictions, we ran moscot with different hyperparameters, including the choice of the latent embedding (incorporating only one modality, that is, PCA space of gene expression, VAE embedding of gene expression (scVI space[129]), LSI space of ATAC, VAE embedding of peaks (PoissonATAC space), as well as incorporating both modalities by concatenation of the spaces (PCA–LSI, PCA–PoissonVI[63], scVI–LSI and scVI–PoissonVI) as well as the MultiVI[62]) and the choice of cost (squared Euclidean, cosine cost, geodesics from heat diffusion with different number of neighbours $k \in [5, 10, 30, 50, 100]$). This resulted in 63 different configurations of hyperparameters of the moscot model. We measured the stability using two metrics. We globally analysed the stability of the transport matrix by computing the Sinkhorn divergence between the transport plan aggregated to cell-type level and the aggregated transport plan of the moscot predictions we used to analyse the pancreas dataset throughout the rest of the article. Second, we assessed the stability with respect to the research question of the origin of delta and epsilon cells by measuring the differences in the probability of certain cell-type transitions. We also report the transitions from *Fev*+ beta cells to beta cells, which is biologically confirmed. We compared the mean (and s.d.) across all 63 configurations with the predictions of the outer coupling. We observed that the mean Sinkhorn divergence was much lower than the Sinkhorn divergence between the reference coupling and the outer coupling, which meant that all couplings are close to the reference coupling, which has been used throughout the article. Similarly, we observed much higher cell-type to cell-type transitions than those obtained with the outer coupling.

We aggregated the computed transport matrix to the cell-type level (A) and consecutively column-normalized to obtain probabilities of ancestry. We then computed the Sinkhorn divergence between the aggregated transport matrix A and the aggregated transport matrix of the reference B, which is the moscot coupling we used for the analysis of the pancreas dataset. The cost of the Sinkhorn divergence was chosen as a binary distance, that is, only with entries 0 and 1. We also considered the following transitions: delta from *Fev*+ delta; epsilon from epsilon progenitors; epsilon from *Fev*+ delta; *Fev*+ delta from epsilon progenitors; and beta from *Fev*+ beta (as a biologically known transition). We compared the values of these transitions with the transitions we obtained with the outer coupling.

We also report the probabilities of delta cells as being directly derived from *Ngn3*Low, a transition that is biologically implausible. Here we saw that the choice of the cost function mattered a lot, but was independent of the embedding. That is, when using the squared Euclidean cost or the cosine cost, we observed a significant proportion of this transition, whereas with the geodesic cost, we did not observe this transition.

**Driver feature analysis with moscot.time.** We computed driver features by correlating the density of the pull-back distribution with feature values. Moreover, when analysing the transition from epsilon to alpha cells, we excluded the *Fev*+ alpha population (the main progenitor cell type of alpha cells) from the set of considered cell types. That is, we set their ancestry probability to zero. This helped identify genes that are particularly activated in the epsilon cells when correlating the pull-back distribution with the processed gene expression.

**Marker regions of chromatin accessibility.** To identify marker regions of chromatin accessibility, a Wilcoxon test was run by calling FindMarkers provided in Seurat. The test was performed with default settings and the considered cell type was run with respect to all remaining cell types (subset to endocrine cells and endocrine progenitors; that is, *Ngn3*Low, *Ngn3*High, epsilon progenitors, *Fev*+, *Fev*+ delta, *Fev*+ alpha, *Fev*+ beta, alpha, beta, delta and epsilon).

**Motif analysis.** Motif data were downloaded from cisBP[130]. Position weight matrices and corresponding visualizations and metadata were downloaded as a bulk download after filtering by species (*Mus musculus*) on 1 March 2023. cisBP contains data from both experimentally measured binding activities and inferred ones (for example, from other species). TFs with DNA-binding domain amino-acid similarities above a certain threshold (defined for each DNA-binding domain class separately and provided by cisBP) were also considered as binding candidates.

We defined a TF to have an association with a motif if it was either directly measured or inferred and had a sufficiently high DNA-binding domain amino-acid similarity. That is, is reported as such by cisBP. This way, one motif can have an association with multiple TFs and one TF can have an association with multiple motifs.

To obtain motif scores on a single-cell level, chromVAR was run using the API provided in Signac. In effect, AddMotifs was called, followed by RunChromVAR. To obtain marker motifs with moscot, we considered the temporal order of gene expression and activity of a motif. Of note, moscot comes with a list of TFs for different species (human, mouse and *Drosophila*) obtained from the SCENIC+ database[131]. Thus, we computed driver TFs using the capability of moscot to compute driver features. Moreover, we performed a differential motif-activity test (Wilcoxon test using scanpy's rank_genes_groups) based on the ChromVAR scores. Subsequently, we identified our marker motifs by combining these two sources of information. Therefore, we only kept marker TFs for which we had an associated TF.

**Cell-cycle analysis.** We used the pancreatic endocrinogenesis dataset. As we studied the cell cycle in proliferative ductal cells, we removed immature and mature acinar cells. Consecutively, we computed OT couplings between E14.5 and E15.5, as well as between E15.5 and E16.5 on the shared MultiVI embedding (as for the main analysis of this dataset). Consecutively, we assigned cell-cycle phases as proposed[132,133] to all proliferative ductal cells. By aggregating the transport matrix to cell-cycle-phase annotations and consecutive row normalization, we obtained transition probabilities for cell-cycle phase $i$ to cell-cycle phase $j$ (with ordered cell-cycle phases being defined as G1S, S, G2M, M and MG1). Note that we did not take transitions into account, which include non-cycling cells. In effect, the aggregated matrix A represents the probability that a cell of the early time in one cell-cycle phase transitions to a certain cell-cycle phase in the later time point. Correct (or wrong) directionality means that cells in cell-cycle phase $i$ are more (or less) likely to transition to cell-cycle phase $i + 1$ rather than to cell-cycle phase $i - 1$. Here we assumed that $i + 1 = 1$ if $i = 5$ (hence the transition from MG1 (index 5) to G1S (index 1)) and $i - 1 = 5$ if $i = 1$ (hence the transition from G1S (index 1) to MG1 (index 5)) to close the circle. In other words, we considered a transition $i$ to $i - 1$ as bad (wrong direction) and $i$ to $i + 1$ as good (right direction). Thus, we were able to build a score from the 5-by-5 aggregated transport matrix A by adding the scores $-A_{i, i-1}$ (note the minus) and $+A_{i, i+1}$ (note the plus). If the score was positive, this meant that the direction is correct (as more cells go from stage $i$ to $i + 1$ than from stage $i$ to stage $i - 1$). We computed the score for both transitions E14.5–E15.5 and E15.5–E16.5, and performed a permutation test (by permuting all non-diagonal entries of A) with 10,000 permutations.

**Trajectory inference with different trajectory-inference methods.** We used diffusion pseudo-time[134], scVelo[39], veloVI[135], MultiVelo[136],

CytoTrace[137] and the ConnectivityKernel[16] in CellRank to predict trajectories in the pancreatic endocrinogenesis dataset. As we were interested in the endocrine-cell trajectories, we filtered the dataset to endocrine cells and their progenitors. We then applied the GPCCA estimator in Cell-Rank to each corresponding trajectory-inference kernel. To compute fate probabilities, we used compute_fate_probabilities and aggregate_fate_probabilities to plot the fate probabilities and aggregated cell-type to cell-type transition matrices. We used the plot_projection method to generate the stream embedding plots. We used all default arguments provided for the methods, and only increased max_epochs in VeloVI to 50. When using graphs, we used the WNN graph as described above. For building the RealTimeKernel with moscot, we set the weight of the ConnectivityKernel to 0.001 to strengthen the influence of moscot and weaken the influence of the ConnectivityKernel. We highlight that the transition probabilities computed with CellRank rely on a different procedure than the transition probabilities we computed with moscot.

**Labelling of the quality of trajectory-inference methods.** To assess the reliability of the trajectory-inference methods, we assessed their performance by focusing on the well-studied fate probabilities of the alpha and beta cell lineage. In effect, we consider the fate probabilities of alpha, beta, *Fev*+ alpha and *Fev*+ beta cells, with alpha, beta, delta and epsilon being possible lineages. We considered a transition to be correct if the highest fate probability is the biologically most likely one (that is, alpha→alpha, *Fev*+ alpha→alpha, beta→beta, *Fev*+ beta→beta). Finally, we assigned a method a green dot if all of the four transitions are correct, an orange dot if three out of four transitions are correct, and a red dot for other outcomes.

**Assessment of the consistency with the fate probabilities of moscot.** To assess the consistency of the predictions between the mode and moscot, we computed the Pearson's correlation coefficient between aggregated fate probabilities as output using CellRank.

**Diffusion pseudotime based on different modalities.** For studying the influence of the modality for the prediction of diffusion pseudotime, we constructed a graph (using scanpy.pp.neighbors with default parameters) based on the PCA for the gene expression and LSI for the ATAC modality. For diffusion pseudotime computation, we used the weighted nearest neighbour implementation in muon with default parameters.

**Interpretability of the transport map with Sparse Monge.** We applied sparse Monge (described above) to endocrine progenitors in time points E15.5 and E16.5. We selected the most highly variable genes using scanpy.pp.highly_variable_genes with the default configuration, that is, flavor='seurat'. This resulted in 2,551 highly variable genes. We then transported the cells from E15.5 to cells in E16.5 in normalized and $\log_1 P$-transformed space, using the elastic L1 cost with regularizor $\gamma = 10$ (equation (13)). As previously described[47], we then identified relevant genes per single cell by determining whether the displacement of a cell is higher than $10^{-6}$. Subsequently, we referred to this cell-specific set of genes as important genes. We identified the most relevant genes per cell type by computing the fraction of cells of this cell type to have the gene as an important gene. For recovering variability in fate decisions, we computed a nearest neighbour graph ($k = 50$) and computed the Jaccard similarity between the important genes of the cell considered and the important genes of the neighbourhood. We computed one minus this score to obtain a notion of dissimilarity.

**Gene KO of *NEUROD2* and iPS cell differentiation to pancreatic endocrine cells.** All statistical analyses were performed using one-way analysis of variance with GraphPad Prism 10.

**In vitro differentiation of iPS cells to pancreatic endocrine cells.** Two homozygous *NEUROD2* KO, nuclear H2B-Venus reporter (NEUROD2nVenus/nVenus) iPS cell clones and the heterozygous hiPSC-INS-T2A-H2B-Cherry reporter (INSCherry/WT) iPS cell line (as the control) were used. We used the multistep differentiation protocol for in vitro differentiation of iPS cells into endocrine and islet cells that includes the stage (S) definitive endoderm (S1), primitive gut tube (S2), pancreatic progenitor 1 (PP1) (S3), PP2 (S4) and endocrine lineage (S5, 6) as previously described[138,139]. The C-peptide–mCherry reporter human iPS cell line (HMGUi001-A-8) and NEUROD2nVenus/nVenus iPS cell clones (HMGUi001-A-42) were used. All cell lines were routinely tested to ensure that they were negative for mycoplasma. The sample size was determined on the basis of the available experimental data. Antibodies used were goat anti-somatostatin, 1:300, polyclonal (D-20) (Santa Cruz Biotechnology, SC-7819) and mouse anti-ghrelin, 1:250, monoclonal (2F4) (Santa Cruz Biotechnology, SC-293422).

**RNA extraction and qPCR analysis.** Total RNA was isolated from samples using a miRNeasy mini kit (Qiagen). Reverse transcription was then performed using a SuperScript Vilo cDNA synthesis kit (Thermo Fisher Scientific) according to the manufacturer's instructions. Predesigned TaqMan probes (Life Technologies) were used for qPCR analysis (sequences listed in Supplementary Table 1). The reaction mix for each sample contained 20 ng cDNA, 4.5 µl nuclease-free water, 5 µl TaqMan Advanced master mix (Life Technologies) and 0.5 µl TaqMan probe (Life Technologies). The reactions were run on a QuantStudio 7 Flex instrument (Thermo Fisher Scientific). *GAPDH* was used as the reference gene for normalization. To preserve the spread in the data and to facilitate statistical analysis assuming equal variance, $C_t$ values from KO samples were normalized to the average control value[140]. Data from independent samples within a single qPCR run were analysed together (more detail is provided in the source data). The following primers were used: Hs02758991_g1 for *GAPDH*, Hs00356144_m1 for *SST*, Hs01074053_m1 for *GHRL*, Hs02741908_m1 for *INS*, Hs01031536_m1 for *GCG* and Hs00242160_m1 for *HHEX*.

**Immunostaining and imaging.** Cryosection preparation, fixing and immunostaining were performed as previously described[45]. The following primary antibodies were used: somatostatin (D-20) goat polyclonal (Santa Cruz sc-7819) and ghrelin mouse monoclonal (Santa Cruz sc-293422). Pictures were taken using a Leica DMI 6000 microscope with using LAS AF software. Images were analysed and quantified using LAS AF and ImageJ software programs.

**Spatial analysis**

**Benchmarking moscot.space.mapping across a range of spatial datasets.** We benchmarked the mapping problem of moscot against two state-of-the art methods, Tangram[23] and gimVI[24], as implemented in scVI tools[60]. We used previously published datasets[25]. From these datasets, we selected ones that we were able to reprocess, which resulted in 14 that we considered for the benchmark. Furthermore, in contrast to the original benchmark, we did not use the single-cell dataset as reference, as we were not confident that such data represented a faithful ground-truth for comparing the methods. Therefore, we split the spatial dataset such that 50% of the data points were treated as the single-cell reference and 50% were treated as spatial data. We also explicitly maintained the data type as input consistent with model requirements. Therefore, we normalized and $\log_1 P$-transformed counts for both moscot and Tangram and we kept raw unnormalized counts for gimVI. We randomly held out 100 genes if the total number of genes in a dataset was >2,000, otherwise we held out 10 genes. We trained models on the remaining genes and evaluated performance using Spearman's correlation. We report the mean Spearman's correlations across three random seeds (including random seeds both for dataset split and initialization and training routines). For some datasets, Tangram or gimVI could not be run either due to time complexity (we set a maximum budget of 5 GPU h$^{-1}$ for each method to run) or errors of the models

(for example, an inability to match gene identifiers between training and imputed data).

Specifically, we ran the sweep on the following parameters:
- moscot: epsilon entropy regularization parameter, alpha interpolation parameter between W term and GW term, and tau_a unbalancedness term for the spatial dataset (source). For the cost, we tried cosine and squared Euclidean cost for the linear and quadratic term and joint_attr; that is, the representation to use for the linear term. We assessed both PCA and gene expression on a common set of genes present in both spatial and single-cell datasets.
- Tangram: learning rate and number of epochs.
- gimVI: number of epochs and number of latent dimensions.

We also report memory and time complexity for each algorithm across datasets and seeds. All experiments were run on GPUs on the Helmholtz Cluster (mix of V100 and A100 GPUs).

**Spatial correspondence.** Spatial correspondence was computed as follows. First, we computed $n$ increasing spatial distance (Euclidean) thresholds between all data points in the dataset. Then, at each threshold level, we computed the gene-expression similarity (Euclidean distance) between all genes in all the spots for which (Euclidean) distance was below the selected threshold. The spatial correspondence was then calculated as the Pearson's correlation between gene-expression similarity and the spatial-distance thresholds. The computation is implemented as a method of moscot's mapping problem.

**Moscot.space.mapping on the liver.** We applied the mapping problem of moscot.space to the mouse liver dataset from Vizgen MERSCOPE downloaded from https://vizgen.com/data-release-program/. We processed the dataset following standard scanpy and squidpy processing. For the single-cell reference, we downloaded the CITE-seq dataset from https://www.livercellatlas.org/, which was reported in a previous study[26]. We used the mapping problem in the following way: we used the set of 336 common genes for the linear term, whereas for the quadratic term, we used the PCA of gene expression for the single-cell reference dataset and the PCA gene expression concatenated to the spatial coordinates for the spatial dataset. We then performed the gene expression and protein imputation by computing the barycentric projection (equation (16)) of protein expression to the spatial dataset. The same barycentric projection approach was also used to transfer annotations of cell types from the single-cell reference to the spatial dataset.

**Moscot.space.mapping on spatial ATAC–seq data.** To benchmark whether leveraging the multimodal representation improves modality mapping, we considered a previously processed dataset for joint multimodal RNA and ATAC profiling of human tonsils[141]. The dataset consists of a single slide of human tonsil biopsy samples, profiled with a modified version of the DBiT-seq technology, which is able to profile both chromatin accessibility and gene expression for the same capture locations. In total, the dataset consists of 2,500 unique capture locations. We performed feature selection, PCA and dimensionality reduction with UMAP with the standard Scanpy workflow. We randomly split the dataset into two parts and used the first half as the proxy single-cell dataset, which consisted of gene-expression and chromatin-accessibility information, and the second half as the proxy spatial dataset, which consisted of gene-expression and spatial coordinates. We then set out to evaluate whether utilizing the additional modality for the quadratic term in the mapping problem would improve the prediction of chromatin accessibility in the ATAC case. We used the ATAC information for the spatial dataset only for evaluation, which was measured using the Pearson's correlation of peak accessibility as predicted by the barycentric projection of the single-cell dataset. In particular, we evaluated the average correlation of the top ten marker peaks for the seven clusters provided by the authors, which resulted in a total of 70 accessibility peaks. All experiments were run on GPUs

on the Helmholtz Cluster (mix of V100 and A100 GPUs). Benchmarks were run using Hydra.

**Benchmarking of moscot.space.alignment on simulated data.** We benchmarked the alignment problem of moscot against two other state-of-the-art alignment methods: PASTE[4] and GPSA[30]. We chose the same computational budgets across all methods; that is, 12 unique sets of hyperparameters:
- Moscot: epsilon (entropy regularization parameter) and alpha (interpolation parameter between W term and GW term).
- PASTE: alpha (interpolation parameter between W term and GW term) and norm (scaling of the cost matrix).
- GPSA: kernel (kernel for the Gaussian Process), n_epochs (number of epochs) and lr (learning rate).

Owing to the inability to run GPSA on GPUs, we ran all methods on CPUs. We generated four synthetic datasets based on the data generation described in the GPSA publication[30]. In brief, samples from random normal distribution were generated to build a synthetic gene-expression file arranged in a grid. Points were then randomly subsampled by a fraction of 0.7, 0.8 and 0.9 of the original datasets, so that the total number of points did not match in the source and target dataset. This approach was similar to the previously used benchmark settings[4,30]. However, to make all three methods comparable, we used the barycentric projection (equation (16)) of spatial coordinates with respect to the coupling for both PASTE and moscot. Because of the low sample size of the experiments, we ran moscot in full-rank mode (as opposed to low-rank mode). Larger datasets, such as the one analysed in the main text, would be prohibitively large for both PASTE and GPSA.

**Moscot.space.alignment on mouse brain coronal sections.** We applied the alignment problem of moscot to a large-scale MERFISH dataset from Vizgen MERSCOPE (https://vizgen.com/data-release-program/). Specifically, two sections of the mouse coronal brain. We aligned three samples from three different mice for each section. We performed the first alignment with the alignment problem of moscot in the affine mode (equation (17)). Thus, two out of the three slices were aligned to the remaining one, which was chosen as the reference. Furthermore, we performed a second alignment on FGW-aligned coordinates with a W-type problem to obtain an improved warped alignment. This turned out to prove useful in low-rank settings. We performed the same operations for both triplets of coronal sections.

**Gene-consistency analysis of aligned slices.** We assessed the quality of the alignment based on gene expression only, as we did not have cell-type annotations for the brain sections of interest. To this end, we computed the neighbour graph in the aligned space using squidpy ($k$NN mode with at least 30 neighbours for each observation). Then, for each gene, we filtered cells with no expression and retrieved neighbours of the reference section (0) from the two other sections (1 and 2). We then assessed the gene-expression histogram across all cells in the query sections that were neighbours in the reference section and reported the expression of the gene of interest. We performed this analysis across all genes and reported the L1 Wasserstein distance between gene-expression histograms. A low L1 Wasserstein distance between the gene-expression density of the query section and the gene-expression density of the reference section meant that the set of cells in the reference is similar to the matched cells of the aligned section. Conversely, if the L1 Wasserstein distance was high, it meant that neighbouring cells in the query and reference slides are not similar in gene-expression distribution, which therefore highlights a potential mismatch in the alignment. It should be noted that a source of such a mismatch could also be the intrinsic biological variability between tissue sections. Nevertheless, because we did not have access to tissue-section annotations, we decided to use the gene-expression similarity metric described above to quantitatively evaluate alignment consistency. We further evaluated

whether the distribution of L1 Wasserstein distance between query and reference sections showed a correlation with the mean expression of the gene. We did not observe a strong association, which highlighted the fact that this analysis is robust to gene-expression variability. All results are reported in Supplementary Fig. 14.

## Spatiotemporal analysis of mouse embryogenesis Stereo-seq data

**Preprocessing.** We used a previously published mouse embryogenesis Stereo-seq dataset[10]. The data were preprocessed and annotated by the authors of that study and available for download as AnnData objects from https://db.cngb.org/stomics/mosta/. In the reported analysis, for full embryo mapping, we used Mouse_embryo_all_stage.h5ad, a file that contains a single slide for each time point and annotations to major tissue and organs (hereafter referred to as annotations). This file was also used to extract brain bins from early time points. For the latest time point, E16.5, we used the detailed brain annotation slide given in 16.5_E1S3_cell_bin_whole_brain.h5ad. For each section, we used the 'count' layer and performed standard preprocessing with scanpy. We filtered bins (min_genes = 200) and genes (min_cells = 3), normalized cell counts and log-transformed the data.

To perform analysis over brain bins, and to transfer the annotation from E16.5 to earlier time points (E13.5–E15.5), we extracted bins annotated as 'brain' from the full embryo AnnData object and merged them with the E16.5 annotated brain AnnData object.

**Mapping accuracy.** We used moscot.spatiotemporal on each time pair of the data and calculated annotation-transition rates. We compared the accuracy to moscot.time, TOME[7] and PASTE2 (ref. 31) using the germ-layer and annotation-transition accuracy as described above. In both moscot settings, we fixed epsilon ($\epsilon = 1e - 4$), used the unbalanced low-rank approach, used rank = 500, $\gamma = 100$ and performed a grid search for $\tau_a$, $\tau_b \in \{0.01, 0.05, 1.0\}$. We included biologically informed priors using growth-rate and death-rate modelling computed by moscot. For moscot.spatiotemporal, we also performed a grid search for the interpolation parameter $\alpha \in \{0.0, 0.2, 0.4, 0.6, 0.8, 0.9\}$. To assess TOME performance, we followed the code base https://github.com/ChengxiangQiu/tome_code. For PASTE2 evaluation, we used published code https://github.com/raphael-group/paste2. However, owing to scaling limitations, we subsampled the data at each time point to $n = 2,000$ cells. We evaluated performance over subsamples from ten different random seeds.

**Moscot analysis.** Using dedicated analysis functions in moscot, we identified TF driver genes and putative target genes for the liver. For the analysis of driver genes, we first computed the pull-back of liver annotation from each pair of time points. Next, driver genes were obtained using moscot.compute_features_correlation between the aggregated pull-back and gene expression. Similarly, for the identification of target genes, we computed the push-forward of *Hnf4a* across time points and evaluated its correlation with gene expression.

**Mapping annotations across time points.** We used the detailed cell-type annotation provided for E16.5 brain to infer annotations of earlier time points. We mapped bins across time points using a higher rank (rank = 10,000), now possible as we were considering a subpopulation of the bins. To obtain the annotation, we started from the last couple (E15.5 and E16.5) and used the moscot.spatiotemporal transition matrix aggregated over annotations. We assigned each bin at E15.5 with the most probable annotation. Once we had the annotations for E15.5, we repeated this procedure to earlier time points. To evaluate the accuracy of the annotations, we used Scanpy's rank_genes_groups with respect to the inferred annotations. For each annotation, we queried whether the marker genes, as previously reported[10], were within the top 50 ranked genes. We reported the percentage of annotations for which this condition held.

**CellRank analysis.** We used CellRank to infer marker genes associated with terminal states. To define the CellRank kernel, $K$, a matrix containing bins from all time points, was used to obtain the transition probabilities between bins as follows:

1. Obtain a sparse representation of the moscot.spatiotemporal transition maps. These transition matrices occupy the superdiagonal of $K$ as they transport bins from early to late time points.
2. Compute the transition matrices within each time point based on gene-expression similarity. These values occupy the diagonal of $K$.
3. Combine the above with weights 0.9 and 0.1, respectively to obtain $K$.
4. Row-normalize $K$.

We used GPCCA estimator[142] in CellRank to compute terminal states, independently, for the full embryo and brain bins. We defined each terminal state by assigning the 30 most likely bins to it. We computed absorption probabilities on the Markov chain to these combined sets per terminal state group and interpreted these as fate probabilities. We correlated expression for each gene with the computed fate probabilities across all bins. We identified the top 20 most strongly correlated genes and TFs per terminal group. The list of mouse TFs was downloaded from AnimalTFDB (http://bioinfo.life.hust.edu.cn/AnimalTFDB/#!/download).

## Moscot.spatiotemporal of *Drosophila* embryo

We set out to investigate how moscot.spatiotemporal could be used to study the embryo development of *Drosophila* and we leveraged a 3D dataset profiled using Stereo-seq technology[10]. We downloaded the preprocessed dataset provided by the authors[33]. We performed highly variable gene selection, PCA and dimensionality reduction using UMAP with Scanpy. We then solved a moscot.spatiotemporal problem using PCA embedding for the linear term and normalized spatial coordinates in the quadratic term. We performed cell transition analysis as described for moscot.time and visualized transported mass between the source and target using both the push and pull operator and cell-transition matrices. We further performed a pull operation of indicator vectors of cell types of the CNS and muscle tissue and correlated it with the expression of all genes. We visualized the highest correlating genes, which revealed key TFs, both previously unknown and previously reported by the authors, involved in CNS and muscle tissue development.

## Ethics statement

Animal studies were conducted with adherence to relevant ethical guidelines for the use of animals in research in agreement with German animal welfare legislation with the approved guidelines of the Society of Laboratory Animals (GV-SOLAS) and the Federation of Laboratory Animal Science Associations (FELASA). The study was approved by the Helmholtz Munich Animal Welfare Body and by the Government of Upper Bavaria. NVF mice were kept at the central facilities at Helmholtz Munich under specific pathogen-free conditions in animal rooms with light cycles of 12–12 h, temperature of 20–24 °C and humidity of 45–65%. The mice received sterile filtered water and a standard diet for rodents ad libitum.

## Reporting summary

Further information on research design is available in the Nature Portfolio Reporting Summary linked to this article.

## Data availability

The mouse embryogenesis atlas[7] is available at http://tome.gs.washington.edu. The mouse liver CITE-seq data[26] is available at https://www.livercellatlas.org/. The Vizgen MERSCOPE liver and brain coronal sections dataset is available at the Vizgen public dataset release website https://vizgen.com/data-release-program/. The datasets for

benchmarking the spatial mapping problems were taken from a previous publication[25]. The spatiotemporal atlas of mouse embryogenesis (MOSTA)[10] is available at https://db.cngb.org/stomics/mosta/. The spatiotemporal *Drosophila* dataset[33] is available at https://db.cngb.org/stomics/flysta3d/. The single-cell RNA-seq dataset[34] is available from the Gene Expression Omnibus (GEO; https://www.ncbi.nlm.nih.gov/geo/query/acc.cgi?acc=GSE132188). The pancreas multiome data are available from the GEO (accession code GSE275562).

## Code availability

The moscot software package is available at https://moscot-tools.org, which includes documentation, tutorials and examples. The code to reproduce our analysis is available from GitHub (https://github.com/theislab/moscot-framework_reproducibility) as is the code to reproduce the benchmarking experiments (https://github.com/theislab/moscot_benchmarks).

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

**Acknowledgements** We thank M. Ansari for the design of the moscot logo; staff at the Genomics Core Facility of Helmholtz Munich who performed the sequencing of the pancreas data; I. Ibarra for input on motif databases and comments on the manuscript; S. Jimenez for improvements to the manuscript; M. Scetbon for insights into the low-rank OT approximations; A. Tong and G. Huguet for advice on geodesic cost functions; A. Danilina and S. Ozleyen for contributions to the moscot software; all members of the Theis and Treutlein laboratories for discussions; and J. Jaki for technical support. This work was supported by the BMBF-funded de.NBI Cloud within the German Network for Bioinformatics Infrastructure (de.NBI) (031A532B, 031A533A, 031A533B, 031A534A, 031A535A, 031A537A, 031A537B, 031A537C, 031A537D and 031A538A). G.P. is supported by the Helmholtz Association under the joint research school Munich School for Data Science and by the Joachim Herz Foundation. M.L. acknowledges financial support from the Joachim Herz Foundation and through an EMBO Postdoctoral Fellowship. Z.P. is supported by a scholarship for outstanding doctoral students in data science from the Israeli Council for Higher Education and the Clore Scholarship for Ph.D. students. M.N. is supported by an Azrieli Foundation Early Career Faculty Fellowship, the Center for Interdisciplinary Data Science Research at the Hebrew University of Jerusalem, the Israel Science Foundation (grant no. 1079/21) and the European Union (ERC, DecodeSC, 101040660). F.J.T. acknowledges support by Wellcome Leap as part of the ΔTissue Program and by the European Union (ERC, DeepCell no. 101054957). For all support coming through EU funding, the views and opinions expressed are those of the author(s) only and do not necessarily reflect those of the European Union or the European Research Council. Neither the European Union nor the granting authority can be held responsible for them.

**Author contributions** M.L. conceived the initial idea of the project, guided by F.J.T. D.K., G.P., M.L., M.K. and Z.P. conceptualized the project. M.K., D.K. and G.P. designed and implemented the moscot software package with contributions from I.G., Z.P. and L.M.-P. M.G. benchmarked and conducted analyses for moscot.time together with M.L., with contributions from D.K. G.P. benchmarked and conducted analyses for moscot.space with contributions from D.K. Z.P. conducted analyses for moscot.spatiotemporal with contributions from G.P. and D.K. A.B.-P., M.T.-M., C.J. and M.S. generated the pancreas data under supervision of M.B. and H.L. D.K. processed and analysed the pancreas dataset with contributions from M.S., M.B., S.P., C.J. and M.L. L.S. and P.C. generated iPS cell differentiation data under the supervision of M.B. D.K., M.L. and G.P. wrote the manuscript with contributions from all authors. M.L. and D.K. wrote the Supplementary Notes. F.J.T., M.C. and M.N. supervised the project and contributed to the conception of the project. All authors read and approved the final manuscript.

**Funding** Open access funding provided by Helmholtz Zentrum München - Deutsches Forschungszentrum für Gesundheit und Umwelt (GmbH).

**Competing interests** F.J.T. consults for Immunai, Singularity Bio, CytoReason, Cellarity and Omniscope, and has ownership interest in Dermagnostix and Cellarity. The remaining authors declare no competing interests.

**Additional information**
**Correspondence and requests for materials** should be addressed to Heiko Lickert or Fabian J. Theis.

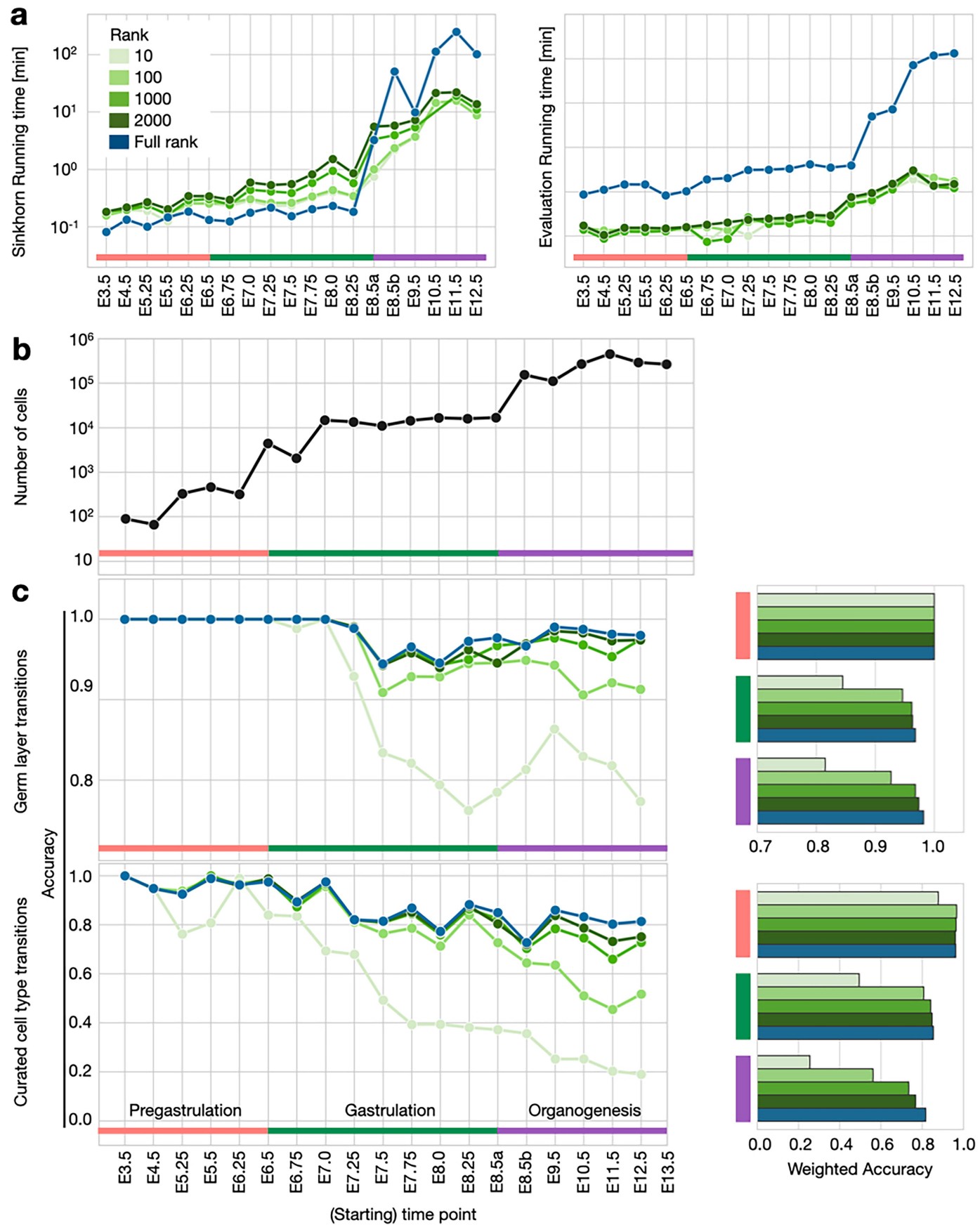

**Extended Data Fig. 1** | See next page for caption.

**Extended Data Fig. 1 | Low-rank approximates full-rank Sinkhorn at faster running times. a**. Runtime in minutes to compute a coupling matrix (left) and to evaluate algorithm performance (right), across time points on the embryogenesis data[7] of Fig. 2, for full-rank Sinkhorn (default moscot.time) and low-rank Sinkhorn[13–15] for various ranks (Methods). **b**. Cell number per time point. **c**. Comparing low and full-rank approaches in terms of the germ-layer (top) and curated transition (bottom) metrics of Fig. 2, for individual time points (left) and aggregated over time-windows (right, Methods).

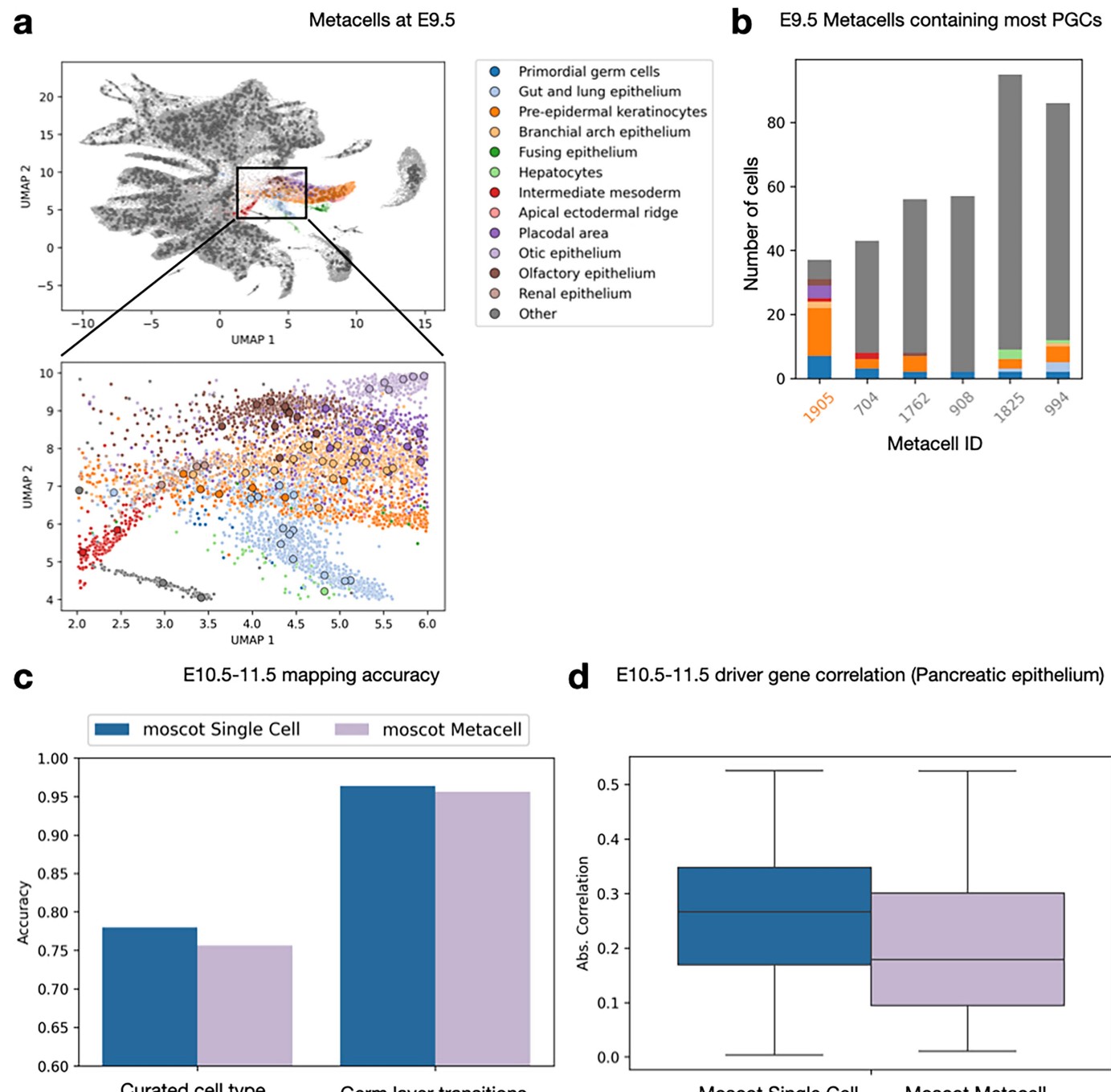

**Extended Data Fig. 2 | Metacells do not resolve PGCs and metacell mapping degrades driver gene correlation for Pancreatic epithelium. a.** UMAP of E9.5 cells, visualizing individual cells (small dots) and metacells (large dots) computed using Metacell-2[21] (Methods). Colors indicate PGCs and cell types that co-occur in metacells with PGCs. The zoom-in highlights PGCs, which are not captured by any metacell. **b.** Bar chart over cell-type composition for the six metacells at E9.5 containing most PGCs. No metacell received the "PGC" label because they are dominated by other cell types. **c,d.** Comparing moscot mapping at E10.5-11.5 on the single cell versus metacell levels in terms of the curated transition and germ layer scores (**c**) and correlation between Pancreatic epithelium ancestor probabilities and known driver gene expression (**d**; Methods).

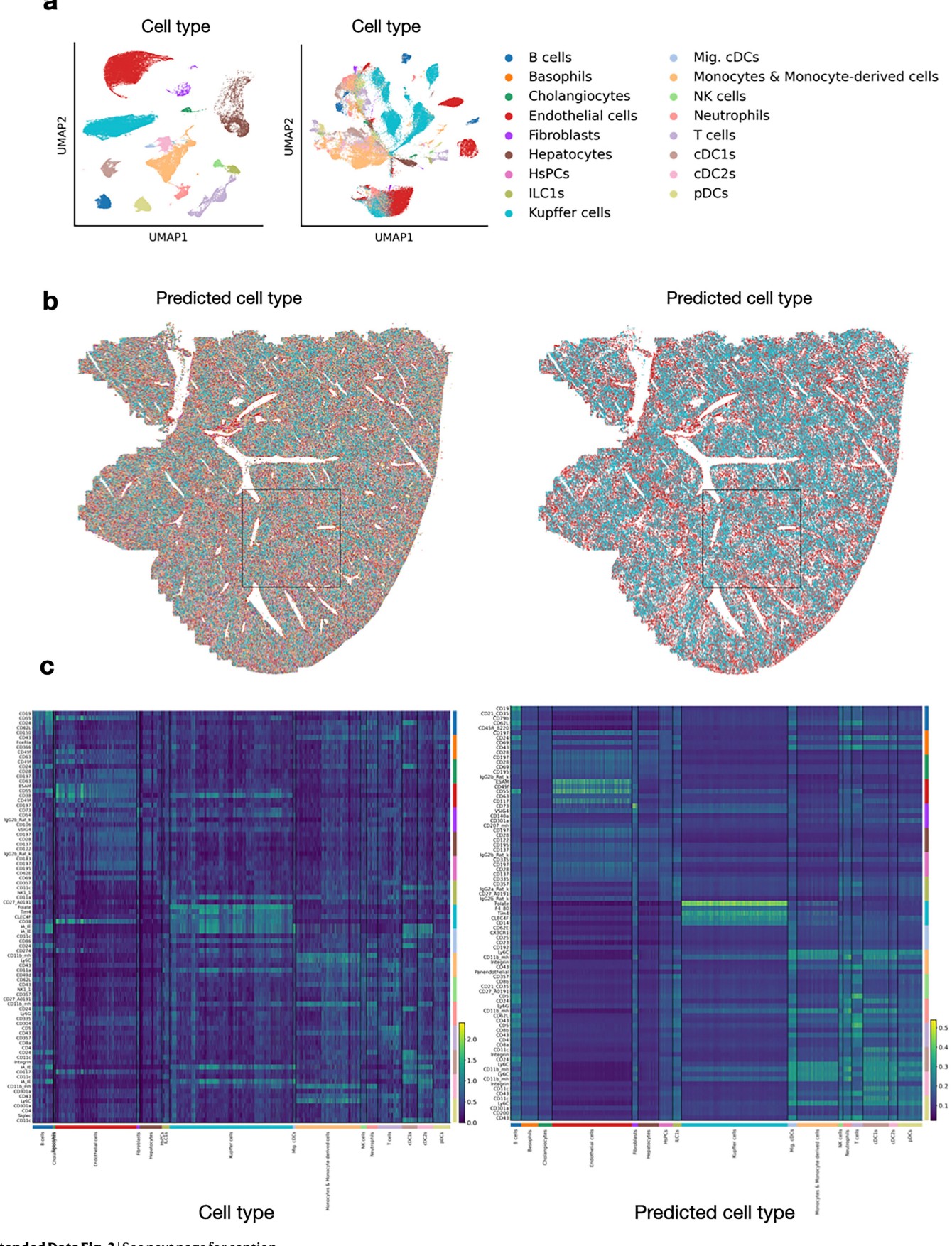

**Extended Data Fig. 3** | See next page for caption.

**Extended Data Fig. 3 | Overview of CITE-seq data and mapped annotations.**
**a**. UMAP embedding of single-cell (left) and CITE-seq (right) dataset, respectively. Labels were provided in the original publication. **b**. Cell type annotation mapped in spatial coordinates. All cell types visualized in space (left), and spatial plot of only Kupffer cells (blue) and Endothelial cells (red, right). Boxes in solid lines correspond to insets in Fig. 3. **c**. Top five differentially expressed proteins (five genes/proteins per cluster in rows) in original CITE-seq dataset (left) and predicted cell types and protein expression in space (right).

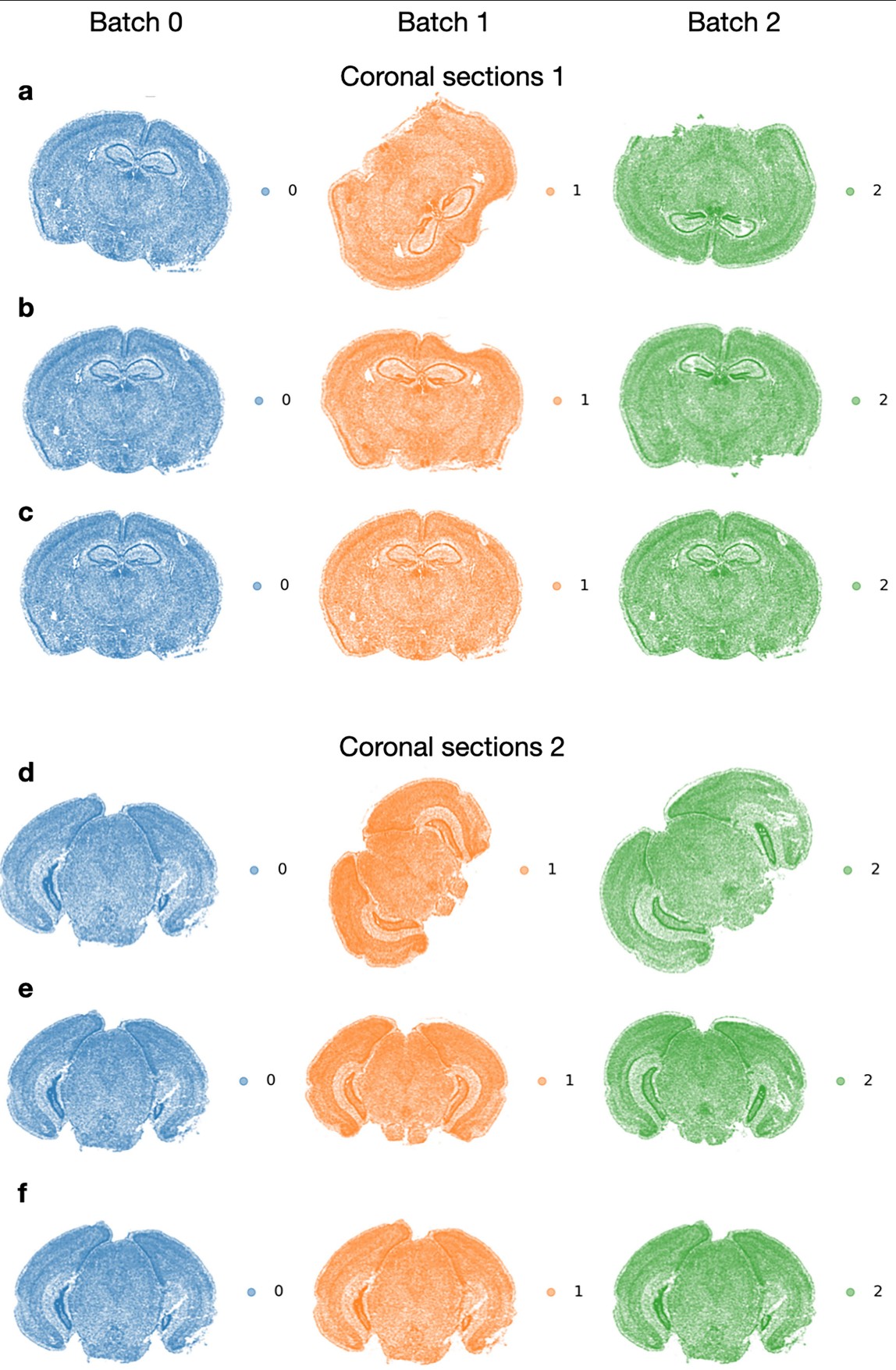

Batch 0  Batch 1  Batch 2

Coronal sections 1

**a**

**b**

**c**

Coronal sections 2

**d**

**e**

**f**

**Extended Data Fig. 4** | See next page for caption.

**Extended Data Fig. 4 | Alignment of spatial transcriptomics data of sections of the mouse brain. a**. Spatial visualization of the three coronal sections from three different mouse brains before the alignment. **b**. Spatial visualization of the three coronal sections after affine alignment. **c**. Spatial visualization of the three coronal sections after warping alignment. **d.-f**. Original, affine transform and warped transformed tissue slices from the second set of three coronal sections from three different mouse brains.

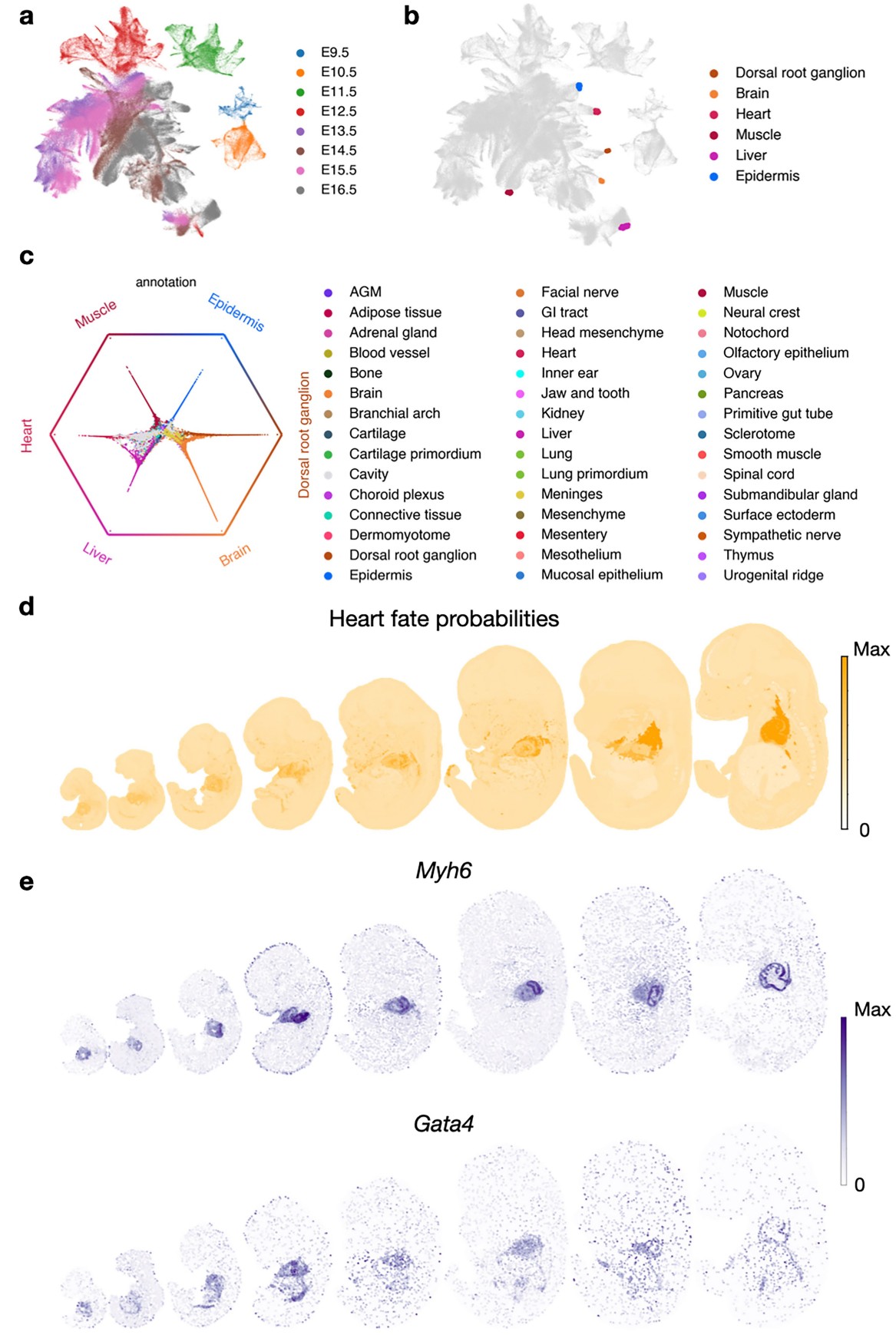

**Extended Data Fig. 5** | See next page for caption.

**Extended Data Fig. 5 | Analysis of development lineages by interfacing moscot.spatiotemporal with CellRank 2. a**. UMAP representation of the spatiotemporal atlas of mouse embryogenesis (MOSTA) over eight time points, from E9.5 to E16.5[10] colored by time points. **b**. UMAP colored by macrostates identified by CellRank 2. **c**. Projection of cell's absorption probabilities towards identified macrostates. Cells colored by lineage annotation. **d**. CellRank 2 fate probabilities for heart fate visualized in spatial coordinates **e**. Spatial visualization of driver genes identified for the heart development lineage, *Myh6* (top) and *Gata4* (bottom).

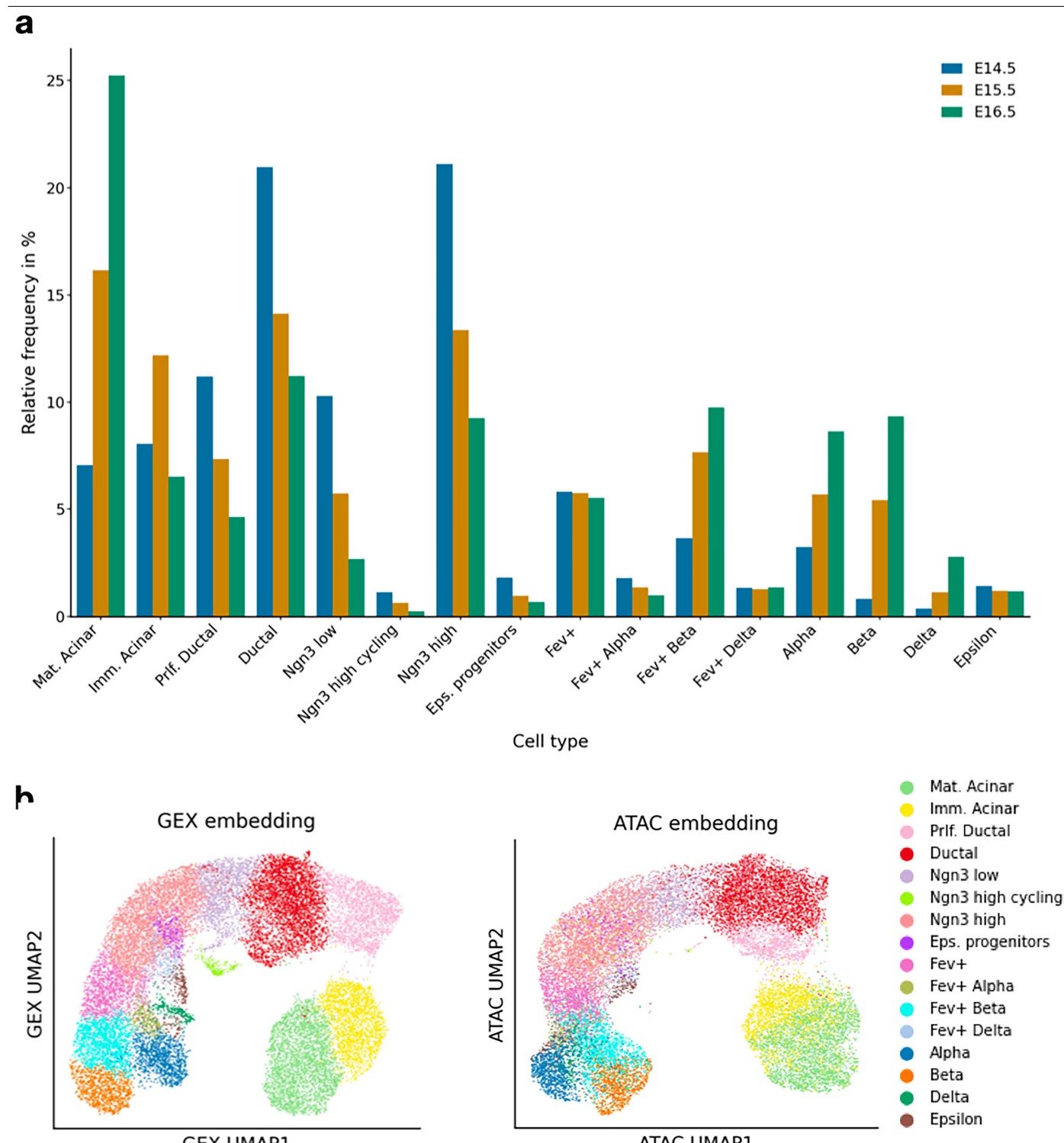

**Extended Data Fig. 6 | Summary statistics and visualization of the pancreatic endocrinogenesis dataset. a**. Distribution of cell types per time point. **b**. UMAP embeddings based on graphs constructed from gene expression (left, Methods) and open chromatin accessibility (right, Methods).

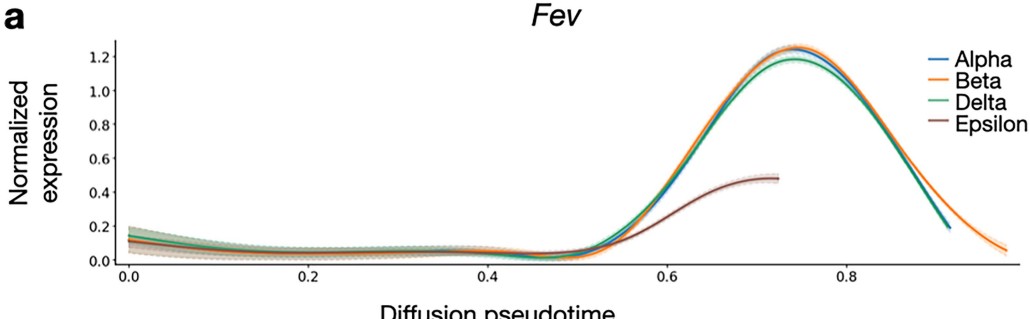

**a**

*Fev*

**b**

Extended Data Fig. 7 | *Fev* expression over pseudotime per lineage.
**a**. Normalized expression of *Fev* over pseudotime[134] computed with *cellrank*.
*pl.gene_trends* building on CellRank's pseudotime kernel. **b**. Normalized gene expression of *Fev* and each islet hormone for the respective lineage plotted over pseudotime.

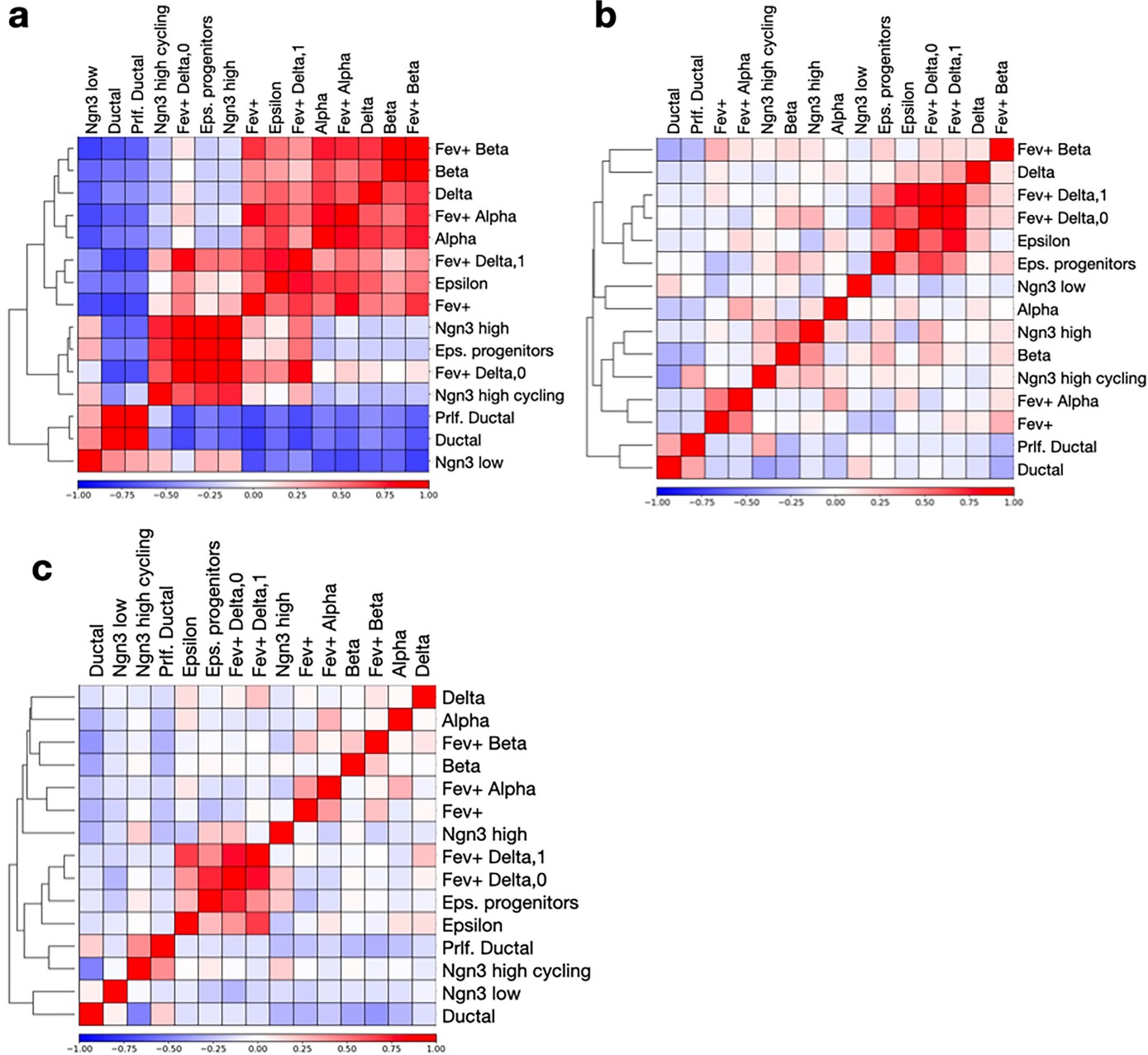

**Extended Data Fig. 8 | Similarity of cell types based on different modalities.**
**a**. Aggregated correlation matrix of refined cell types based on processed gene expression, computed via scanpy's *dendrogram* function, followed by *scanpy.pl.correlation_matrix*. The gene expression data was preprocessed by normalization (*scanpy.pp.normalize_total*) and log1p-transformation, followed by 30-dimensional PCA computation. **b**. Aggregated correlation matrix of cell types based on processed ATAC peak counts, computed via *scanpy.tl.dendrogram* followed by *scanpy.pl.correlation_matrix*. The peak counts were preprocessed using tfidf-transformation (*muon.atac.pp.tfidf*), followed by normalization and log1p-transformation, before computing a singular value decomposition and removal of dimensions which are highly correlated with library size. **c**. Aggregated correlation matrix of cell types based on both gene expression and open chromatin accessibility. After scaling both modalities to unit variance, the processed gene expression was concatenated with the processed LSI embedding.

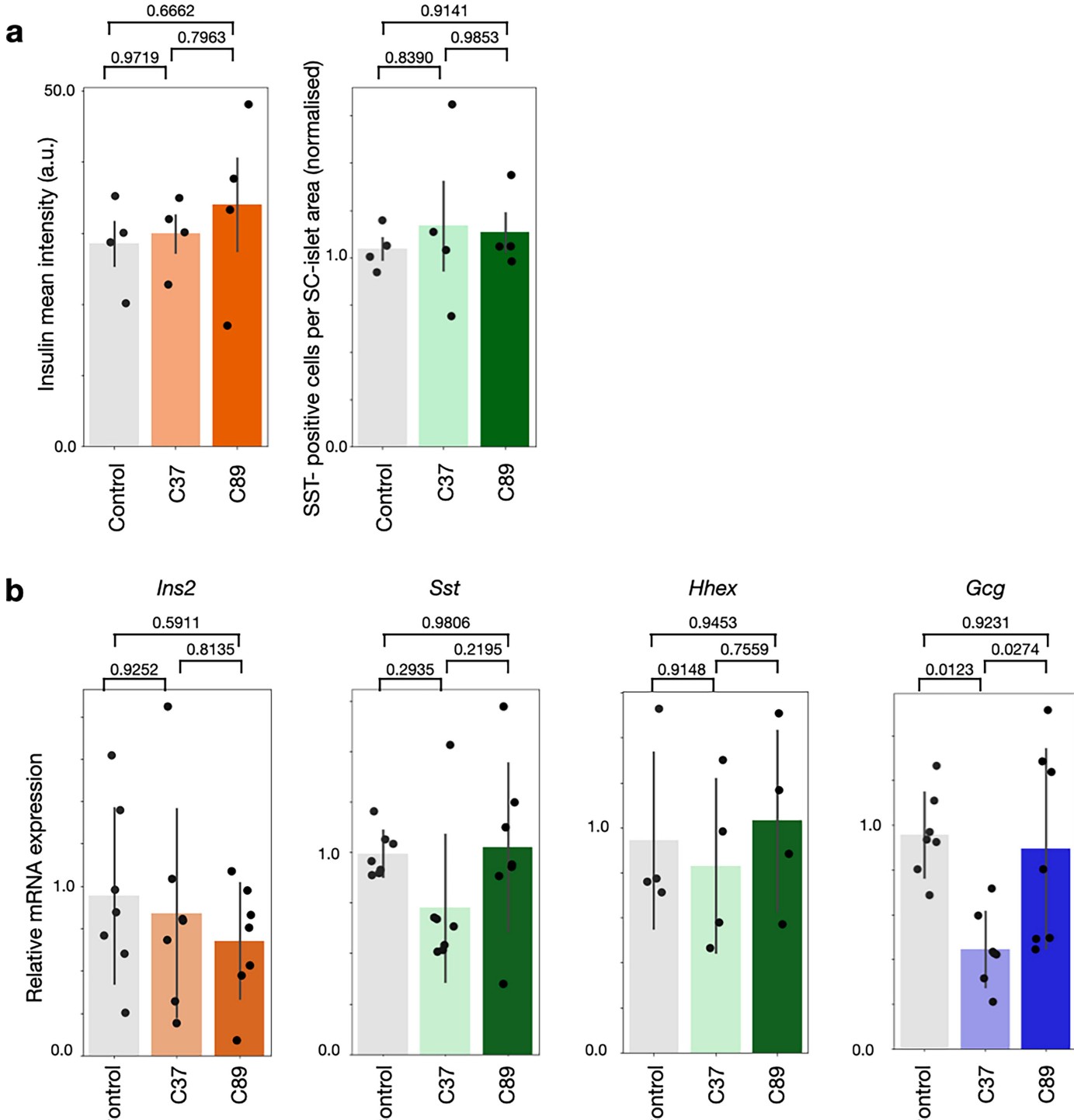

**Extended Data Fig. 9 | NEUROD2 knockout experiments in human iPSCs-derived islet cells. a**. Insulin mean intensity and the number of SST-positive cells measured with immunostaining (Methods) for control, clone 37 and clone 89[79] ($n$ = 4 independent experiments, standard error shown). **b**. Relative mRNA expression of *Ins2*, *SSt*, *HHEX*, and *GCG* measured with qPCR (n = 7 biologically independent samples for *Ins2*, *Sst*, n = 4 for *Hhex*, n = 7 for *Gcg*). We report mean and standard deviation (Methods), p-values obtained from one-sided ANOVA test with Tukey multiple comparison correction.

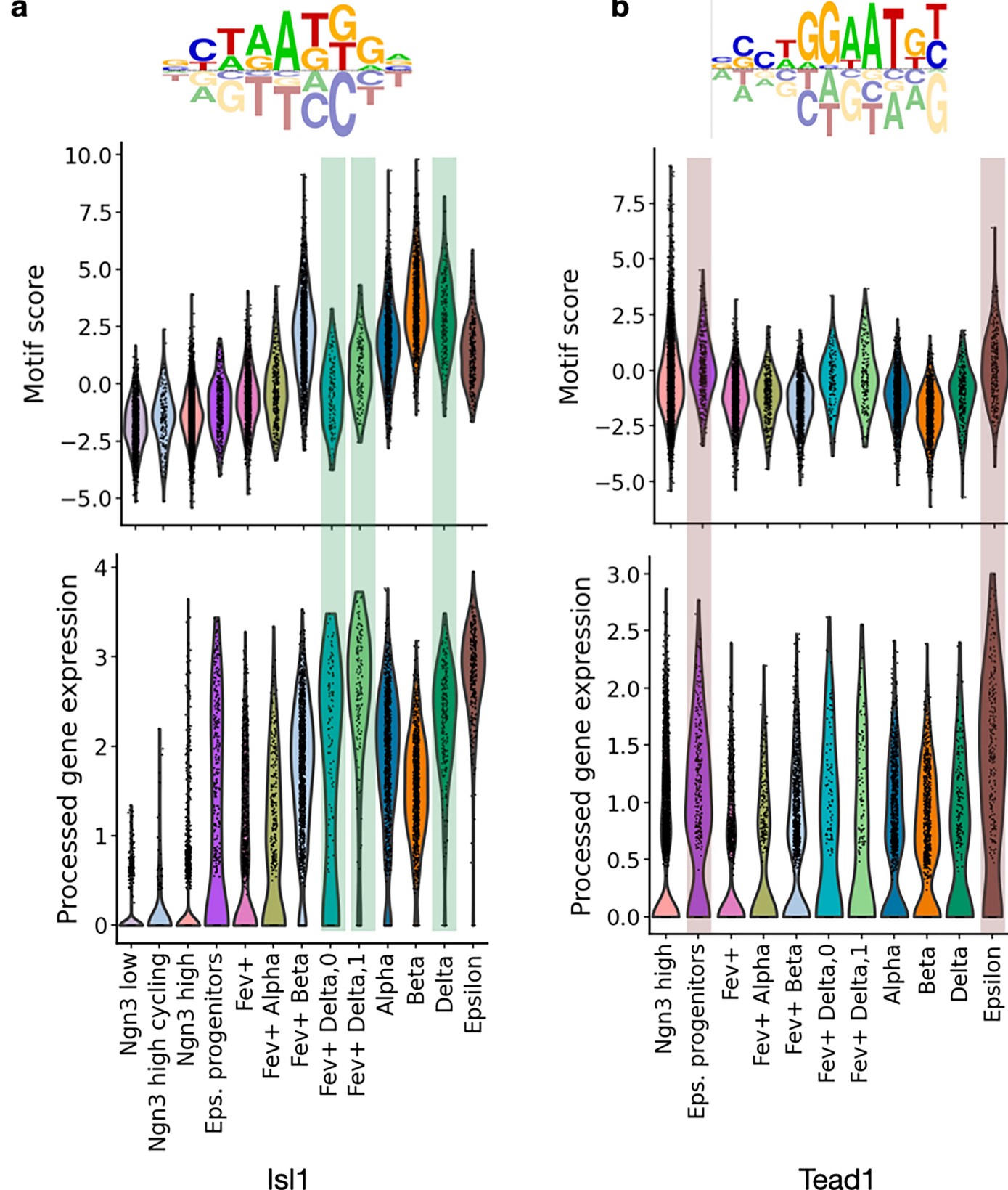

**Extended Data Fig. 10 | Delta and epsilon motif activity calculated with moscot.time. a**. Motif with cisBP identifier M09209_2.00, a marker motif identified for the delta population. For a motif to be active both the motif activity score (top) and the gene expression (associated transcription factor, here *Isl1*) should be high. Delta cells and their conjectured progenitors are underlaid in green (Supplementary Table 30). **b**. Motif with cisBP identifier M09438_2.00, which we identified to be a marker motif for the epsilon population (Supplementary Table 31). The motif is associated with the *Tead1* transcription factor (Methods).

# Reporting Summary

## Statistics

For all statistical analyses, confirm that the following items are present in the figure legend, table legend, main text, or Methods section.

| n/a | Confirmed | |
|---|---|---|
| ☐ | ☒ | The exact sample size (*n*) for each experimental group/condition, given as a discrete number and unit of measurement |
| ☒ | ☐ | A statement on whether measurements were taken from distinct samples or whether the same sample was measured repeatedly |
| ☐ | ☒ | The statistical test(s) used AND whether they are one- or two-sided<br>*Only common tests should be described solely by name; describe more complex techniques in the Methods section.* |
| ☐ | ☒ | A description of all covariates tested |
| ☐ | ☒ | A description of any assumptions or corrections, such as tests of normality and adjustment for multiple comparisons |
| ☐ | ☒ | A full description of the statistical parameters including central tendency (e.g. means) or other basic estimates (e.g. regression coefficient) AND variation (e.g. standard deviation) or associated estimates of uncertainty (e.g. confidence intervals) |
| ☐ | ☒ | For null hypothesis testing, the test statistic (e.g. *F*, *t*, *r*) with confidence intervals, effect sizes, degrees of freedom and *P* value noted<br>*Give P values as exact values whenever suitable.* |
| ☒ | ☐ | For Bayesian analysis, information on the choice of priors and Markov chain Monte Carlo settings |
| ☒ | ☐ | For hierarchical and complex designs, identification of the appropriate level for tests and full reporting of outcomes |
| ☐ | ☒ | Estimates of effect sizes (e.g. Cohen's *d*, Pearson's *r*), indicating how they were calculated |

*Our web collection on statistics for biologists contains articles on many of the points above.*

## Software and code

Policy information about availability of computer code

| Data collection | No software was used |
|---|---|

| Data analysis | Python 3.11<br>moscot 0.3.4<br>ott-jax=0.4.5<br>scanpy=1.9.4<br>pywavelets=1.5.0<br>squidpy=1.3.1<br>cellrank=2.0.0<br>anndata=0.9.2<br>muon=0.1.5<br>mudata=0.2.3<br>scvelo=0.2.5<br>pygpcca=1.0<br>r-shiny=1.8.<br>signac=2.1.0<br>r-seurat=4.4.0<br><br>reproducibility: https://github.com/theislab/moscot-framework_reproducibility |
|---|---|

For manuscripts utilizing custom algorithms or software that are central to the research but not yet described in published literature, software must be made available to editors and reviewers. We strongly encourage code deposition in a community repository (e.g. GitHub). See the Nature Portfolio guidelines for submitting code & software for further information.

## Data

Policy information about availability of data

All manuscripts must include a data availability statement. This statement should provide the following information, where applicable:
- Accession codes, unique identifiers, or web links for publicly available datasets
- A description of any restrictions on data availability
- For clinical datasets or third party data, please ensure that the statement adheres to our policy

The mouse embryogenesis atlas by Qiu et al is available at http://tome.gs.washington.edu. The mouse liver CITE-seq by Guilliams et al. is available at https://www.livercellatlas.org/. The Vizgen MERSCOPE Liver and Brain coronal sections dataset is available at the Vizgen public dataset release website https://vizgen.com/data-release-program/. The datasets for benchmarking the spatial mapping problems were taken from the original publication of Li et al. The spatiotemporal atlas of mouse embryogenesis (MOSTA) by Chen et al is available at https://db.cngb.org/stomics/mosta/. The spatiotemporal drosophila dataset is available at https://db.cngb.org/stomics/flysta3d/. The pancreas data by Basitdas-Ponce et al. is available under GEO accession code GSE132188 at https://www.ncbi.nlm.nih.gov/geo/query/acc.cgi?acc=GSE132188. The pancreas multiome data is available under GEO accession code GSE275562.

## Research involving human participants, their data, or biological material

Policy information about studies with human participants or human data. See also policy information about sex, gender (identity/presentation), and sexual orientation and race, ethnicity and racism.

| Reporting on sex and gender | N/A |
|---|---|
| Reporting on race, ethnicity, or other socially relevant groupings | N/A |
| Population characteristics | N/A |
| Recruitment | N/A |
| Ethics oversight | N/A |

Note that full information on the approval of the study protocol must also be provided in the manuscript.

# Field-specific reporting

Please select the one below that is the best fit for your research. If you are not sure, read the appropriate sections before making your selection.

☒ Life sciences          ☐ Behavioural & social sciences          ☐ Ecological, evolutionary & environmental sciences

For a reference copy of the document with all sections, see nature.com/documents/nr-reporting-summary-flat.pdf

# Life sciences study design

All studies must disclose on these points even when the disclosure is negative.

| | |
|---|---|
| Sample size | Sample size was determined based on the available experimental data. |
| Data exclusions | Data exclusion based on standard single-cell preprocessing pipelines. |
| Replication | Replication was not feasible due to financial reasons |
| Randomization | no experimental groups |
| Blinding | no experimental groups |

# Reporting for specific materials, systems and methods

We require information from authors about some types of materials, experimental systems and methods used in many studies. Here, indicate whether each material, system or method listed is relevant to your study. If you are not sure if a list item applies to your research, read the appropriate section before selecting a response.

## Materials & experimental systems

| n/a | Involved in the study |
|---|---|
| ☐ | ☒ Antibodies |
| ☐ | ☒ Eukaryotic cell lines |
| ☒ | ☐ Palaeontology and archaeology |
| ☐ | ☒ Animals and other organisms |
| ☒ | ☐ Clinical data |
| ☒ | ☐ Dual use research of concern |
| ☒ | ☐ Plants |

## Methods

| n/a | Involved in the study |
|---|---|
| ☒ | ☐ ChIP-seq |
| ☒ | ☐ Flow cytometry |
| ☒ | ☐ MRI-based neuroimaging |

## Antibodies

| | |
|---|---|
| Antibodies used | goat anti-somatostatin, 1:300, polyclonal (D-20) (Santa Cruz Biotechnology, SC-7819)<br>Mouse anti ghrelin, 1:250, monoclonal (2F4) (Santa Cruz Biotechnology, SC-293422) |
| Validation | These are commercially available antibodies and their specificities have been tested using primary human islets. |

## Eukaryotic cell lines

Policy information about cell lines and Sex and Gender in Research

| | |
|---|---|
| Cell line source(s) | Human iPSC cell lines: C-peptide-mCherry reporter hiPSC line (HMGUi001-A-8) (Siehler et al., 2021) and NEUROD2nVenus/nVenus iPSC clones (HMGUi001-A-42) (Cota et al., 2023) were used. |
| Authentication | No cell line was authenticated |
| Mycoplasma contamination | All used cell lines were routinely tested to ensure that they were negative for mycoplasma. |
| Commonly misidentified lines<br>(See ICLAC register) | one of the used cell lines are reported in ICLAC register |

## Animals and other research organisms

Policy information about studies involving animals; ARRIVE guidelines recommended for reporting animal research, and Sex and Gender in Research

| | |
|---|---|
| Laboratory animals | The embryonic stages of the used embryos are embryonic days (E) E14.5, E15.5 and E16.5. The strain of the mouse line Ngn3-Venus fusion was on mixed background (C57BL/6J × 129/SvJ). The mice were kept at the central facilities at Helmholtz Center Munich (HMGU) under Specific-pathogen-free (SPF) conditions. Animal rooms had a light cycle of 12/12 h, temperature of 20–24 C and humidity of 45–65%. Mice received sterile filtered water and standard diet for rodents ad libitum. |
| Wild animals | No wild animals were used in this study |

| Reporting on sex | Since we used only embryonic samples there is no information on the sex of the animals. |
|---|---|
| Field-collected samples | The study did not include field collected samples. |
| Ethics oversight | Animal studies were conducted with adherence to relevant ethical guidelines for the use of animals in research in agreement with German animal welfare legislation with the approved guidelines of the Society of Laboratory Animals (GV-SOLAS) and the Federation of Laboratory Animal Science Associations (FELASA). The study was approved by the Helmholtz Munich Animal Welfare Body and by the Government of Upper Bavaria. |

Note that full information on the approval of the study protocol must also be provided in the manuscript.

## Plants

| Seed stocks | N/A |
|---|---|
| Novel plant genotypes | N/A |
| Authentication | N/A |

