## [Peer Review file · Nature]

Manuscript Title: Mapping cells through time and space with moscot

Editorial Notes:

Redactions – unpublished data

Reviewer Comments & Author Rebuttals

Reviewer Reports on the Initial Version:

Referees' comments:

Referee #1 (Remarks to the Author):

Manuscript Review (Nature)
Mapping cells through time and space with moscot
Accept with major revision

Summary

In the paper, the authors present a new software framework called moscot, which consists of four functionalities: moscot.time, moscot.space.mapping, moscot.space.alignment, moscot.spatiotemporal. In moscot.time, the authors make a clever use of the Sinkhorn algorithm which vastly expands the scaling of the Waddington-OT method to big datasets, which they show in figure 2. In moscot.space.mapping, they implement a variant of the NovoSpaRc algorithm which allows them to map dissociated single cell data (such as scRNA) onto spatial transcriptomics data (using cost matrices in RNA space and spatial space, respectively). In moscot.space.alignment this algorithm is used with the aim of integrating spatial transcriptomics datasets (where the cost matrices are in spatial space for both datasets); and in moscot.spatiotemporal they perform trajectory inference by integrating the spatial transcriptomic datasets of the different time points (as in moscot.space.mapping).

The authors compare their variant of NovoSpaRc with Tangram and gimVI, and they document significant improvement in correlation between spatial correspondence and predicted gene expression (figure 3). They also show that moscot.spatiotemporal can be used to perform cell type annotation with significant matching with ground truth annotation precisely thanks to its use of the spatial information (figure 4). Finally, interestingly, in a multiome (scRNA+ATAC), time-resolved pancreas development dataset, moscot.spatiotemporal maps delta cells to epsilon cell progenitors, thereby suggesting a shared lineage, which is corroborated by several evidence provided in terms of gene markers and similarity of genome accessibility.

Overall, although the methodological development is not groundbreakingly innovative, their method has the benefits of vastly improving data scalability, accommodating multimodal datasets, and providing a common framework for any kind of optimal transport analysis which the community will find useful.

Overall evaluation: Accept with major revisions

Major Points for Revision

- In Waddington-OT the underlying assumption is that development proceeds locally linearly, which indeed is inconsistent with a global optimal transport. In moscot, the authors seem to adopt the hidden assumption of such global optimal transport, for instance when they map scRNA data on spatial transcriptomic data, they assume that optimal transport is the way biology introduces batch effects. The paper needs a further experiment as follows to validate this point: take three different batches (A,B,C; say batch B is the spatial transcriptomic dataset) find the matrices $P_{\{AB\}}$, $P_{\{BC\}}$, $P_{\{AC\}}$. If the global optimal transport assumption holds then it should hold that $P_{\{AB\}} P_{\{BC\}} = P_{\{AC\}}$.
- Similarly, there should be a test for moscot.spatiotemporal test where we see if the optimal transport is consistent across time points, ie for time points t_1, t_2, t_3 , it should hold that $P_{\{t_1,t_2\}} P_{\{t_2,t_3\}} = P_{\{t_1,t_3\}}$.
- In fig. 3, there needs to be a comparison of moscot.space.alignment with NovoSpaRc to test whether including many modalities has an improvement.
- It would be better if the newly hypothesised shared ancestry between the delta and epsilon cells is experimentally validated, perhaps by perturbing driver transcription factors at the early precursor stage of development.

Minor Points for Revision

It is not clear how the a marginal allows for cell birth. Specifically since $0 \leq a_i \leq 1$, it means the $\sum_i P_{\{ij\}} = a_i = 1$ so the model could only accommodate loss of cells. More explanation is needed in the text. In supplementary note 5, it is unclear how the Gibbs kernel which has a parameter ϵ is mapped to the heat kernel with parameter t , so the authors need to elaborate on this.

In fig.2 there are only comparisons with TOME, not with Waddington-OT. It would be useful to have a plot showing that the differences between Waddington-OT and moscot.time are negligible.

In methods section 1.3.1 the authors mention that the definition of C^X will follow below. In the following section (below equation 15), they allude to a definition of C^X made "above", but the definition is nowhere in between. The definition needs to be clearly stated upon first mention of C^X .

Referee #2 (Remarks to the Author):

Klein et al. have developed an integrative and scalable method called "moscot" that utilizes optimal transport (OT) for analyzing multimodal single-cell data. This framework demonstrates great potential for analyzing complex single-cell genomics datasets, including spatial and temporal single-cell data. The authors showcase the feasibility of moscot through the analysis of various published datasets, such as mouse embryogenesis scRNA-seq, spatial liver MERSCOPE data, and a time-resolved spatial

transcriptomics dataset of mouse embryogenesis. Additionally, they generate paired scRNA-seq and scATAC-seq data from Ngn3+ cells during mouse pancreatic development and predict the lineage hierarchy of delta and epsilon cells.

Overall, moscot presents a promising tool for single cell genomic analysis. Optimal transport is certainly the right theoretical framework for solving these mapping and alignment problems, and the moscot framework introduces a scalable implementation which can be integrated into existing workflows. However, the biology prediction is weak without experiment validation. Some major concerns need to be addressed:

1. In the pancreatic development part, the limited time points (E14.5 and E15.5) covered in this study may overlook other potential differentiations and the prediction is not experimentally validated. Ngn3's expression can be detected from E12.5 (Byrnes et al, 2018). Different endocrine cells including delta and epsilon cells are likely to fully differentiate at late development stage (Yu et al, 2021). However, this study only covers two development time points in the middle. It is very possible to miss other differentiation possibilities and the data can be misinterpreted. Another thing is EPs are identified only by Ngn3, possibly neglecting important distinctions between EPs. Previous lineage tracing experiments have indicated that islet cell fate is determined before hormone expression (Herrera et al, 1998 & 2000). However, when EPs diverge to differentiate into specific islet cell types is not known, therefore whether this decision occurs before, during, or after Ngn3 expression is unknown. Therefore, samples from more time points are necessary and experiment validation should be done.

2. How dependent are these results on existing high quality latent representations?

For example, in supp methods sec 2.1.1, analysis seems dependent on proper batch correction done by seurat's Find Integration Anchors function which can already take some fine-tuning to get a desirable result.

Or in section 2.1.2, many specific steps must be taken to properly embed each modality separately (e.g remove 1st and 5th PCs, discarding ambient genes, etc).

Or in analysis of the pancreatic development data, many different filtering criteria had to be applied to each timepoint with varying (likely carefully tuned) parameters.

This is sometimes glossed over in single-cell projects as a major time sink and source of variability in results so I wonder how robust moscot is to variations in the initial latent representations? Especially when the geodesic cost functions depend on the knn graph produced from these representations.

Minor points:

3. In first paragraph of results section (pg 4) "moscot takes as input unpaired datasets"

But in moscot supp methods sec 1.2.3 "this naturally extends beyond two modalities towards any number of jointly measured modalities"

-Somewhat unclear how suitable moscot is for diagonal integration (i.e unpaired datasets--different cells, different features)

-There are some examples of diagonal integration (e.g the pancreatic development dataset) but does this differ from how moscot is applied to paired data?

4. When adjusting the marginals for growth and death (supp methods pg 4) are the apoptosis scores and proliferation scores computed from known markers?

How would moscot behave if these marginals were not adjusted?

5. Might want to include explanations/intuition for what push and pull operations are in OT and how they can be interpreted in the context of cell state transitions

-Specifically in supp methods 1.2.4 I found it unclear why correlating pull and push distributions with gene expression will uncover putative driver genes

6. What does it mean in supp methods sec 1.3.3 that this "is the first method to make use of multimodal information in the actual mapping problem"?

-Is this referring to jointly learning a multi-modal latent representation?

Referee #3 (Remarks to the Author):

In this submitted manuscript, Dominik and others have made commendable strides in creating Moscot, a general-purpose toolkit for analyzing single cell and spatial transcriptomics datasets. This represents a significant engineering effort and is poised to contribute meaningfully to the genomics field building upon their scverse ecosystem. I appreciate the study's timeliness and potential. I would like to present my review in the following first addressing general comments and then discussing specific aspects of the study.

One of my principal reservations regarding the manuscript is its lack of novelty and significant biological insights, especially considering the high standards of a Nature submission. Concerning the computational advancements, the use of optimal transport (OT)-based methods for single cell and spatial transcriptomics is not new but multiple studies over the past a few years (e.g., WOT, COMMOT, PASTE2) has reported similar efforts (although in more isolated manner). While the authors' optimization of OT based approaches based on recent low rank approximation from others, resulting in considerable computational speedup and their impressive software engineering make Moscot uniformity across diverse applications, is commendable, it is important to note that alternative, potentially more efficient approaches exist, such as SIFT variants in computer vision, see for example From SIFT to PointSIFT — — Applying SIFT on Point Clouds | by Jian Shi | Medium.

The provided biological insights, though spanning a wide range of datasets, lacked depth and novelty. The demonstrations of transcription factors linked to studied biological processes (e.g., embryogenesis, liver zonation, heart and brain development) was expected, yielding minimal new insight. The attempt to use a new multi-omics dataset for pancreatic endogenesis was commendable, but the findings were not particularly significant or groundbreaking for a prestigious publication such as Nature.

My concerns extend to the OT approach's inherent limitations of ignoring underlying cellular dynamics. For instance, during a cell cycle process, OT mapping could incorrectly align two cells in an anti-clockwise orientation, even when the cell cycle flows clockwise. It might be beneficial for the authors to

illustrate Moscot's effectiveness in revealing cell cycle progression and be clear about OT's limitations in the figure 1 or supplementary figure 1. You may consider using this cell cycle dataset to demonstrate this (The transcriptome dynamics of single cells during the cell cycle | Molecular Systems Biology (embopress.org)).

In Figure 2, the computational speedup of Moscot over WOT is very valuable, but the quality of the alignment of Moscot with and without the low-rank approximation for computational speedup should also be explored. It is probably hard to directly benchmark the quality of the temporal mapping on the datasets from Figure 2 because of the lack of groundtruth. A potential evaluation scheme could involve testing the OT-based method's ability to recover alignments between the same cell using matrices of spliced RNAs or unspliced RNA. Comparisons with WOT using multi-omics datasets by mapping RNA to ATAC-seq information could also provide meaningful results. Ideally, your method should align multi-modality data from the same cell. You may also study how the performance changes as a function of the number of cells increases and when the method switches from full rank to low rank approximation.

It is interesting to see that Moscot approach yields nearly identical results to TOME, a simpler method based on KNN and gene expression similarity in Figure 2C. The conclusion drawn from this comparison should be carefully considered. This may indicate that achieving single-cell-level alignment might not be necessary or even relevant given the quality of existing scRNA-seq and spatial datasets, as cell-type level alignment may already reach the limits of what the existing data quality can offer. Similarly, this conclusion may affect the value of the spatiotemporal modeling of the heart and brain datasets, especially given the fact that the Stereo-seq dataset from this study don't really provide single cell information but binning data. Basically, do we really need the cell level alignment, can we just use TOME like method for spatiotemporal modeling given the general low quality of the existing data? I understand that your method can recover apoptosis and proliferation rates better than the scTOME approach (you extend TOME to cl-TOME). However, as stated, you also tuned the parameters in Moscot to achieve this goal. There's a lack of benchmarking on how these rates are exclusively derived from your methods. A comparison with simpler methods as outlined in the original WOT or the PRESCIENT, which directly uses the gene expression level of cell-cycle related genes, would be insightful.

Although the authors presented the spatiotemporal mapping from figure 4 as a "novel" application, this novelty is limited and appears to originate more from the data than from the Moscot tool itself. With the Stereo-seq data analyzed in this study, concerns of the unfortunate lack of correspondence between cells across timepoints or spatial slices also arise due to imbalanced and limited sampling. Some data point may also have outliers that has no correspondence to other cells. All these will lead to a partial alignment, a problem which OT-based methods may not adequately address and is extensively studied in the paste2 work (PASTE2: Partial Alignment of Multi-slice Spatially Resolved Transcriptomics Data | bioRxiv). Can

The author mentioned the Spateo (Spateo: multidimensional spatiotemporal modeling of single-cell spatial transcriptomics | bioRxiv) work (which has its own limitations) in the discussion section. Although Spateo requires downsampling, it was applied in the context of a 3D spatial transcriptomics that largely

avoid the limited and unbalanced sampling of cells and integrated with the morphometric vector field approach so that the prediction of the cell migration extends to the all cells, in addition to the original sampled cells, in a whole embryo/organ.

Furthermore, since the key advances of Moscot is its scalability. Is this really useful in complicated system (in addition to what has shown in figure 2), can for example, Moscot predict the binning coupling on the entire embryonic slices across multiple time points of the MOSTA dataset? If it is operated on the entire slice, is it still capable of predicting the evolution of each cell type? Will it actually leading to cases where a particular cell type be mapped to a different cell types ove time?

The reliance of biological insights presented in figures 4 and 5 on Cellrank makes the findings seem like an application of a previous work. I recommend that the authors attempt to reveal new biological insights directly from Moscot's results. For example, can you demonstrate directly, the spatial, temporal and spatiotemporal coupling matrices' structure can reveal novel biological insights already? Because this matrix doesn't rely on other methods which is not really a direct result from Moscot work itself. Some ideas maybe that: 1. A widely mapping spectrum of cells (in studies of figure 2/5) may implies plasticity of a cell type in the temporal mapping; 2. A narrow mapping distribution of cells (in studies of figure 3/4) may tells you something about the physical/structural constraints or the biological importance of a particular tissue domain or organ region. 3. can you identity genes that significantly affect the mapping after you remove a particular gene or a group of genes before the OT mapping?

Figure 5 on lineage relationship of delta and epsilon cells. In this section the authors generated a new datasets that aim to disentangling the lineage relationship of delta and epsilon cells. There are a few things we can consider to further enhance this section. First, my concern of OT method for mapping cell fate is that it doesn't consider dynamics and thus the mapping can be problematic. You can first fix this by including RNA velocity analyses with your scVelo tool. Second, you can also leverage your multi-omics datasets to perform multi-velo to potentially further improve your dynamics quantification. Third, a lineage tracing experiment, even the one with static barcode or a marker-gene based live cell tracking of cell lineage can be helpful.

To conclude, while the study represents a substantial effort to scale the OT method and provides valuable improvements over several existing approaches, the novelty of the methods and the biological insights derived are limited for a journal of Nature's caliber.

Minor points for consideration:

1. The use of the term "mapping cells" in the paper's title may be misleading, considering that the analyses on the MOSTA Stereo-seq study provide bins rather than cells.
2. The resolution of figure 3a/c is low, making it difficult to read.
3. The authors should demonstrate Moscot's performance under different batch conditions and its ability to handle batch effects, especially considering the high batch efforts of the MOSTA and other spatial datasets.

4. It would be valuable to see the actual aligned cell pairs for both the aligned heart and brain cells across time for the figure 4 analyses.
5. It is important to note that the reason the brain cells from E13.5 to E15.5 are not annotated is potentially that cells at these time points are less mature and thus not necessarily having the same cell identities as in E16.5. Also, this spatiotemporal mapping will suffer the same issue of the developing heart as mentioned in the major comments (limited and unbalanced sampling, partial alignment and batch effects).
6. Another potential issue of spatial and spatiotemporal mapping and alignment of Mosto in figure 3/4 is that they are done sequentially. If alignment from A/B is already problematic and B/C is additionally problematic, will that making the alignment between A/C, which is required for looking for cells across the long spatial and temporal axis, to be even more problematic?

Reviewer #1

Summary

In the paper, the authors present a new software framework called moscot, which consists of four functionalities: `moscot.time`, `moscot.space.mapping`, `moscot.space.alignment`, `moscot.spatiotemporal`. In `moscot.time`, the authors make a clever use of the Sinkhorn algorithm which vastly expands the scaling of the Waddington-OT method to big datasets, which they show in figure 2. In `moscot.space.mapping`, they implement a variant of the NovoSpaRc algorithm which allows them to map dissociated single cell data (such as scRNA) onto spatial transcriptomics data (using cost matrices in RNA space and spatial space, respectively). In `moscot.space.alignment` this algorithm is used with the aim of integrating spatial transcriptomics datasets (where the cost matrices are in spatial space for both datasets); and in `moscot.spatiotemporal` they perform trajectory inference by integrating the spatial transcriptomic datasets of the different time points (as in `moscot.space.mapping`).

The authors compare their variant of NovoSpaRc with Tangram and gimVI, and they document significant improvement in correlation between spatial correspondence and predicted gene expression (figure 3). They also show that `moscot.spatiotemporal` can be used to perform cell type annotation with significant matching with ground truth annotation precisely thanks to its use of the spatial information (figure 4). Finally, interestingly, in a multiome (scRNA+ATAC), time-resolved pancreas development dataset, `moscot.spatiotemporal` maps delta cells to epsilon cell progenitors, thereby suggesting a shared lineage, which is corroborated by several evidence provided in terms of gene markers and similarity of genome accessibility.

Overall, although the methodological development is not groundbreakingly innovative, their method has the benefits of vastly improving data scalability, accommodating multimodal datasets, and providing a common framework for any kind of optimal transport analysis which the community will find useful.

Overall evaluation: Accept with major revisions

We thank the reviewer for the constructive and insightful comments. We agree with the reviewer that besides the novel biological insights in the pancreatic endocrinogenesis study, the scalability, generality, flexibility, and extensibility of moscot will benefit the community.

Major Points

1. In Waddington-OT the underlying assumption is that development proceeds locally linearly, which indeed is inconsistent with a global optimal transport. In moscot, the authors seem to adopt the hidden assumption of such global optimal transport, for instance when they map scRNA data on spatial transcriptomic data, they assume that optimal transport is the way biology introduces batch effects. The paper needs a further experiment as follows to validate this point: take three different batches (A,B,C; say batch B is the spatial transcriptomic dataset) find the matrices $P_{\{AB\}}$, $P_{\{BC\}}$, $P_{\{AC\}}$. If the global optimal transport assumption holds then it should hold that $P_{\{AB\}} P_{\{BC\}} = P_{\{AC\}}$.

We thank the reviewer for evaluating our model assumptions. We comment below on global optimal transport and on our assumptions and validations for the Temporal and SpatialMapping problems.

Global optimal transport

We assume the reviewer alludes to global Waddington-Optimal Transport (gWOT) as introduced in ref. ⁶. gWOT represents an extension over the original Waddington-OT (WOT) algorithm which jointly solves one large convex optimisation problem, involving all time points. The authors remark that for enough samples per time point, gWOT will converge to WOT:

“In Section 5.1 we remarked that Waddington-OT is accurate when each time-point consists of a large number of observations and is a good approximation of the population. Such a scenario is not particularly interesting as the resulting performance of gWOT and Waddington-OT would be very similar. Instead, we consider subsampling each time-point in the full dataset to 100 cells per time-point in order to make a comparison between the methods in the regime of limited sampling at each time-point.” ⁶

Thus, in their real-data applications, they subsample to 100 cells per time point. Our datasets include many more cells per time point, justifying moscot.temporal’s design as a scalable and multi-modal extension of WOT (rather than gWOT).

Temporal applications

For temporal applications, we make the same assumptions as the original WOT approach⁷. Following the reviewers comment, we now state these assumptions more explicitly in the Methods section:

We follow the Waddington-OT model⁷ in assuming that cells collectively minimize the distance they travel in phenotypic space and that cellular fate decisions are Markov, i.e., cellular fate depends only on the current state and not on earlier history.

In our revised manuscript, we check that moscot.temporal reproduces WOT results for those time points of Fig. 2 that are small enough to run WOT without memory errors (see Response Fig. 3 and our reply to your minor comment 3).

These OT-assumptions have been extensively validated in the original WOT publication⁷. For example, given three time points $t_1 < t_2 < t_3$, Schiebinger et al. validated that the OT-coupling computed based on t_1 and t_3 could interpolate the cell-state distribution at t_2 , up to a level of accuracy that would be expected given batch-to-batch variation:

Fig. 2J from ref. ⁷: “Quality of interpolation in serum for OT (red), null models with growth (blue) and without growth (teal). Shaded regions indicate 1 SD. Note that OT is almost as accurate as the batch-to-batch baseline (green).”

Further, Schiebinger et al., demonstrated that their OT-couplings were robust when downsampling cells or reads, varying the initial estimates for growth-and death rates, and perturbing key model parameters (STAR Methods in ref. ⁷).

We further validate OT-assumptions in our own analysis of Fig. 2, where we recover many previously known transitions during mouse development and recover previously known driver genes. In our revised manuscript, we further show that moscot’s couplings are robust when we change latent cell representations (see Response Fig. 10 and our reply to Reviewer 2, Major comment 2), and that our computed marginals correlate well with gene-set-derived growth- and death rates (see Response Fig. 14 and our reply to Reviewer 3, Major comment 5).

Spatial applications

For the spatial mapping problem, we make the same assumptions as the original NovoSpaRc approach⁸. Following the reviewers comment, we now state these assumptions more explicitly in the Methods section:

We follow the NovoSpaRc model⁸ in assuming that cells in physical proximity tend to have similar gene expression profiles. In other words, we assume that there exists a (possibly noisy and imperfect) correspondence between physical and expression distances

These assumptions have been extensively validated in the original NovoSpaRc publication⁸. For example, Nitzan et al., validated the structural correspondence assumption in intestinal epithelium and liver lobules (their Fig 2) and in the drosophila embryo (their Fig. 3). They further corroborated OT-method assumptions by finding high correlation values between predicted and ground-truth gene expression values for intestinal epithelium, liver lobules, drosophila embryo, zebrafish embryo and mouse cerebellum datasets.

We conducted a similar analysis in our Fig. 3, where we show across 11 datasets from ref. ⁹ that the mapping accuracy of moscot.space increases with spatial correspondence:

Fig. 3b: Spatial correspondence is associated with prediction accuracy in moscot. The median Spearman correlation (spatial correspondence) is plotted on the x-axis against the median Spearman correlation across predicted gene expression for the spatial gene expression mapping task (y-axis). Each point corresponds to a different dataset and the color represents the method. The overlaid line segment is the result of a linear regression for each method.

However, even for datasets with low correspondence between physical and expression distances, we find that moscot outperforms state-of-the-art methods gimVI (Lopez et al. 2019) and Tangram (Biancalani et al. 2021). Taking the median across 14 datasets from ref. ⁹, we find higher correlation between predicted and ground-truth expression values for moscot compared to the competing methods, validating our method assumptions (Supplementary Figure 9). We note that, upon fixing a bug in our benchmarking strategy related to how Tangram processes the data, moscot's median performance across all 14 datasets is only slightly better than Tangram's.

In summary, we follow previous successful approaches to map cells across time points (moscot.time: WOT⁷ rather than gWOT⁶) or between physical and expression spaces (moscot.space.mapping: NovoSpaRc⁸). We extend these approaches to make them applicable to multi-modal atlas-scale data, and we integrate them into one consistent, user-friendly and extensible API. We inherit their method assumptions, which we now make more explicit in our methods text. Throughout our applications, we include validation experiments which support these assumptions. In addition, we now mention method

assumptions in our main text (first section of the Results) and point to the methods for further details:

We build on previous OT-based method assumptions to map cells across temporal and spatial domains (Methods).

2. Similarly, there should be a test for moscot.spatiotemporal test where we see if the optimal transport is consistent across time points, ie for time points t_1, t_2, t_3 , it should hold that $P_{\{t_1, t_2\}} P_{\{t_2, t_3\}} = P_{\{t_1, t_3\}}$.

We thank the reviewer for suggesting this additional validation. Using the known annotations, we suggest the following test for moscot.spatiotemporal to validate consistency. We obtain the couplings as described in the above above equation,

$$P^{\{1\}} = P_{\{t_1, t_2\}} P_{\{t_2, t_3\}}, P^{\{2\}} = P_{\{t_1, t_3\}}.$$

We consider the first three time points from the mouse embryogenesis spatiotemporal transcriptomic atlas (MOSTA), ($t_1=E9.5, t_2=E_{10.5}, t_3=E_{11.5}$) and compute the three couplings $P_{\{t_1, t_2\}} P_{\{t_2, t_3\}}, P_{\{t_1, t_3\}}$ using moscot.spatiotemporal. Given these mappings we obtain $P^{\{1\}}$ and $P^{\{2\}}$ as described above. First, visually comparing the mappings aggregated by bin annotations we observe good correspondence between $P^{\{1\}}$ and $P^{\{2\}}$ (Response Figure 1).

Next, to provide a quantitative assessment we consider a third naive mapping, obtained using the outer-product of the prior marginals, a, b $P^{\{3\}}= a(X) b$. To quantify the distance between annotation aggregated transport matrices we compute the Sinkhorn divergence⁽¹⁰⁾:

1. $\text{dist}(P^{\{1\}}, P^{\{2\}}) = 4.82$
2. $\text{dist}(P^{\{2\}}, P^{\{3\}}) = 6.00$
3. $\text{dist}(P^{\{1\}}, P^{\{3\}}) = 7.34$

We find that $P^{\{1\}}$ and $P^{\{2\}}$ are consistent according to this test as $\text{dist}(P^{\{1\}}, P^{\{2\}}) < \text{dist}(P^{\{2\}}, P^{\{3\}})$ and $\text{dist}(P^{\{1\}}, P^{\{2\}}) < \text{dist}(P^{\{1\}}, P^{\{3\}})$. This means that the similarity between the coupling $P_{\{t_1, t_2\}} P_{\{t_2, t_3\}}$ and $P_{\{t_1, t_3\}}$ is much larger than the similarity of $P_{\{t_1, t_2\}} P_{\{t_2, t_3\}}$ to the baseline coupling and the similarity of $P_{\{t_1, t_3\}}$ to the baseline coupling.

a

$$P^1 = P_{t_1, t_2} \cdot P_{t_2, t_3}$$

late

**b**

$$P^2 = P_{t_1, t_3}$$

late

Response Fig. 1 | Consistency analysis of moscot.spatiotempral

Cell annotation transition matrices between $t_{\{1\}}=E9.5$ and $t_{\{3\}}=E11.5$ from the MOSTA dataset as obtained by `moscot.spatiotemporal`. Transition matrices are evaluated by **a.** multiplication of consecutive time points, $P_{\{t1,t2\}} P_{\{t2,t3\}}$, for $t_2=E10.5$ or **b.** direct computation $P_{\{t1,t3\}}$.

3. In fig. 3, there needs to be a comparison of `moscot.space.alignment` with `NovoSpaRc` to test whether including many modalities has an improvement.

We thank the reviewer for the remark. We assume the reviewer is alluding to `moscot.space.mapping` as the comparison is asked with respect to `NovoSpaRc`. We agree that it is an important validation to assess how including the additional modality can improve the results of `moscot.space.mapping`. We set out to investigate this for the datasets of the mouse liver analyzed in Figure 3. However, we did not observe any clear effect when evaluating the effect of the additional modality being included in the quadratic term of `moscot.space.mapping`, as can be seen in Response Fig. 2a, b. In particular, utilizing correlation metrics between true and predicted gene expression didn't seem to provide a clear indication (Response Fig. 2a,b). We think that this stems from the fact that we are using correlation of gene expression prediction, and not protein expression (CITE-seq), to evaluate `moscot.space.mapping` performance. We chose this evaluation metric because we only have ground truth for gene expression for this dataset, and hence we can use only gene expression as our ground truth for evaluation.

To overcome this challenge, and perform a more insightful comparison on whether including an additional modality would improve the mapping, we looked for a dataset for which multimodal spatial information is available. This allows us to assess whether using both gene expression and ATAC in the quadratic term would improve the mapping compared to just using the gene expression in the quadratic term (with the linear term being constant across both configurations). We used a spatial multimodal dataset of human tonsil profiled with DBIT-seq technology by Deng et al.¹¹. This dataset consists of a single slide of human tonsil biopsy, where both ATAC-seq and gene expression is measured for 2500 spatial locations. We processed the dataset using standard Scanpy analysis, and relied on the preprocessing provided by the authors as much as possible. We then randomly split the data into two subsets, and utilized the first split as the "single cell" source dataset, and the second split as the "spatial" target dataset. To perform the mapping from the first to the second dataset, we constructed the source distribution to contain gene expression in the linear term, and **either only** gene expression in the quadratic term **or both** gene expression **and** chromatin accessibility, while we used gene expression (linear term) and spatial coordinates (quadratic term) to define the target distribution. We then mapped chromatin accessibility from the source to the target distribution. This allowed us to quantitatively evaluate the mapping by correlating the predicted chromatin accessibility with the ground truth measurements (which was also present in the target data but not utilized for computing the OT coupling). We evaluated the performance by computing average chromatin accessibility Pearson and Spearman correlation for the top 10 chromatin markers per cluster (using the clustering annotation provided by the authors, and selecting the top 10 markers with a Wilcoxon-test). Interestingly, leveraging both chromatin accessibility and gene expression provide a more

accurate and robust mapping for various hyperparameters of moscot, such as the entropy regularization term *epsilon* and unbalancedness parameter *tau* (Response Fig. 2c-d). Furthermore, the performance is consistently higher than the one achieved by Tangram, which we use as a baseline for this task. We decided to not use NovoSparc since it is a non-scalable, slower and less flexible version of moscot.space.mapping. Overall, our analysis suggests that leveraging additional modalities can be useful when the mapping task of interest is to project modalities present only in the source single cell data to the target spatial dataset. We have added the following statement to the main text:

To quantitatively assess whether including multiple modalities into moscot.space.mapping improves the performance, we imputed ATAC-seq on a spatial multiome dataset of human tonsils, consisting of the joint profiling of spatially resolved ATAC-seq and RNA-seq. We found that including the ATAC-seq information results in improved performance (Supplementary Fig. 12).

Response Fig. 2 | **moscot.space.mapping improves performance when including additional modality** -> included in our manuscript as **Supplementary Fig. 12**

a. Median Spearman correlation of all genes in the mouse liver dataset (Figure.3) using only RNA-seq or CITE-seq + RNA-seq information for the mapping. Box plots represent at least 3 runs with different initializers and ranks. **b.** Same as a), but measured with Pearson correlation. **c.** Median Spearman correlation for the 70 chromatin accessibility peaks (10 for each cluster, 7 clusters) using RNA-seq or RNA-seq + ATAC-seq for the spatial multiomics dataset. Each point is a different value of alpha (weighing factor for the linear and quadratic term). **d.** Same as c., but median Pearson correlations.

Minor Points

1. It is not clear how the a marginal allows for cell birth. Specifically since $0 \leq a_i \leq 1$, it means the $\sum_i P_{ij} = a_i = 1$ so the model could only accommodate loss of cells. More explanation is needed in the text.

We thank the reviewer for bringing this point to our attention. For N and M early and late cells, respectively, balanced optimal transport could distribute $1/N$ probability mass per early cell. Thus, values greater or smaller than $1/N$ correspond to cellular growth or death, respectively. We now clarify this in the Methods text:

We estimate growth- and death rates from curated marker gene sets; moscot comes with pre-defined gene sets for mice and humans. Intuitively, our adjustment allows t_1 -cells likely to proliferate or die to distribute more or less probability mass towards t_2 -cells, respectively. In the absence of cellular growth and death, every t_1 cell would be allowed to distribute $1/N$ probability mass; thus, values greater or smaller than $1/N$ indicate proliferation or apoptosis, respectively.

2. In supplementary note 5, it is unclear how the Gibbs kernel which has a parameter ϵ is mapped to the heat kernel with parameter t , so the authors need to elaborate on this.

We thank the reviewer for the comment, we have adapted this in the Supplementary Note 5. With respect to the relation between the Gibbs kernel and the heat kernel, we added:

In particular, [Crane et al., Huguet et al.] choose $t = \frac{\epsilon}{4}$ with ϵ denoting the entropy regularization parameter, which allows using the heat kernel \mathcal{H}_t as a drop-in replacement for the usual Gibbs kernel (Methods), i.e.,

$$K(x, y) = \mathcal{H}_t(x, y)$$

for cells x and y , and heat diffusion parameter t .

3. In fig.2 there are only comparisons with TOME, not with Waddington-OT. It would be useful to have a plot showing that the differences between Waddington-OT and moscot.time are negligible.

We agree with the reviewer that readers will benefit from a direct comparison between moscot.time and Waddington-OT (WOT). We include such additional analysis in our revised manuscript and in Response Fig. 3.

We now compare moscot.time and WOT across the pre-gastrulation and gastrulation stages, where time points contain few enough cells for us to run WOT. Our additional analysis shows that moscot.time and WOT yield very similar results in all of the following metrics: germ-layer and curated transition metrics of Main Figure 2 (Response Fig. 3a,b), correlations on the single-cell transition level (Response Fig. 3c,d) and correlations on the cell-type transition level (Response Fig. 3e-g). Accordingly, we added the following sentence to our main text:

For the smaller time points corresponding to pre-gastrulation and gastrulation, where we could run the original WOT approach, we verified that moscot.time's results were consistent with WOT's (Supplementary Fig. 4).

Response Fig. 3 | **Moscot reproduces Waddington OT results -> included in our manuscript as Supplementary Fig. 4**

Moscot.time versus Waddington OT (WOT) in terms of our benchmark metrics (a,b), and on the level of single-cell (c,d) and cell-type (e-g) transitions. Throughout this figure, we show earlier time points (Pre-gastrulation and Gastrulation) containing few enough cells for WOT to run (Methods). a,b. Comparing moscot.time and Waddington OT (WOT) in terms of the germ-layer and curated transition metrics of Fig. 2, for individual time points (a) and aggregated over time-windows (b). Accuracy is weighted by the number of cell states per time point (Methods). c. Pearson correlation (y-axis) of moscot.time with WOT normalized coupling matrix entries as a function of time (x-axis; Methods). d. Scatter plots, illustrating the four exemplary time point pairs indicated in (c) with circles. e. As in (c), but aggregated to the cell-type level. We omit the first pair of time points, as they only contain a single cell state. f. Heatmaps of moscot.time (left) and WOT (right) backwards cell-type transitions for

the E7.0/7.25 time point pair, indicated in (e) with a circle. Row and column colors correspond to (g). g. UMAP of the E7.0/7.25 time point, colored by cell types.

4. In methods section 1.3.1 the authors mention that the definition of C^X will follow below. In the following section (below equation 15), they allude to a definition of C^X made “above”, but the definition is nowhere in between. The definition needs to be clearly stated upon first mention of C^X .

In Methods Section 1.3.1, we describe the spatial mapping problem and define the cost matrices C^X and C^Y , corresponding to molecular distances in the dissociated reference and physical distances in the spatial sample, respectively. Specifically, we write in the text:

In addition, let $C^Y \in \mathbb{R}^{M \times M}$ encode spatial similarity among the M samples in Y (we define C^X below).

In the following section, below Equation 15, we define C^X in the following sentence:

[EQUATION 15],

for spatial distance matrix $C^Y \in \mathbb{R}^{M \times M}$, defined as above, **and reference distance matrix $C^X \in \mathbb{R}^{N \times N}$, quantifying molecular similarity among cells in the dissociated reference.** To compute C^X , we measure expression distance among reference cells in a latent space defined using PCA or scVI^[Lopez2018deep].

The text below equation 15 alludes to a definition of C^Y made earlier, not C^X . We conclude that our methods text contains definitions of both matrices.

Reviewer #2

Summary

Klein et al. have developed an integrative and scalable method called "moscot" that utilizes optimal transport (OT) for analyzing multimodal single-cell data. This framework demonstrates great potential for analyzing complex single-cell genomics datasets, including spatial and temporal single-cell data. The authors showcase the feasibility of moscot through the analysis of various published datasets, such as mouse embryogenesis scRNA-seq, spatial liver MERSCOPE data, and a time-resolved spatial transcriptomics dataset of mouse embryogenesis. Additionally, they generate paired scRNA-seq and scATAC-seq data from Ngn3+ cells during mouse pancreatic development and predict the lineage hierarchy of delta and epsilon cells.

Overall, moscot presents a promising tool for single cell genomic analysis. Optimal transport is certainly the right theoretical framework for solving these mapping and alignment problems, and the moscot framework introduces a scalable implementation which can be integrated into existing workflows. However, the biology prediction is weak without experiment validation. Some major concerns need to be addressed:

We thank the reviewer for the constructive comments, and in particular the points on the experimental validation. We give a detailed point-by-point response below, in particular highlighting the additional experimental validations we have added in the revision.

Major points

1. In the pancreatic development part, the limited time points (E14.5 and E15.5) covered in this study may overlook other potential differentiations and the prediction is not experimentally validated. Ngn3's expression can be detected from E12.5 (Byrnes et al, 2018). Different endocrine cells including delta and epsilon cells are likely to fully differentiate at late development stage (Yu et al, 2021). However, this study only covers two development time points in the middle. It is very possible to miss other differentiation possibilities and the data can be misinterpreted. Another thing is EPs are identified only by Ngn3, possibly neglecting important distinctions between EPs. Previous lineage tracing experiments have indicated that islet cell fate is determined before hormone expression (Herrera et al, 1998 & 2000). However, when EPs diverge to differentiate into specific islet cell types is not known, therefore whether this decision occurs before, during, or after Ngn3 expression is unknown. Therefore, samples from more time points are necessary and experiment validation should be done.

We thank the reviewer for raising these important aspects. Indeed, we agree it is important to consider additional time points. Hence, we generated new samples for E15.5 and E16.5, and analyzed E12.5/E13.5 of external validation datasets¹². Moreover, we performed a gene

knockout experiment of NEUROD2 in human iPSCs, providing evidence of NEUROD2 being a regulator of the epsilon lineage (Response Fig. 4).

As a result, we specified our hypothesis to the claim that progenitors of delta and epsilon cells follow a similar developmental trajectory. Below you will see that we corroborate this claim based on further analysis of our dataset. Moreover, we adapted the section *Disentangling lineage relationships of delta and epsilon cells in pancreatic development leveraging multiple modalities with moscot* and updated supplementary figures 22, 23, 24, 27, 28, 29, 30, 33, 34, 35, 39, 40, 43, 44, Supplementary Notes 7 and 8, as well as Supplementary Tables 11, 12, 13, 14, 15, 16, 17, 18, 19, 20, 21, 22.

In the following, we discuss all aspects of major point 1 in multiple paragraphs.

Emergence of endocrine cells before E14.5

Byrnes et al.¹³ as well as Bastidas-Ponce et al.¹² indeed report that endocrine differentiation can be observed at E12.5. Yet, Byrnes et al find no delta cells at E12.5 and only very few epsilon cells at E12.5 (Supplementary figure 5, Byrnes et al.). This finding aligns well with the data published in Bastidas-Ponce et al.¹², the cell type distribution of which we added in Response Fig. 5e. Yet, Response Fig. 5 shows that a low number of delta and epsilon cells, as well as their hypothesized ancestors, can be observed at E13.5. Hence, we analyzed the trajectories of the developing mouse pancreas using moscot for time points E13.5 and E14.5. While the results have to be considered with care due to the relatively low number of cells, the results corroborate our hypotheses of lineage dependencies of delta and epsilon cells as well as their progenitors (Response Fig. 6).

Emergence of endocrine cells after E15.5/ Samples from more time points

Yu et al.¹⁴ report that endocrine cells also develop at later stages than E15.5. In particular, they state “Flow cytometry analyses revealed that few GHRL+, SST+ and PPY+ cells originated at E13.5, E14.5 and E16.5, respectively [...]” To study the origin of endocrine cells with a specific focus on delta and epsilon cells, we generated a new dataset capturing cells at E15.5 and E16.5 following the same protocol as the initial one capturing E14.5 and E15.5 (we describe the experimental details in the methods, section 2.1.2 Application: moscot.time on multi-model pancreas development, paragraph *Dataset generation*). As mentioned above, we updated the analyses and corresponding figures to incorporate the new samples.

Definition of EPs

We would like to point out that we indeed distinguish between different stages of endocrine progenitors. In particular, we define the clusters “Ngn3 low”, “Ngn3 high”, “Ngn3 high cycling”, “Fev+”, “Fev+ Alpha”, “Fev+ Beta”, “Fev+ Delta” (which we further subcluster into “Fev+ Delta,0” and “Fev+ Delta,1”), and “Epsilon progenitors”.

Experimental validation

We identified driver genes for the Epsilon progenitor population and the Fev+ Delta population using moscot, and found *Neurod2* to be the second most relevant one (Supplementary Tables 19, 20). As we had found the Fev+ delta and the epsilon progenitor population to be similar in gene expression and ATAC profile (Response Fig. 7, Supplementary Fig. 34, 35, 36, 37), the computational recovery of similar sets of

transcription factors (TFs) is not surprising. To distinguish the regulation between the epsilon and delta lineage, we took advantage of a NEUROD2 knock-out and control human iPSC line that were recently generated by us¹⁵ to analyze the possible impact of NEUROD2 loss on epsilon cell development. Differentiating the generated clones towards stem cell-derived islets (SC-islets) resulted in reduced levels of GHRL (the main hormone produced by epsilon cells) mRNA assessed by qPCR analysis. Additionally, we supported this data by immunostaining and confocal microscopy that revealed a significant reduction in the number of GHRL-positive cells in two independent NEUROD2 KO clones when compared to wild-type control (Response Fig. 8). However, we didn't find any impact of loss of NEUROD2 on delta cell formation assessed by qPCR analysis and confocal microscopy. Moreover, we didn't find any changes in the beta and alpha cells similar to our previous report (Supplementary data).

Hence, we found the NEUROD2 transcription factor to specifically regulate pancreatic epsilon cell formation that likely functions in an evolutionary conserved manner.

Response Fig. 4 | **moscot disentangles lineage ancestries of delta and epsilon cells** -> included in our manuscript as Fig. 5

a. Schematic of the experimental protocol to generate paired gene expression and ATAC data capturing the development of the mouse pancreas. **b.-c.** Multimodal UMAP^{16,17} embedding, colored by time (b) and cell type annotation (c) (Methods). **d.** Heatmap visualizing descendency probabilities of cell types in E14.5 as obtained by moscot.time. **e.** UMAP embedding colored as in (c), including the refined *Fev+* delta populations. The inset highlights the cells for which we compute a PHATE embedding¹⁸. The middle row shows epsilon cells at E16.5 (left) as well as the progenitor population at E15.5 (middle) and E14.5 (right) as predicted by moscot. The bottom row shows the corresponding plots for delta cells. **f.** Sankey diagram of the cell type transitions between E14.5 and E15.5 (top) and E15.5 and

E16.5 (bottom). **g.** Similarity in ATAC profile between different cell types (Methods). The green boxes highlight delta and epsilon cells and their putative progenitors. **h.** Representative confocal images and quantification of GHRL-expressing cells in the control and NEUROD2 knockout (C37, C89) stem cell-derived islets at stage 6, day 14 (S6.14, Methods). White arrowheads; GHRL+ cells. Scale bar, 50 μm . $n=4$ independent experiments. **i.** qPCR analysis of expression levels of GHRL at S6.14. Data are represented as mean and standard error.

Response Fig. 5 | **Overview of the pancreatic endocrinogenesis dataset published by Bastidas-Ponce et al.**¹² -> **included in our manuscript as Supplementary Fig. 32**

a. UMAP embedding colored by cell type of the pancreatic endocrinogenesis dataset published by Bastidas-Ponce et al.¹². The data was subset to endocrine cells and their progenitors. Subsequently, the data was preprocessed by normalization, log_{1p}-transformation and PCA computation before calculation of the neighborhood graph, based on which the UMAP was computed. **b.** UMAP embedding colored by cell type. **c.** Processed gene expression of *Fev* on the UMAP, visually suggesting there is no expression of *Fev* in the population annotated as *Fev*⁺ epsilon. **d.** Quantitative confirmation that there is barely any expression of *Fev* in the cell type annotated as *Fev*⁺ epsilon in Bastidas-Ponce et al.

Response Fig. 6 | **Moscot analysis of E13.5 and E14.5 of pancreatic endocrinogenesis dataset is consistent with the analysis on the novel multiome dataset -> included in our manuscript as Supplementary Fig. 33**

a. UMAP embedding colored by cell type of the pancreatic endocrinogenesis dataset published by Bastidas-Ponce et al.¹² filtered to embryonic day 13.5 and 14.5. The data was subset to endocrine cells and their progenitors. **b.** UMAP embedding colored by cell type annotated as provided by the authors. The *Fev*⁺ epsilon population is similar to the epsilon progenitor population defined in this work, see Supplementary Fig. 32. **c.** Probability of a cell type in time point E14.5 to originate from a cell type in E13.5 calculated by moscot. Highlighted are the predicted cell type origins of delta and epsilon cells **d.** Visualization of ancestry likelihood of single cells predicted by moscot for epsilon, *Fev*⁺ epsilon, delta, and *Fev*⁺ delta cells.

a *Neurod2* target gene activity *Neurod2* target enhancer activity

c Target genes of *Neurod2* per cell type

Response Fig. 7 | **Regulatory analysis of *Neurod2* identifies target genes -> included in our manuscript as Supplementary Fig. 42**

a. Expression of predicted target genes of *Neurod2* and accessibility of predicted enhancers of *Neurod2* based on the eRegulon computed with Scenic+¹⁹. **b.** Expression levels of a subset of *Neurod2* target genes (predicted by Scenic+) in different endocrine clusters (scaled heatmap, Supplementary Tables 23-28). **c.** Expression of target genes of *Neurod2* in cell types of interest, with thresholds provided by Scenic+ (Supplementary Table 29).

Response Fig. 8 | **NEUROD2 knockout experiments in human iPSCs-derived islet cells**
 -> included in our manuscript as Supplementary Fig. 43

a. Insulin mean intensity and the number of SST-positive cells measured with immunostaining (Methods) for control, clone 37 and clone 89¹⁵. b. Relative mRNA expression of *Ins2*, *SSt*, *HHEX*, and *GCG* measured with qPCR. We report mean and standard errors.

2. How dependent are these results on existing high quality latent representations? For example, in supp methods sec 2.1.1, analysis seems dependent on proper batch correction done by Seurat's Find Integration Anchors function which can already take some fine-tuning to get a desirable result. Or in section 2.1.2, many specific steps must be taken to properly embed each modality separately (e.g remove 1st and 5th PCs, discarding ambient genes, etc). Or in analysis of the pancreatic development data, many different filtering criteria had to be applied to each timepoint with varying (likely carefully tuned) parameters. This is sometimes glossed over in single-cell projects as a major time sink and source of variability in results so I wonder how robust moscot is to variations in the initial latent representations? Especially when the geodesic cost functions depend on the knn graph produced from these representations.

We thank the reviewer for pointing us to the question of latent cell representations for OT-cost computations, which we investigate in more detail in our revised manuscript.

Latent space robustness analysis

For our moscot.time embryogenesis data example, described in Section 2.1.1 in the Methods, we originally used Seurat's Find Integration Anchors-based embedding to enable a fair comparison with the competing TOME method, which used this embedding in their original publication²⁰. In our revised manuscript, we include comparisons of moscot.time with TOME based on PCA and scVI-latent space representations. We include the additional analysis in our manuscript in a new Supplementary Figure 3 and in this letter in Response Fig. 10.

Using both the germ-layer and curated transition metrics of Main Figure 2, we find that moscot.time's performance is robust with respect to changes in latent cell representations, while TOME's performance degrades and appears to depend on the batch integration of the original embedding (Response Fig. 10a,b). An extreme example of TOME's decreased performance is the case of E8.5a/b samples in a scVI embedding, where TOME maps almost all E8.5b cell states to E8.5a extraembryonic visceral endoderm cells (Response Fig. 10c). Both samples correspond to the same time point, profiled using 10x Genomics on single-cells (E8.5a) and sci-RNA-seq3 on single nuclei (E8.5b), introduced in the original TOME publication to bridge a early-stage single-cell with late-stage single-nuclei time course datasets²⁰.

We think that there are two reasons for TOME's decreased performance in this setting:

1. TOME uses a **3D UMAP representation** to link late cells to early cells. However, E8.5a extraembryonic visceral endoderm cells are outliers in this representation, causing TOME to match them with almost all E8.5b cells. We speculate that in the original Seurat embedding, batch effect correction mitigated this issue.
2. TOME uses nearest-neighbor matching and **does not conserve probability mass**. In the current example, not conserving probability mass means that TOME is allowed

to match a single cell state in one sample to almost all states in the other sample. Such boundary effects are known problems of kNN-based matchings, see e.g., Fig. 1 in the recent CINEMA-OT paper ²¹.

Moscot.time avoids these pitfalls by computing cell-cell distances in higher dimensional spaces, which capture more biological variability, and by weakly enforcing probability mass conservation while allowing some deviations to account for cellular growth and death. We added the following description to our main text:

Moscot.time's performance was more robust than TOME's when we varied the underlying embedding used to compute cell-cell distances (Supplementary Fig. 3).

Pre-processing of the pancreatic endocrinogenesis dataset

Concerning the preprocessing of our pancreas data example, described in Section 2.1.2 in the Methods, we followed best practices for single-cell data analysis ²². For example, we removed the 1st and 5th principal components due to their high correlation with the number of peaks, a common practice in preprocessing scATAC data ²³.

Hyperparameter robustness in the latent space in the pancreatic endocrinogenesis dataset

To demonstrate the stability of the OT predictions, we run moscot with different hyperparameters, including the choice of latent embedding (incorporating only one modality, i.e. PCA space of gene expression, VAE embedding of gene expression (scVI space ²⁴), LSI space of ATAC, VAE embedding of Peaks (PoissonATAC space ²⁵), as well as incorporating both modalities by concatenation of the spaces (PCA/LSI, PCA/PoissonVI, scVI/LSI, scVI/PoissonVI) as well as the MultiVI²⁶), and the choice of cost (squared Euclidean, cosine cost, geodesics from heat diffusion with different number of neighbors k in [5, 10, 30, 50, 100]). This results in 63 different configurations of hyperparameters of the moscot model. We measure the stability with two metrics. We globally analyze the stability of the transport matrix by computing the Sinkhorn divergence ¹⁰ between the transport plan aggregated to cell type level and the aggregated transport plan of the moscot predictions we use to analyze the pancreas dataset throughout the rest of the manuscript. Second, we assess the stability with respect to the research question of the origin of delta and epsilon cells by measuring the differences in the probability of certain cell type transitions. We also report the transitions from Fev+ Beta to Beta as a which is biologically confirmed. We compare the mean (and standard deviation) across all 63 configurations with the predictions of the outer coupling. We can see that the mean Sinkhorn divergence is much lower than the Sinkhorn divergence between the reference coupling and the outer coupling, meaning all couplings are close to the reference coupling, which has been used throughout the manuscript. Similarly, we report much higher cell type to cell type transitions than those obtained with the outer coupling. We added a section to the methods in section 2.1.2 (Application: moscot.time on multi-modal pancreas development) in paragraph *Studying the influence of the embedding and the cost*, where we also discuss technical details.

Response Fig. 9 a) shows that we obtain the hypothesized transitions independent of the cost and the embedding, while Response Fig. 9 b) shows that all moscot couplings are similar, but not completely independent of the embedding and the choice of the cost. Response Fig. 9 c) focuses on one transition which differs significantly with respect to the

cost function (but relatively independent of the embedding): Choosing the squared Euclidean cost or the cosine cost results in biologically implausible results of proportions of delta cells being directly derived from Ngn3 low cells. Choosing the geodesic cost (with different hyperparameters) prevents this. Hence, we chose the geodesic cost for analyzing the pancreatic endocrinogenesis dataset. Accordingly, we added the following paragraph to the main text:

Additionally, we studied the influence of the cost and embeddings, revealing the necessity of using geodesic costs while being robust with respect to the embedding (Supplementary Fig. 26, Methods).

Response Fig. 9 | **Stability of coupling of moscot.time with respect to different embeddings, costs and parameters of the cost. -> included in our manuscript as Supplementary Fig. 26**

Different statistics derived from the cell type-aggregated transport matrix across different configurations of the embedding, cost, and hyperparameters of the cost (Methods).

a. Selected cell type transitions of moscot run with different configurations for the E14.5/15.5 coupling and the E15.5/16.5 coupling. The baseline corresponds to the independent coupling. The higher the cell type transition probabilities, the more signal the coupling captures. **b.** Sinkhorn divergence between the aggregated transport matrix of the reference configuration (used in the analysis, Methods) and the aggregated transport matrix obtained from different configurations of the embedding, cost, and parameters of the cost. The baseline is the Sinkhorn divergence between the reference configuration and the outer coupling. The lower the Sinkhorn divergence, the more similar to the reference coupling. **c.** Proportion of delta cells which are predicted to be derived directly from Ngn3 low for different embeddings, costs and specifications of the cost (Methods). As a direct transition from Ngn3 low to delta is very unlikely, the lower the score, the better. In all plots, mean and standard error are reported.

Response Fig. 10 | **Moscot is robust to changes in latent cell space representation** -> included in our manuscript as **Supplementary Fig. 3**

a,b. Comparing moscot.time and TOME in terms of the germ-layer and curated transition metrics of Fig. 2, for individual time points (**a**) and aggregated over time-windows (**b**), for the original Seurat embedding of ref.²⁰ (top row), a PCA embedding (middle row) and an scVI embedding (bottom row; Methods). **c.** Heatmaps of cell-type backwards transition probabilities for moscot (left) and TOME (right) on the E8.5a/b pair of time points where TOME performs worse than moscot in PCA and scVI representations. The letters a/b correspond to different experimental technologies employed at the same time point as a “bridge” (Methods). Colors correspond to clusters in (**d**). **d.** 3D UMAP embedding, used by the TOME algorithm for k-NN graph construction²⁰, colored by time points (left) and cell types (right). Circle and arrow highlight a cluster of E8.5a extraembryonic visceral endoderm cells, which TOME falsely connects to almost all E8.5b cells because it is an outlier in the 3D UMAP. Moscot avoids such boundary effects by using higher-dimensional representations and by weakly enforcing probability mass conservation (Methods).

Minor points

1. In first paragraph of results section (pg 4) "moscot takes as input unpaired datasets". But in moscot supp methods sec 1.2.3 "this naturally extends beyond two modalities towards any number of jointly measured modalities"

We thank the reviewer for bringing this point to our attention. While we assume the datasets X and Y to be unpaired, i.e. measured for different cells, each of them may contain several jointly measured modalities. For example, in the temporal context, X and Y correspond to early and late time points, respectively, and each of them might contain jointly measured gene expression and chromatin accessibility values. In the spatial mapping context, X and Y correspond to a dissociated single-cell dataset and a spatial sample, respectively, and each of them might contain several jointly measured modalities. We agree that the old sentence describing this setup was not fully clear and we adapted it to read:

Moscot takes as input unpaired datasets, which can represent different time points or spatial representations, each containing one or more molecular modalities (Methods and Fig. 1a).

2. Somewhat unclear how suitable moscot is for diagonal integration (i.e unpaired datasets--different cells, different features)

We believe that minor point 2 is closely related to minor point 3, and hence we will respond to minor points 2 and 3 jointly below.

3. There are some examples of diagonal integration (e.g the pancreatic development dataset) but does this differ from how moscot is applied to paired data?

While our manuscript focuses on the extension of existing and presentation of novel optimal transport applications in single-cell genomics in spatial, temporal and spatiotemporal datasets, we agree with the reviewer that optimal transport can also be used for diagonal integration, as presented in SCOT²⁷. Indeed, extensions of SCOT (e.g. to the low-rank setting, which makes SCOT scalable) are implemented in moscot, and corresponding

tutorials can be found on moscot-tools.org:

https://moscot.readthedocs.io/en/latest/notebooks/tutorials/600_tutorial_translation.html.

Moreover, we would like to point out that the pancreatic endocrinogenesis dataset includes paired measurements of RNA and ATAC, hence no integration across modalities has to be performed. We believe this property of the dataset to be particularly valuable as we do not have to rely on integration performance for the analysis of the pancreatic endocrinogenesis dataset.

4. When adjusting the marginals for growth and death (supp methods pg 4) are the apoptosis scores and proliferation scores computed from known markers? How would moscot behave if these marginals were not adjusted?

We describe the marginal adjustment in our Methods text (Section 1.2.2). In particular, we write:

We estimate growth- and death rates from **curated marker gene sets**; `\moscot\` comes with pre-defined gene sets for mice and humans.

In addition to these pre-defined marker gene sets, the user can supply their own gene sets. If no markers are provided, the user is advised to use a higher notion of unbalancedness, which automatically adjusts the marginals (Methods; Section 1.2.2). When we compare `moscot.time` with the TOME algorithm in terms of cellular growth and death rates on the mouse embryogenesis data of Figure 2, we do not adjust the marginals to enable a fair comparison with TOME, which does not include a feature to adjust marginals. We describe this in the Methods, Section 2.1.1:

For `\moscotTime\`, we did not initialize the growth rates using marker genes to enable a fair comparison with `cl-TOME`, which does not support such initialization.

In our revised manuscript, we include a more extensive analysis of cellular growth and death rates predicted by `moscot.time` and TOME on an additional dataset⁷, see our response to Reviewer 3, Major comment 5. Importantly, we did not adjust `moscot`'s marginals for cellular growth and death in this new analysis; nevertheless, predicted growth rates correlated well with scanpy-computed cell cycle scores (Response Fig. 14).

5. Might want to include explanations/intuition for what push and pull operations are in OT and how they can be interpreted in the context of cell state transitions. Specifically in supp methods 1.2.4 I found it unclear why correlating pull and push distributions with gene expression will uncover putative driver genes

We thank the reviewer for this remark and added the following paragraph to our Methods (Section 1.2.4) to better explain push and pull operations in the context of cell state transitions and our correlative strategy to pinpoint putative driver genes:

In summary, we use pull and push operations based on our computed transport matrix P to recover putative ancestors and descendants of cell states of interest, respectively. In biological terms, for a given t_1 cell state \mathcal{P} , we interpret its push distribution over

t_2 cells as the likelihood of \mathcal{P} giving rise to these cells. Analogously, for a given t_2 cell state \mathcal{Q} , we interpret its pull distribution over t_1 cells as the likelihood of these cells to give rise to \mathcal{Q} . Accordingly, we correlate gene expression with the pull-back distribution to pinpoint putative driver genes of transitioning into state \mathcal{Q} . Using positive and negative correlations, such a strategy will reveal consistently up or down-regulated genes in cells likely to transition to state \mathcal{Q} , respectively.

In addition, we would like to point out that the terms pull(-back) and push(-forward) distribution are commonly used terms in probability theory, and hence we refer to text books for a mathematical definition, see e.g. Peyre and Cuturi²⁸.

6. What does it mean in supp methods sec 1.3.3 that this "is the first method to make use of multimodal information in the actual mapping problem"? Is this referring to jointly learning a multi-modal latent representation?

We thank the reviewer for bringing this point to our attention. Throughout the manuscript, we refer to the mapping problem as the problem of assigning single cell gene expression profiles to spatial gene expression profiles. For all the methods benchmarked, and for moscot, this is achieved by computing some type of similarity between the gene expression profiles in dissociated cells and spatial coordinates and thus deriving a form of (probabilistic) assignment matrix, which is then used for mapping.

Considering a scenario in which we have jointly profiled ATAC and RNA information in a dissociated reference dataset X and spatial gene expression in dataset Y, competing methods like Tangram²⁹ learn a mapping based on gene expression in X and Y and subsequently apply that mapping to the additional ATAC modality in Y. While this allows the method to transfer all modalities into space, it does not make use of the additional information of chromatin accessibility when learning the actual mapping. While we believe that moscot was the first method to make use of multimodal information in the actual mapping process when we first submitted the manuscript, there may be other spatial mapping techniques by now with the same property. Thus, we changed the statement in the Methods (Section 1.3.3) to read:

While previous methods could apply a mapping learned from gene expression data to other modalities collected for the same cells\cite{biancalani2021deep}, \code{moscotSpaceMapping} is **among the first methods** to make use of multimodal information in the actual mapping problem.

We added another sentence to better explain this statement:

In other words, our approach uses multimodal information when learning the mapping, rather than learning the mapping based on uni-modal data and subsequently applying it to jointly captured modalities.

Reviewer #3

Summary

In this submitted manuscript, Dominik and others have made commendable strides in creating Moscot, a general-purpose toolkit for analyzing single cell and spatial transcriptomics datasets. This represents a significant engineering effort and is poised to contribute meaningfully to the genomics field building upon their scverse ecosystem. I appreciate the study's timeliness and potential.

We thank the reviewer for careful discussion and comments, which considerably helped clarify moscot's impact vs other methods and biological interpretation.

I would like to present my review in the following first addressing general comments and then discussing specific aspects of the study.

Major points

1. One of my principal reservations regarding the manuscript is its lack of novelty and significant biological insights, especially considering the high standards of a Nature submission. Concerning the computational advancements, the use of optimal transport (OT)-based methods for single cell and spatial transcriptomics is not new but multiple studies over the past a few years (e.g., WOT, COMMOT, PASTE2) has reported similar efforts (although in more isolated manner). While the authors' optimization of OT based approaches based on recent low rank approximation from others, resulting in considerable computational speedup and their impressive software engineering make Moscot uniformity across diverse applications, is commendable, it is important to note that alternative, potentially more efficient approaches exist, such as SIFT variants in computer vision, see for example From SIFT to PointSIFT — — Applying SIFT on Point Clouds | by Jian Shi | Medium.

We thank the reviewer for this remark. We have considered the algorithm suggested by the reviewer (referring to the original academic publication from ³⁰) but we were not able to run the original implementation (<https://github.com/MVIG-SJTU/pointSIFT>) after repeated tries by multiple researchers due to significant issues in the installation, reported by many other

users during the year (the last update in the repository is 6 years old and the code relies on TensorFlow 1.4, which represents a significant challenge to run with modern GPUs). Furthermore, we would also like to specify that the task of the suggested method is not point cloud registration but of point cloud segmentation, which is reported also in the title of the academic article from which the blog post takes inspiration from "PointSIFT: A SIFT-like Network Module for 3D Point Cloud Semantic Segmentation". In fact, all the experiments reported in the original publication reports *accuracy* and *mean Intersection over Union (meanIoU)* which consists of metrics typically used to evaluate (semantic) segmentation algorithms.

Nevertheless, we agreed on the reviewer to benchmark moscot against competing algorithms that are not specifically designed for the challenges in the biomedical domain (and specifically spatial transcriptomics). We therefore looked for alternatives that were able to handle specifically 2D Point Clouds registration (in contrast to the majority of modern algorithms in the computer vision domain that consists of registration methods for 3D points clouds or meshes). We found the pointSIFT-related Deformable-Coherent Point Drift (D-CPD), which is suitable to be applied to this data³¹ to be suitable for the problem, and tested it on the synthetic dataset presented in the main text. Moscot with default values outperforms D-CPD, and we also notice that with a lower alpha value (that is, with more strength given to the linear term that evaluates gene expression similarity) moscot slightly improves performance Response Fig. 11. We hypothesize that the reason for such improvement is due to the fact that moscot can directly leverage the gene expression similarity, and hence matching more accurately the source spots to the target spots. We have added the following statement to the main text:

[...] as well as with other registration methods not specific to spatial omics data.

Response Fig. 11 | **moscot's spatial alignment method is superior to alignment with deformable-coherent point drift.** -> included in our manuscript as **Supplementary Fig. 14**

a. Benchmark across 8 synthetic datasets for moscot-default (default parameters of moscot), d-cpd³¹ (default parameters) and moscot-alpha=0.2 (where the alpha parameter is set to 0.2). **b.** Example of the original alignment problem for the two set of points clouds (source in orange, target in blue), and the result for d-cpd and moscot-default

2. The provided biological insights, though spanning a wide range of datasets, lacked depth and novelty. The demonstrations of transcription factors linked to studied biological processes (e.g., embryogenesis, liver zonation, heart and brain development) was expected, yielding minimal new insight. The attempt to use a new multi-omics dataset for pancreatic endogenesis was commendable, but the findings were not particularly significant or groundbreaking for a prestigious publication such as Nature.

We thank the reviewer for the comment but we respectfully disagree with respect to the depth and novelty of the findings. First, we would like to once again highlight that *current OT implementations are not usable* for the majority of single-cell datasets due to time and memory complexity (and hence are indeed not used as much as one would expect, given the explosion of recent time series/spatial data sets). Second, we would like to highlight that for most modalities other than gene expression, there exists a *very limited range of analysis and integration methods*, while we demonstrate that we can use optimal transport in single-cell genomics for a wide range of tasks and modalities.

Third, we would like to emphasize that the findings with respect to the pancreatic endocrinogenesis dataset are important for a better understanding of the developmental system, and how it might be dysregulated in disease. In particular, elucidating signal and factors regulating pancreatic endocrine lineage formation will not only help to bioengineer islet clusters from pluripotent stem cells in vitro, but also allows to decode the pathomechanisms of monogenic forms of diabetes by iPSC disease modeling. As we report the first experimentally validated transcription factor for activating specifically epsilon lineage

formation, we are convinced to contribute substantially to the understanding of the development of the murine pancreas.

However, we agree that in the first version of the manuscript, we have clearly not sufficiently communicated these points. We have now significantly extended our biological analysis on the pancreas dataset (Figure 5, Supplementary Figures 22, 23, 24, 27, 28, 29, 30, 33, 34, 35, 39, 40, 43, 44, Supplementary Notes 7 and 8, as well as Supplementary Tables 11, 12, 13, 14, 15, 16, 17, 18, 19, 20, 21, 22, e.g. including regulatory analyses of *Neurod2*) and on the MOSTA dataset (e.g. by identifying target genes of the transcription factor *Hnf4a* (Supplementary Table 6)). We also added analyses on datasets which we hadn't included in the first version of the manuscript, e.g. the multiome spatial transcriptomics dataset of human tonsils (Supplementary Figure 12), and the 3D spatiotemporal dataset of the drosophila embryo (Supplementary Figure 22). In addition, we have added novel validation experiments for the pancreas example including a knockout experiment as outlined in the response to reviewer 2, major point 1, where we also detail all the changes made to the analysis and manuscript.

3. My concerns extend to the OT approach's inherent limitations of ignoring underlying cellular dynamics. For instance, during a cell cycle process, OT mapping could incorrectly align two cells in an anti-clockwise orientation, even when the cell cycle flows clockwise. It might be beneficial for the authors to illustrate Moscot's effectiveness in revealing cell cycle progression and be clear about OT's limitations in the figure 1 or supplementary figure 1. You may consider using this cell cycle dataset to demonstrate this (The transcriptome dynamics of single cells during the cell cycle | Molecular Systems Biology (embopress.org)).

We thank the reviewer for pointing us to the nontrivial and exciting question to what extent OT can incorporate cell cycles. We interestingly find that OT can indeed reconstruct cell cycle states under rather weak assumptions, which we will illustrate both on synthetic data and on real-world data (Response Fig. 12).

Sketching the cell cycle following the model of Schwabe et al. ³²

To illustrate how OT can recover correct directions in a cell cycle we reproduce the 3d model which was used in the abstract figure of the proposed manuscript Schwabe et al³². We define a source and a target distribution on the data manifold (i.e. helix) corresponding to an early and later time point of measurements of cells, respectively. We then solve an optimal transport problem and sample (roughly equally space to facilitate visualization) data points in the source distribution and one data point from the conditional distribution in the target space, i.e. we sample from the predicted descending population of each single cell. The green lines connect the simulated cells in the early time point and their predicted descendants, and show that the transitions follow the simulated cell cycle.

Moscot is able to correctly identify the direction of the cell cycle

We demonstrate that this concept translates to real-world data by considering the proliferative ductal cell population in our newly generated pancreatic endocrinogenesis dataset. Therefore, we remove the Immature and Mature Acinar cells from the endocrinogenesis dataset (time points E14.5, E15.5, and E16.5) and compute OT couplings E14.5/E15.5 and E15.5/E16.5. We then assign a cell cycle phase (using marker genes from ³³) to Proliferative Ductal cells by following the procedure described in ³²/Materials and Methods/Filtering. We then aggregate the transport matrices to the level of cell cycle phases. This provides us with transition probabilities between cell cycle phases. From these transitions, we can infer the direction of the predicted cell cycle. To verify that the directionality is significant, we perform a permutation test. The test statistic lies in the 88th and 96th percentile for the E14.5/15.5 and E15.5/16.5 transition, respectively (with the 50th percentile denoting a random direction). Hence, moscot can recover the directions of a cell cycle. We added section 2.1.3 on the cell cycle analysis to the Methods, and refer to it in the main text:

Moreover, we recover the correct cell cycle direction with moscot (Supplementary Fig. 27 and Supplementary Note 6).

Limitations of recovering cell cycles with OT

As in all applications of OT, the distributional shift between cells (e.g. the “difference between the cells in an earlier time point and the cells in a later time point”) plays an important role to what extent the OT coupling is meaningful. Hence, we would like to clarify that we were able to show this phenomenon for this particular dataset, but do not claim that we will be able to recover such a signal in all datasets. We added Supplementary Note 6 *On the limitations of optimal transport for recovering cell cycles* where we state (given the cell cycle model by Schwabe et al. ³²):

Indeed, the distributional shift between two time points must be sufficiently small such that cells advance less than 180 degrees (in expectation) in the cell cycle. Note that the advancement in the cell cycle is due to the progression in the non-cycling dimensions of the cell trajectory. This limitation is due to a limited number of samples and moscot relying on discrete optimal transport, and hence not being able to recover the continuous trajectory of cells, but only the (stochastic) conditional distribution.

Response Fig. 12 **Moscot can recover cell cycles** -> included in our manuscript as **Supplementary Fig. 27**

a. Implementation of the sketch used for explaining transcriptomic dynamics during cell cycles in Schwabe et al. ³² as toy data example for cell cycle dynamics. Left: source distribution corresponding to an earlier time point. Middle: target distribution corresponding to a later time point. Right: Samples from the learnt transport map to model the trajectory of cells. **b.** Sketch of the ground truth biological cell cycle with indices used in the explanation. **c.** UMAP of the pancreatic endocrinogenesis dataset (E14.5, 15.5, 16.5) colored by

assigned cell cycle stage. **d.** Permutation test scores for assessing the correctness of the directionality of the cell cycle computed with `moscot.time` for time points E14.5/15.5 (left) and time points E15.5/16.5 (right).

4. In Figure 2, the computational speedup of `Moscot` over `WOT` is very valuable, but the quality of the alignment of `Moscot` with and without the low-rank approximation for computational speedup should also be explored. It is probably hard to directly benchmark the quality of the temporal mapping on the datasets from Figure 2 because of the lack of groundtruth. A potential evaluation scheme could involve testing the OT-based method's ability to recover alignments between the same cell using matrices of spliced RNAs or unspliced RNA. Comparisons with `WOT` using multi-omics datasets by mapping RNA to ATAC-seq information could also provide meaningful results. Ideally, your method should align multi-modality data from the same cell. You may also study how the performance changes as a function of the number of cells increases and when the method switches from full rank to low rank approximation.

We thank the reviewer for acknowledging the computational speedup of `moscot.time` compared to `WOT`. Following the reviewer's request, we include additional analysis in our revised manuscript where we validate that `moscot.time` reproduces `WOT` results on those time points of the embryogenesis example of Main Fig. 2 which are small enough to run `WOT` (see Response Fig. 3 and our reply to Reviewer 1, Minor comment 3) and investigate the performance of the low-rank approximation for different ranks and cell numbers (Response Fig. 13).

Benchmarking low-rank performance and speed

We compared the performance of default `moscot.time` with the low-rank approximation across time points on the embryogenesis example of Main Fig. 2. As expected, the low-rank approximation was faster than full-rank `moscot.time` for large enough time points (Response Fig. 13a,b). To compare the quality of the computed mappings of cells across time points, we used the germ-layer and curated cell-type transition metrics of Main Fig. 2. We found that larger cell numbers require larger ranks, and that rank 2000 provides a good approximation to full-rank performance across time points (Response Fig. 13c and **Methods**). We included the additional analysis in our revised manuscript in a new Supplementary Figure 2, and we added the following sentences to the main text:

We also included a low-rank OT approximation in `moscot`^{34,35}, which was faster than default `moscot.time` once we exceeded 75,000 cells per time point (Fig. 2b, Methods, and Supplementary Fig. 2a).

For the low-rank approximation, we found that accuracy in both metrics converged to default `moscot.time` for large enough ranks while being faster (Supplementary Fig. 2b).

Optimal transport for modality translation

We would like to point out that WOT is not able to translate RNA to ATAC data (or, more generally, any modality to any modality), but indeed SCOT²⁷ is able to do so. While we focus on temporal, spatial, and spatiotemporal problems in this manuscript, `moscot` also supports modality translation, which can be found in our documentation:

https://moscot.readthedocs.io/en/latest/notebooks/tutorials/600_tutorial_translation.html

We feel that including a benchmark on modality translation is beyond the scope of the current manuscript, which focuses on temporal, spatial, and spatiotemporal mapping and alignment problems.

Response Fig. 13 | **Low-rank approximates full-rank Sinkhorn at faster running times**
 -> included in our manuscript as **Supplementary Fig. 2**

a. Runtime in minutes to compute a coupling matrix (left) and to evaluate algorithm performance (right), across time points on the embryogenesis data of Fig. 2 (ref. ²⁰), for full-rank Sinkhorn (default `moscot.time`) and low-rank Sinkhorn^{1,34,35} for various ranks (Methods). **b.** Cell number per time point. **c.** Comparing low and full-rank approaches in

terms of the germ-layer (top) and curated transition (bottom) metrics of Fig. 2 for individual time points (left) and aggregated over time-windows (right; Methods).

5. It is interesting to see that Moscot approach yields nearly identical results to TOME, a simpler method based on KNN and gene expression similarity in Figure 2C. The conclusion drawn from this comparison should be carefully considered. This may indicate that achieving single-cell-level alignment might not be necessary or even relevant given the quality of existing scRNA-seq and spatial datasets, as cell-type level alignment may already reach the limits of what the existing data quality can offer. Similarly, this conclusion may affect the value of the spatiotemporal modeling of the heart and brain datasets, especially given the fact that the Stereo-seq dataset from this study don't really provide single cell information but binning data. Basically, do we really need the cell level alignment, can we just use TOME like method for spatiotemporal modeling given the general low quality of the existing data? I understand that your method can recover apoptosis and proliferation rates better than the scTOME approach (you extend TOME to cl-TOME). However, as stated, you also tuned the parameters in Moscot to achieve this goal. There's a lack of benchmarking on how these rates are exclusively derived from your methods. A comparison with simpler methods as outlined in the original WOT or the PRESCIENT, which directly uses the gene expression level of cell-cycle related genes, would be insightful.

In our revised manuscript, we have added additional analyses that compare moscot with TOME across applications and tasks. Specifically, for our temporal applications, we show that moscot is more robust than TOME when we change the latent cell representation (Response Fig. 10) and that moscot's growth rates correlate better with scanpy-derived growth rates (Response Fig. 14). Further, we have added a comparison to TOME in our spatiotemporal application to the Stereo-seq mouse embryogenesis dataset, where we show that moscot.spatiotemporal attains better accuracy than TOME (Response Fig. 15 and Figure 4 in the manuscript). We relate to this analysis in detail below.

Embedding robustness comparison

We respectfully disagree with the reviewer on the point that single-cell level mappings might not be necessary or relevant given the quality of existing scRNA-seq datasets. Indeed, TOME and moscot.time yield very similar results in terms of the germ-layer and curated transition scores of Main Figure 2; however, we show in our revised manuscript that moscot's results depend much less on the original Seurat batch effect correction (which we carried over from the original study²⁰) compared to TOME (Response Fig. 10). Please see our reply our Reviewer 2, major comment 2. In practice, this makes it easier for users to apply moscot to their own data.

Growth-rate comparison

As requested by the reviewer, we include additional analysis of moscot.time and TOME-derived cellular growth and death rates in our revised manuscript. In particular, we compare these predicted growth rates with scanpy-computed cell cycle scores³⁶ on an in-vitro reprogramming time course dataset⁷. We include the additional analysis in our manuscript in a new Supplementary Figure 4, and in this letter in Response Fig. 14.

Moscot and cl-TOME's predicted growth rates include the effects of stochastic cell sampling, proliferation, and apoptosis. Our main dataset of Figure 2 is an in-vivo embryogenesis dataset where each time point corresponds to a different individual²⁰. To counteract the effects of biased cell sampling between different individuals, which is unavoidable to some extent in such an in-vivo design, we choose to carry out this analysis in a different dataset, representing an in-vitro setting where we expect predicted growth rates to be largely driven by proliferation and apoptosis (Response Fig. 14a,b).

We used both moscot.time and cl-TOME to compute couplings across all 36 time points in this study, computed predicted growth rates, and correlated these with scanpy-computed cell-cycle scores on the cell-type level. We highlight in an exemplary pair of time points (day 15.5 to 16.0) how moscot's predicted growth rates correlate better than cl-TOMEs with the scanpy score (Pearson $r = 0.81$ vs. 0.71), and we show how this result generalizes across all pairs of time points (Pearson $r = 0.48$ vs. 0.13). Accordingly, we added the following sentences to the main text:

We also compared predicted growth rates with scanpy-computed cell cycle scores³⁶ on an in-vitro reprogramming dataset⁷, where we expected predictions to be less biased by stochastic cell sampling effects. We found that moscot.time's predictions across time points correlate better with cell-set averaged growth rates than cl-TOME (Pearson r of 0.48 versus 0.13 ; Supplementary Fig. 5).

Response Fig. 14 | **Predicted growth rates correlate well with cell cycle scores -> included in our manuscript as Supplementary Fig. 4**

a,b. Force-directed layout (FLE) of 165,892 cells across 39 time points, spanning days 0 to 18 of a reprogramming time course⁷, colored by experimental time point (**a**) and major cell sets (**b**). **c.** PCA embeddings of day 15.5/16.0 cells, colored by experimental time point (left), major cell sets as in (**b**) (middle), and scanpy-compute cell cycle scores³⁶ (right). **d.** PCA embeddings of the cell subset of (**c**), colored by moscot.time (top) and ci-TOME (bottom) predicted growth rates (Methods). **e.** Scatter plots of predicted (x-axis) versus

scanpy-computed (y-axis) growth rates, averaged over the major cell sets of (b), for moscot (top) and cl-TOME (bottom), over the 15.5/16.0 time point (left) and all time points (right).

Spatiotemporal comparison

We thank the reviewer for the suggestion. We have indeed also applied the TOME method to the spatiotemporal Stereo-seq dataset of mouse embryogenesis, and assess the reviewer's intuition that, due to the lack of single cell data, an approach like TOME could still provide accurate results in delineating the cell fate trajectories of mouse embryonic tissues. However, from our analysis using the cell transition scores as explained in the Methods, we show that TOME is inferior to both moscot's SpatioTemporal as well as moscot's Temporal problem, see Response Fig. 15. This benchmark highlights how learning a high-resolution cell fate mapping, which is also able to leverage spatial dependencies, is crucial even in applications where single-cell resolution is not available.

Response Fig. 15 | **Comparison of moscot.spatiotemporal with moscot.Temporal and TOME on the spatiotemporal mouse atlas -> included in our manuscript as Fig. 4b**

Curated cell transition score (Methods) on the MOSTA dataset³⁷ yielded by moscot.spatiotemporal, moscot.temporal, and TOME²⁰.

6. Although the authors presented the spatiotemporal mapping from figure 4 as a "novel" application, this novelty is limited and appears to originate more from the data than from the Moscot tool itself. With the Stereo-seq data analyzed in this study, concerns of the unfortunate lack of correspondence between cells across timepoints or spatial slices also arise due to imbalanced and limited sampling. Some data point may also have outliers that has no correspondence to other cells. All these will lead to a partial alignment, a problem which OT-based methods may not adequately

address and is extensively studied in the paste2 work (PASTE2: Partial Alignment of Multi-slice Spatially Resolved Transcriptomics Data | bioRxiv).

We thank the reviewer for this important remark. The incorporation of the recently published unbalancedness of low-rank optimal transport¹ now allows us to automatically discard outliers. We would like to emphasize that moscot (together with SPATEO) is the first method which explicitly incorporates spatial information into the trajectory algorithm and hence we consider this as a valuable and novel contribution. As prompted by the reviewer, we also benchmarked PASTE2 with moscot.spatiotemporal. We had to resort to a sub sampling strategy for PASTE2 given known scaling limitations of (<https://github.com/raphael-group/paste2/issues/2>; we consider n=10 independent sub samples for each time point) see Response Fig. 16. We point to the results in the main text by writing

In addition, moscot.spatiotemporal outperformed PASTE2 (Liu et al. 2023) being 10% more accurate on average across timepoints (Supplementary Fig. 18).

Response Fig. 16 | **Moscot.spatiotemporal outperforms PASTE2 -> included in our manuscript as Supplementary Fig. 18a**

Moscot.spatiotemporal's performance compared to PASTE2. Moscot's mapping is obtained using the unbalanced low rank approximation that uses the entire spatial slide for each time point whereas PASTE2 is evaluated using subsampling. Subsampling is performed over n=10 independent repetitions and error-bars depict 95% confidence interval.

7. The author mentioned the Spateo (Spateo: multidimensional spatiotemporal modeling of single-cell spatial transcriptomics | bioRxiv) work (which has its own limitations) in the discussion section. Although Spateo requires downsampling, it was

applied in the context of a 3D spatial transcriptomics that largely avoid the limited and unbalanced sampling of cells and integrated with the morphometric vector field approach so that the prediction of the cell migration extends to the all cells, in addition to the original sampled cells, in a whole embryo/organ.

We appreciate the thoughtful feedback from the reviewer. Following your recommendation, we conducted a moscot analysis on the spatiotemporal *Drosophila* embryo dataset, utilizing the same original data source as Spateo and specifically referencing the study by Wang et al.³⁸. This dataset, generated using stereo-seq technology, encompasses 2 *Drosophila* embryos across multiple z-stacks for the periods E14-16 and E16-18 (Response Fig. 17a,b). A thorough quality control and data examination revealed a pronounced batch effect between these two time points, as clearly depicted in Response Fig. 17c-e. Despite this, we employed moscot.spatiotemporal's analysis approach, which is adept at learning the cellular differentiation dynamics between the two time points by integrating gene expression similarity and spatial coordinates. It is crucial to note that moscot leverages the entire collection of slides for each z-stack, thereby treating each stereo-seq bin as a 3D spatial observation. This approach retains the original slide coordinates (x and y) and incorporates the z-stack level as provided by Wang et al., effectively capturing observations in three-dimensional space.

Through cell transition analysis (detailed in Methods), we successfully mapped the relevant tissue within the *Drosophila* embryo, as illustrated in Response Fig. 17f. Notably, our analysis, conducted in an unbalanced setting, demonstrated that for unique tissues such as hemolymph (present only at E14-16) and amnioserosa (found solely at E16-18), the total transported marginal was notably lower compared to other tissues (Response Fig. 17g). This finding underscores moscot's efficacy in navigating cell fate mapping challenges in unbalanced scenarios. Moreover, we pursued the identification of pivotal driver genes with moscot, recapitulating the orthogonal analysis done by the authors using SCENIC, a gene regulatory network inference method. Our focused examination on the distribution of cell types in the CNS (Response Fig. 17h) and muscle (Response Fig. 17i) tissue regions revealed key transcription factors (TFs) consistent with those identified by the original study (³⁸ Figure 6), alongside genes prominently linked to embryonic development, such as *Obp44a*³⁹, noted for its expression in late-stage embryonic longitudinal glia, ventral nerve cord, and brain, and *fax*⁴⁰ which plays a role in axon development and shows increased expression throughout embryogenesis. For muscle tissue, we pinpointed *Mef2* as a critical TF, as highlighted by the original study³⁸ (Figure 6), in addition to *Mlc1* and *Mlc2*, both integral to muscle development⁴¹.

Our findings suggest that moscot is aptly suited for direct application to 3D spatiotemporal datasets, facilitating the study of tissue development even under less-than-ideal conditions,

such as significant batch effects and the absence of explicit cell identities, due to the binning approach inherent in stereo-seq technologies.

We have added the following text and figures to the manuscript:

To demonstrate the performance of `moscot.spatiotemporal` applied to 3D spatio-temporal omics data, we study embryonic development of the drosophila leveraging 3D Stereo-seq technology to capture embryonic days E14-16 to E16-18³⁸ (Supplementary Fig. 22, Methods). We demonstrate the robustness of `moscot.spatiotemporal` to transient cell states on the hemolymph and amnioserosa cell clusters, reporting that the aggregated transported mass was notably lower compared to other tissues (Supplementary Fig. 22g). Furthermore, our computational analysis revealed key transcription factors (TFs) such as *Fax*⁴⁰ which plays a role in axon development and shows increased expression throughout embryogenesis in the Central Nervous System (CNS) tissue, alongside genes prominently linked to CNS development, such as *Obp44a*³⁹, noted for its expression in late-stage embryonic longitudinal glia, ventral nerve cord, and brain. We were also able to pinpoint *Mef2* as a critical TF for muscle tissue development, as reported by the original study³⁸, in addition to *Mlc1* and *Mlc2*, both integral to muscle development⁴¹.

Response Fig. 17 | [moscot.spatiotemporal](https://doi.org/10.26434/chemrxiv-2024-11) recovers regulatory mechanisms in the developing drosophila embryo leveraging 3D spatial technologies³⁸. -> included in our manuscript as **Supplementary Fig. 22**

a. 3D spatial visualization of cell annotations across various tissue types in Drosophila embryos E14-16, as reported by original Wang et al.³⁸. **b.** 3D spatial visualization of cell

annotations across various tissue types in *Drosophila* embryos E16-18, as reported by Wang et al.³⁸. **c.** UMAP visualization of cell annotations across various tissue types in *Drosophila* embryos, as reported by Wang et al.³⁸. **d.** UMAP visualization of time points 14 which represents E14-16, and 16, which represents E16-18. **e.** UMAP visualization of slide ID, that is the unique ID source of each stereo-seq slide. **f.** Heatmap displaying the cell transition probabilities from source (E14-16) to target (E16-18). **g.** Bar graphs indicate the mass distribution across various tissues in the pushforward and pullback distribution, highlighting how hemolymph and amnioserosa, present only at E14-16 and E16-18 respectively, have lower mass than other tissues. **h.** Identification of top 20 driver genes for CNS tissue development, with transcription factors (TFs) indicated in orange and other genes in blue. The right side of the panel shows UMAP plots with expression levels of selected genes, including *Rbp6*, *Obp44a*, and *fax*. *Rbp5* and *fax* were identified also in the original study of Wang et al. leveraging a different algorithm. **i.** Identification of top 30 driver genes for muscle tissue development, with transcription factors (TFs) indicated in orange and other genes in blue. The right side of the panel illustrates UMAP plots showing expression levels for *Mef2*, *Mlc1*, and *Mlc2*. *Mef2* is also reported by Wang et al., and it was identified leveraging a different algorithm.

8. Furthermore, since the key advances of Moscot is its scalability. Is this really useful in complicated system (in addition to what has shown in figure 2), can for example, Moscot predict the binning coupling on the entire embryonic slices across multiple time points of the MOSTA dataset? If it is operated on the entire slice, is it still capable of predicting the evolution of each cell type? Will it actually leading to cases where a particular cell type be mapped to a different cell types over time?

We would like to point out that scalability is only one out of three key advances of moscot, as displayed in Figure 1. With respect to the first question, the answer is yes, moscot is able to compute couplings for all observations across embryonic slices. The answer to the second question is also positive, moscot is able to predict the evolution of cells, and hence cell types, for the entire slice, across time points. In fact, the analysis shown in Figure 4 shows precisely this result for the high-resolution annotation of the brain region. With respect to the third question, we share the concern of the reviewer about the accuracy of the map when predicting the evolution of a cell type over multiple time points. Nevertheless, we demonstrated the robustness of such analysis in the response to major point 2, reviewer 1. We were able to show that concatenating transport plans across multiple time points on entire slices of the MOSTA dataset yields relatively consistent results. We have extended the main text accordingly as pointed out above.

9. The reliance of biological insights presented in figures 4 and 5 on Cellrank makes the findings seem like an application of a previous work. I recommend that the authors attempt to reveal new biological insights directly from Moscot's results. For example, can you demonstrate directly, the spatial, temporal and spatiotemporal coupling matrices' structure can reveal novel biological insights already? Because this matrix doesn't rely on other methods which is not really a direct result from Moscot work itself. Some ideas maybe that: 1. A widely mapping spectrum of cells (in studies of figure 2/5) may implies plasticity of a cell type in the temporal mapping; 2. A narrow mapping distribution of cells (in studies of figure 3/4) may tells you something about the physical/structural constraints or the biological importance of a particular tissue domain or organ region. 3. can you identity genes that significantly affect the mapping after you remove a particular gene or a group of genes before the OT mapping?

We thank the reviewer for the suggestions, and would like to highlight that CellRank was not used in figure 5. Moreover, we would like to point out that one motivation of creating moscot is its interoperability with other common tools in single-cell genomics, to facilitate workflows in the analysis of single-cell genomics data. Hence, we believe that the interoperability with CellRank is an important contribution, which we demonstrate in figure 4.

We added driver gene identification to the spatiotemporal analysis. First, using moscot.spatiotemporal's mappings of the MOSTA dataset we compute the pull-back of liver annotations across time points. Next, utilizing moscot's compute_feature_correlation method, we identify the driver genes which are correlated with the pull-back. Top correlated genes follow previous reports regarding early development of the liver, with the top three correlated genes being *Afp*, *Alb* and *Apoa2* in accordance with the literature^{42,43}. Top ranked genes identified by this analysis along with references to relevant literature is provided in Supplementary Table 6.

Moreover, we added a method to identify target genes, which is, for example, not possible to do with CellRank. We added a part to the methods section 2.1, stating

To compute target genes of a transcription factor, we correlate the push-forward distribution of the expression of the transcription factor with all genes, and identify highly correlated genes as target genes.

Thanks to moscot, we identified target genes for *Hnf4a*, a TF associated with liver development^{44,45,46}. We found hepatocyte markers, such as *Afp*, *Apoa2*, and *Apoa1*, in accordance with literature studying its role in liver development^{44,45,47}. Top ranked target genes identified by this analysis along with references to relevant literature are provided in Supplementary Table 6.

While we consider suggestions 1 and 2 as problematic (as the result is more influenced by the density of cells rather than uncertainty in many scenarios), we thank the reviewer for suggestion 3, the results of which can be found in Response Fig. 18.

Moreover, we added an analysis of the pancreatic endocrinogenesis dataset with Sparse Monge ². Sparse Monge allows for high-dimensional optimal transport (i.e. non-embedded gene space), while automatically selecting the most relevant features in gene space, building upon the concept of entropic maps ⁴⁸. This allows us to identify single genes per cell which drive the developmental process. We show that this indeed helps to recover relevant genes in pancreatic endocrinogenesis.

We refer to the interpretability of transport maps in the main text:

We confirmed our driver gene identification strategy using an orthogonal, new approach, which is based on feature-sparse OT (Cuturi et al. 2023) (Supplementary Fig. 38, Methods). Moreover, we perform differential feature analysis, resulting in a consistent set of cell type specific genes (Supplementary Fig. 39, Methods).

Response Fig. 18 | **Feature importance by considering the influence on a transport map -> included in our manuscript as Supplementary Fig. 39**

a. Left: Subset of marker genes in pancreatic endocrinogenesis (*Neurod2* as activator of the epsilon lineage, *Fev* as activator of the alpha and beta lineage, *Neurog3* as activator of endocrine formation and *Sst* as hormone emitted by delta cells) and their corresponding significance when leaving them out from the gene set for the computation of the optimal transport matrix. Right: The nine most significant features identified with the same procedure which are also transcription factors (from left to right: *Sox4*⁴⁹, *Mctp2*, *Meis1*, *Neurog3*⁵⁰, *Sim1*⁵¹, *Etv1*⁵², *Cbfa2t2*, *Cers6*, *Prdm16*⁵³). **b.** UMAP embedding of the considered cells, colored by cell type and normalized gene expression corresponding to the 7 most significant genes.

b Variability in gene importance

c Variability in gene importance per cell type

Response Fig. 19 | **Feature-level interpretation of optimal transport maps using Sparse Monge** -> included in our manuscript as **Supplementary Fig. 38**

a. Gene importance per cell type in the pancreatic endocrinogenesis dataset using Sparse Monge² (Methods). **b.** The variability in gene importance computed based on feature-sparse transport maps recovers lineage branching events (Methods). **c.** Aggregation of the variability in gene importance to cell type level.

10. Figure 5 on lineage relationship of delta and epsilon cells. In this section the authors generated a new datasets that aim to disentangling the lineage relationship of delta

and epsilon cells. There are a few things we can consider to further enhance this section. First, my concern of OT method for mapping cell fate is that it doesn't consider dynamics and thus the mapping can be problematic. You can first fix this by including RNA velocity analyses with your scVelo tool. Second, you can also leverage your multi-omics datasets to perform multi-velo to potentially further improve your dynamics quantification. Third, a lineage tracing experiment, even the one with static barcode or a marker-gene based live cell tracking of cell lineage can be helpful.

We thank the reviewer for this remark. First, we would like to emphasize that OT considers distributional mappings. In particular, this mapping considers cellular dynamics (Methods). Hence, we apologize that we do not understand the statement that "OT [...] doesn't consider dynamics and thus the mapping can be problematic". Yet, as suggested by the reviewer, we applied a wide range of commonly used trajectory inference tools, namely scVelo⁵⁴, MultiVelo⁵⁵, VeloVI⁵⁶, Diffusion Pseudotime⁵⁷ and graph dynamics⁵⁸. We analyze the learned dynamics using CellRank⁵⁸ by considering cell type transitions and plotting the velocity stream embedding plots, as well as absorption probabilities (Response Fig. 20, Response Fig. 21, Response Fig. 22). The results show that inferring trajectories on this dataset is not straightforward, as some methods result in transitions which contradict known biology.

We refer to the analysis of the pancreatic endocrinogenesis dataset in the main text:

To validate this finding, we predicted ancestry relationships using established trajectory inference methods (Supplementary Fig. 46, Supplementary Fig. 47, Supplementary Fig. 48, Methods). While some methods support our claim, others predict biologically implausible trajectories, highlighting the complex lineage relationships in the developing pancreas.

Response Fig. 20 | Analysis of the pancreatic endocrinogenesis dataset with different trajectory inference methods -> included in our manuscript as Supplementary Fig. 46

Trajectory analysis of the pancreatic endocrinogenesis dataset with commonly used TI methods (Methods). We use CellRank^{58,59} to compute fate probabilities to the four endocrine cell states alpha, beta, delta and epsilon for each single cell followed by aggregation to cell type level (Methods) based on **a.** diffusion pseudotime⁵⁷, **b.** scVelo⁵⁴, **c.** VeloVI⁵⁶, **d.** MultiVelo⁵⁵, **e.** CytoTrace⁶⁰, and connectivity of the graph⁵⁸.

Response Fig. 21 | **Fate probabilities of the pancreatic endocrinogenesis dataset predicted with different trajectory inference methods -> included in our manuscript as Supplementary Fig. 47**

Trajectory analysis of the pancreatic endocrinogenesis dataset with commonly used TI methods (Methods). We use CellRank^{58,59} to compute fate probabilities to the four endocrine

cell states alpha, beta, delta and epsilon (Methods) based on **a.** diffusion pseudotime ⁵⁷, **b.** scVelo ⁵⁴, **c.** VeloVI ⁵⁶, **d.** MultiVelo ⁵⁵, **e.** CytoTrace ⁶⁰, and connectivity of the graph ⁵⁸.

Response Fig. 22 | **Stream embedding plots of the dynamics obtained by different trajectory inference methods on the pancreatic endocrinogenesis -> included in our manuscript as Supplementary Fig. 4**

Trajectory analysis of the pancreatic endocrinogenesis dataset with commonly used TI methods (Methods). We use CellRank^{58,59} to plot the dynamics learnt with **a.** diffusion pseudotime⁵⁷, **b.** scVelo⁵⁴, **c.** VeloVI⁵⁶, **d.** MultiVelo⁵⁵, **e.** CytoTrace⁶⁰, and connectivity of the graph⁵⁸.

To conclude, while the study represents a substantial effort to scale the OT method and provides valuable improvements over several existing approaches, the novelty of the methods and the biological insights derived are limited for a journal of Nature's caliber.

Minor points

1. The use of the term "mapping cells" in the paper's title may be misleading, considering that the analyses on the MOSTA Stereo-seq study provide bins rather than cells.

We included the following sentence in the Introduction to clarify this point:

Depending on the application, samples may correspond to sequenced cells (e.g. scRNA-seq), segmented cells (e.g. MERFISH⁶¹) or spatial array locations (Stereo-Seq³⁷).

We have also modified corresponding text in our Results section to clarify this point.

2. The resolution of figure 3a/c is low, making it difficult to read.

We improved the quality of figure 3a/c

3. The authors should demonstrate Moscot's performance under different batch conditions and its ability to handle batch effects, especially considering the high batch effects of the MOSTA and other spatial datasets.

We agree with the reviewer that there are substantial batch effects in the MOSTA dataset. Suppl. Figure 17a visualizes these batch effects on a UMAP embedding. Yet, we show in the corresponding section that the results obtained with `moscot.spatiotemporal` are biologically meaningful. Hence, `moscot` is able to handle batch effects in the MOSTA dataset.

4. It would be valuable to see the actual aligned cell pairs for both the aligned heart and brain cells across time for the figure 4 analyses.

We would like to highlight that `moscot` solves *entropy-regularized* optimal transport problems, which has favorable statistical and computational properties as opposed to non-entropy-regularized optimal transport problems^{62,63}. Thus, the resulting optimal transport map (or transport plan) is stochastic, and not deterministic. Hence, we do not align cell pairs, but obtain a stochastic coupling between a cell and a set of cells.

Response Fig. 22 | The stochastic mapping of a single heart bin in E9.5 to all bins in E10.5, that is a single row in the obtained mapping. The opacity of the connecting line is scaled according to the mass allocated for the given pair.

5. It is important to note that the reason the brain cells from E13.5 to E15.5 are not annotated is potentially that cells at these time points are less mature and thus not necessarily having the same cell identities as in E16.5. Also, this spatiotemporal mapping will suffer the same issue of the developing heart as mentioned in the major comments (limited and unbalanced sampling, partial alignment and batch effects).

We agree with this point and also believe that this is the main reason why considering couplings across multiple time points is of limited feasibility. We discuss the concatenation of multiple pairwise OT problems in the temporal setting in the response to reviewer 1, major point 1, and verify the assumption in the response to reviewer 1, major point 2.

6. Another potential issue of spatial and spatiotemporal mapping and alignment of Mosto in figure 3/4 is that they are done sequentially. If alignment from A/B is already problematic and B/C is additionally problematic, will that making the alignment between A/C, which is required for looking for cells across the long spatial and temporal axis, to be even more problematic?

We agree that the more couplings we concatenate, the less accurate the mapping will become as errors are propagated. Yet, we refer to Reviewer 1, major point 2, where we show with different metrics that the spatiotemporal mapping across multiple time points is meaningful.

References

1. Scetbon, M., Klein, M., Palla, G. & Cuturi, M. Unbalanced Low-rank Optimal Transport Solvers. *arXiv [cs.LG]* (2023).
2. Cuturi, M., Klein, M. & Ablin, P. Monge, Bregman and Occam: Interpretable Optimal Transport in High-Dimensions with Feature-Sparse Maps. *arXiv [stat.ML]* (2023).
3. Eyring, L. V. *et al.* Modeling Single-Cell Dynamics Using Unbalanced Parameterized Monge Maps. *bioRxiv* 2022.10.04.510766 (2022) doi:10.1101/2022.10.04.510766.
4. Eyring, L. *et al.* Unbalancedness in Neural Monge Maps Improves Unpaired Domain Translation. *arXiv [cs.CV]* (2023).
5. Klein, D., Uscidda, T. 'eo, Theis, F. & Cuturi, M. Generative entropic neural optimal transport to map within and across spaces. *ArXiv abs/2310.09254*, (2023).
6. Lavenant, H., Zhang, S., Kim, Y.-H. & Schiebinger, G. Towards a mathematical theory of trajectory inference. *arXiv [stat.ML]* (2021).
7. Schiebinger, G. *et al.* Optimal-Transport Analysis of Single-Cell Gene Expression Identifies Developmental Trajectories in Reprogramming. *Cell* **176**, 928–943.e22 (2019).
8. Nitzan, M., Karaiskos, N., Friedman, N. & Rajewsky, N. Gene expression cartography. *Nature* **576**, 132–137 (2019).
9. Li, B. *et al.* Benchmarking spatial and single-cell transcriptomics integration methods for transcript distribution prediction and cell type deconvolution. *Nat. Methods* 1–9 (2022).
10. Interpolating between Optimal Transport and MMD using. <http://proceedings.mlr.press> › ...
...<http://proceedings.mlr.press> › ...
11. Deng, Y. *et al.* Spatial-CUT&Tag: Spatially resolved chromatin modification profiling at the cellular level. *Science* **375**, 681–686 (2022).
12. Bastidas-Ponce, A. *et al.* Comprehensive single cell mRNA profiling reveals a detailed roadmap for pancreatic endocrinogenesis. *Development* **146**, (2019).
13. Byrnes, L. E. *et al.* Lineage dynamics of murine pancreatic development at single-cell resolution. *Nat. Commun.* **9**, 3922 (2018).
14. Yu, X.-X. *et al.* Sequential progenitor states mark the generation of pancreatic endocrine lineages in mice and humans. *Cell Res.* **31**, 886–903 (2021).
15. Cota, P. *et al.* NEUROD2 function is dispensable for human pancreatic β cell specification. *Front. Endocrinol.* **14**, 1286590 (2023).
16. Becht, E. *et al.* Dimensionality reduction for visualizing single-cell data using UMAP. *Nat. Biotechnol.* (2018)

doi:10.1038/nbt.4314.

17. McInnes, L., Healy, J. & Melville, J. UMAP: Uniform Manifold Approximation and Projection for Dimension Reduction. *arXiv [stat.ML]* (2018).
18. Moon, K. R. *et al.* Visualizing structure and transitions in high-dimensional biological data. *Nat. Biotechnol.* **37**, 1482–1492 (2019).
19. Bravo González-Blas, C. *et al.* SCENIC+: single-cell multiomic inference of enhancers and gene regulatory networks. *Nat. Methods* **20**, 1355–1367 (2023).
20. Qiu, C. *et al.* Systematic reconstruction of cellular trajectories across mouse embryogenesis. *Nat. Genet.* **54**, 328–341 (2022).
21. Dong, M. *et al.* Causal identification of single-cell experimental perturbation effects with CINEMA-OT. *Nat. Methods* **20**, 1769–1779 (2023).
22. Heumos, L. *et al.* Best practices for single-cell analysis across modalities. *Nat. Rev. Genet.* **24**, 550–572 (2023).
23. Luecken, M. D. *et al.* A sandbox for prediction and integration of DNA, RNA, and proteins in single cells. in *35th Conference on Neural Information Processing Systems (NeurIPS 2021) Track on Datasets and Benchmarks* (biblio.ugent.be, 2021).
24. Lopez, R. *et al.* A joint model of unpaired data from scRNA-seq and spatial transcriptomics for imputing missing gene expression measurements. *arXiv [cs.LG]* (2019).
25. Martens, L. D., Fischer, D. S., Theis, F. J. & Gagneur, J. Modeling fragment counts improves single-cell ATAC-seq analysis. *bioRxiv* 2022.05.04.490536 (2022) doi:10.1101/2022.05.04.490536.
26. Ashuach, T. *et al.* MultiVI: deep generative model for the integration of multimodal data. *Nat. Methods* **20**, 1222–1231 (2023).
27. Demetci, P., Santorella, R., Sandstede, B., Noble, W. S. & Singh, R. SCOT: Single-Cell Multi-Omics Alignment with Optimal Transport. *J. Comput. Biol.* **29**, 3–18 (2022).
28. Peyré, G. & Cuturi, M. Computational optimal transport: With applications to data science. *Foundations and Trends® in Machine* (2019).
29. Biancalani, T. *et al.* Deep learning and alignment of spatially resolved single-cell transcriptomes with Tangram. *Nat. Methods* **18**, 1352–1362 (2021).
30. Jiang, M., Wu, Y., Zhao, T., Zhao, Z. & Lu, C. PointSIFT: A SIFT-like Network Module for 3D Point Cloud Semantic Segmentation. *arXiv [cs.CV]* (2018).
31. Myronenko, A. & Song, X. Point set registration: coherent point drift. *IEEE Trans. Pattern Anal. Mach. Intell.* **32**, 2262–2275 (2010).
32. Schwabe, D., Formichetti, S., Junker, J. P., Falcke, M. & Rajewsky, N. The transcriptome dynamics of single cells during the cell cycle. *Mol. Syst. Biol.* **16**, e9946 (2020).

33. Macosko, E. Z. *et al.* Highly Parallel Genome-wide Expression Profiling of Individual Cells Using Nanoliter Droplets. *Cell* **161**, 1202–1214 (2015).
34. Scetbon, M., Cuturi, M. & Peyré, G. Low-Rank Sinkhorn Factorization. *arXiv [stat.ML]* (2021).
35. Scetbon, M., Peyré, G. & Cuturi, M. Linear-Time Gromov Wasserstein Distances using Low Rank Couplings and Costs. *arXiv [cs.LG]* (2021).
36. Wolf, F. A., Angerer, P. & Theis, F. J. SCANPY: large-scale single-cell gene expression data analysis. *Genome Biol.* **19**, 15 (2018).
37. Chen, A. *et al.* Spatiotemporal transcriptomic atlas of mouse organogenesis using DNA nanoball-patterned arrays. *Cell* **185**, 1777–1792.e21 (2022).
38. Wang, M. *et al.* High-resolution 3D spatiotemporal transcriptomic maps of developing *Drosophila* embryos and larvae. *Dev. Cell* **57**, 1271–1283.e4 (2022).
39. Fisher, W. W. *et al.* A modERN resource: identification of *Drosophila* transcription factor candidate target genes using RNAi. *Genetics* **223**, (2023).
40. Fabre, B. *et al.* Analysis of *Drosophila melanogaster* proteome dynamics during embryonic development by a combination of label-free proteomics approaches. *Proteomics* **16**, 2068–2080 (2016).
41. Leicht, B. G., Lyckegaard, E. M., Benedict, C. M. & Clark, A. G. Conservation of alternative splicing and genomic organization of the myosin alkali light-chain (*Mlc1*) gene among *Drosophila* species. *Mol. Biol. Evol.* **10**, 769–790 (1993).
42. Lotto, J. *et al.* Single-Cell Transcriptomics Reveals Early Emergence of Liver Parenchymal and Non-parenchymal Cell Lineages. *Cell* **183**, 702–716.e14 (2020).
43. Kwon, G. S. *et al.* Tg(Afp-GFP) expression marks primitive and definitive endoderm lineages during mouse development. *Dev. Dyn.* **235**, 2549–2558 (2006).
44. Parviz, F. *et al.* Hepatocyte nuclear factor 4alpha controls the development of a hepatic epithelium and liver morphogenesis. *Nat. Genet.* **34**, 292–296 (2003).
45. Torres-Padilla, M. E., Fougère-Deschatrette, C. & Weiss, M. C. Expression of HNF4alpha isoforms in mouse liver development is regulated by sequential promoter usage and constitutive 3' end splicing. *Mech. Dev.* **109**, 183–193 (2001).
46. Duncan, S. A. Mechanisms controlling early development of the liver. *Mech. Dev.* **120**, 19–33 (2003).
47. Rouillard, A. D. *et al.* The harmonizome: a collection of processed datasets gathered to serve and mine knowledge about genes and proteins. *Database* **2016**, (2016).
48. Pooladian, A. A. & Niles-Weed, J. Entropic estimation of optimal transport maps. *arXiv preprint arXiv:2109.12004* (2021).
49. Xu, E. E. *et al.* SOX4 cooperates with neurogenin 3 to regulate endocrine pancreas formation in mouse models. *Diabetologia* **58**, 1013–1023 (2015).

50. Schreiber, V. *et al.* Extensive NEUROG3 occupancy in the human pancreatic endocrine gene regulatory network. *Mol Metab* **53**, 101313 (2021).
51. Petersen, M. B. K. *et al.* Single-Cell Gene Expression Analysis of a Human ESC Model of Pancreatic Endocrine Development Reveals Different Paths to β -Cell Differentiation. *Stem Cell Reports* **9**, 1246–1261 (2017).
52. Heeg, S. *et al.* ETS-Transcription Factor ETV1 Regulates Stromal Expansion and Metastasis in Pancreatic Cancer. *Gastroenterology* **151**, 540–553.e14 (2016).
53. Jiang, N., Yang, M., Han, Y., Zhao, H. & Sun, L. PRDM16 Regulating Adipocyte Transformation and Thermogenesis: A Promising Therapeutic Target for Obesity and Diabetes. *Front. Pharmacol.* **13**, 870250 (2022).
54. Bergen, V., Lange, M., Peidli, S., Wolf, F. A. & Theis, F. J. Generalizing RNA velocity to transient cell states through dynamical modeling. *Nat. Biotechnol.* **38**, 1408–1414 (2020).
55. Li, C., Virgilio, M. C., Collins, K. L. & Welch, J. D. Multi-omic single-cell velocity models epigenome–transcriptome interactions and improves cell fate prediction. *Nat. Biotechnol.* **41**, 387–398 (2022).
56. Gayoso, A. *et al.* Deep generative modeling of transcriptional dynamics for RNA velocity analysis in single cells. *Nat. Methods* **21**, 50–59 (2024).
57. Haghverdi, L., Büttner, M., Wolf, F. A., Buettner, F. & Theis, F. J. Diffusion pseudotime robustly reconstructs lineage branching. *Nat. Methods* **13**, 845–848 (2016).
58. Lange, M. *et al.* CellRank for directed single-cell fate mapping. *Nat. Methods* **19**, 159–170 (2022).
59. Weiler, P., Lange, M., Klein, M., Pe'er, D. & Theis, F. J. Unified fate mapping in multiview single-cell data. *bioRxiv* 2023.07.19.549685 (2023) doi:10.1101/2023.07.19.549685.
60. Gulati, G. S. *et al.* Single-cell transcriptional diversity is a hallmark of developmental potential. *Science* **367**, 405–411 (2020).
61. Chen, K. H., Boettiger, A. N., Moffitt, J. R., Wang, S. & Zhuang, X. RNA imaging. Spatially resolved, highly multiplexed RNA profiling in single cells. *Science* **348**, aaa6090 (2015).
62. Genevay, A., Chizat, L., Bach, F., Cuturi, M. & Peyré, G. Sample Complexity of Sinkhorn Divergences. in *Proceedings of the Twenty-Second International Conference on Artificial Intelligence and Statistics* (eds. Chaudhuri, K. & Sugiyama, M.) vol. 89 1574–1583 (PMLR, 16–18 Apr 2019).
63. Mena, G. & Niles-Weed, J. Statistical bounds for entropic optimal transport: sample complexity and the central limit theorem. *Adv. Neural Inf. Process. Syst.* (2019).

Reviewer Reports on the First Revision:

Referees' comments:

Referee #1 (Remarks to the Author):

We want to thank the authors for carefully addressing almost all the points we raised in our previous review. In particular, they show

- 1) that their new framework, called moscot, which has a more computationally efficient implementation of WOT is also able to reproduce its results
- 2) they document the optimal transport consistency in time from figures from the literature, but also with new experiments with moscot.
- 3) they point out their figure 3b that shows how more spatial homogeneity improves the performance of moscot, but also that even in heterogeneous samples where moscot performs worse, it still outperforms the competitors (albeit by a little). However, this doesn't completely address the point about the consistency in assuming that biology introduces batch effects always via optimal transport. It is in fact very possible that batch effects introduced from different experimental protocols influence some cell types differently than others, or even that they cause flows (e.g. with non zero curl) which optimal transport could never capture. The authors should include the originally proposed benchmarking between three batches A,B,C and report the difference $P_{\{A,B\}} * P_{\{B,C\}} * P_{\{C,A\}} - \text{Identity}$. It is a useful benchmarking to have since it will showcase the upper limit of how good optimal transport is for batch effects.

Moreover, We feel that it is important to emphasize on the complexity of trajectory inference in the discussion over the contradictory lineage hypotheses the authors had come up with between previous cellrank paper and this paper (i.e. from Ngn3+ to epsilon and delta cell states), acknowledging that the conclusions regarding lineages are highly dependent on the integrated level of data modalities and the methods used for embedding construction and pseudotime inference. This could be more highlighted along the direction of this statement "While this hypothesis requires further experimental validation, our results demonstrate the potential of moscot to generate novel insights into complex biological systems" in the first paragraph of discussion.

In Suppl. Fig. 48, on testing other pseudotime methods, it would be better to specify which tools output a lineage structure consistent with what you report from moscot; In particular, to clearly compare with previous cellrank based hypothesis and conclude whether the difference mainly comes from integrating more modalities (transcriptome vs. transcriptome + epigenome) or the different models.

We also would like to make the below suggestions:

Showing the trajectory directionality diagram (using cellrank kernel projection method) for moscot like shown for other methods in Supplementary fig. 48.

Running the other pseudotime methods on the Bastidas-Ponce et al. dataset to see if the inferred

branchings remain consistent with moscot results.

Changing Supp. fig 31 with better labeling and structure to clearly indicate which states are ancestral and which states are descendant

Referee #1 (Remarks on code availability):

We were able to install the Moscot and then use it to reproduce all the results in this tutorial https://moscot.readthedocs.io/en/latest/notebooks/tutorials/100_lineage.html.

Referee #2 (Remarks to the Author):

I appreciate that the authors' efforts addressing my questions. The revised version added new figures and make a new biology points that NEUROD2 is an important factor for Epsilon differentiation and validated using previous generated KO NEUROD2 human iPSCs. However, I have some questions regarding the newly added data.

1. P-value/ statistical significance is missing in Figure 5h & I and Suppl. Fig. 43 NEUROD2 KO experiments.
2. What is the dynamic lineage expression pattern of NEUROD2 during mouse development in your data? The timing of the NEUROD2 expression is important for understanding its function on lineage specification for delta and the epsilon cells.
3. NEUROD2 has been described "to be the second most relevant transcription factor for both the Fev+ delta and the epsilon progenitor population" here. However, figure 5h shows NEUROD2 KO can only affect epsilon but not delta differentiation. How to explain the discrepancy?
4. Are the cells between control vs C37, C89 derived from same iPSC lines?

Referee #3 (Remarks to the Author):

I greatly appreciate the authors' efforts to address important issues I raised properly. However, I found the following issues that remain to be addressed:

1. The authors seem to completely ignore several suggestions I made in my reviewer comments, including for example:
 - reviewer comments 4, I mentioned the possibility to use unspliced /unspliced RNA from the same cell as a way to validate the ground-truth mapping of Moscot. However, this was ignored (I agree the multi-modality mapping is not needed given the focus of this work).
 - the author used Scanpy's code to benchmark the calculated apoptosis and proliferation rates instead of "comparison with simpler methods as outlined in the original WOT or the PRESCIENT". My guess is that Scanpy implemented the rate calculations from these tools. Even though this is true, it is worthwhile to explicitly mention that.

2. Some of the claims / arguments from the author are not sound and convincing, including:

- for reviewer comment 5 about the usefulness of single cell level mapping, the authors responded by saying that 1. Moscot and TOME achieve similar results 2. Moscot is more robust comparing to TOME for batch effects. These are fine results, but they don't answer my questions about the need, superiority of single cell level mapping over meta-cell or cell group level mapping.
- for reviewer comments 9, the authors considered the first two suggestions are problematic because "the result is more influenced by the density of cells rather than uncertainty in many scenarios". However, the authors had already analyzed the whole-body drosophila data, they should use this dataset to demonstrate some of these points to confirm their claims as this dataset has no sampling issue and estimation of cell density is ideal.

Referee #4 (Remarks to the Author):

Summary of the key results

This paper presents Multi-Omics Single-Cell Optimal Transport (moscot), a computational framework to solve mapping and alignment problems that arise in single-cell genomics. Compared to other methods for single-cell genomics relying on measurements, using a computational framework is not cell-destructive and can capture the whole molecular information. Compared to other computational frameworks for single-cell genomics, moscot supports multimodal data, is more scalable (because it relies on recent optimal transport techniques), and unifies previous applications of optimal transport to single-cell genomics in a single software.

Originality and significance

In general, this software should be of great interest to the community, thanks to its scalability in the era of large datasets with temporal and spatial dimensions, and its ability to handle multimodal data, without experimentations which can be cell destructive. In particular, moscot could recover murine differentiation trajectories during embryogenesis across time and space, enriched spatial liver samples with multimodal information, and aligned brain tissue slides. One should be able to apply moscot to a wide range of problems in single cell genomics.

Data & methodology: validity of approach, quality of data, quality of presentation

The paper presents the software and convincing use cases. The algorithmic behind moscot relies on recent advances in understanding and implementing optimal transport techniques. These optimal transport techniques have already been applied successfully in various fields of statistics and machine learning, and their application to single cell genomics makes perfect sense.

Appropriate use of statistics and treatment of uncertainties

I am not sure if uncertainty quantification when mapping and aligning or fitting trajectories is supported by the software. Could the authors clarify please?

Conclusions: robustness, validity, reliability

The conclusions drawn from the different use cases of moscot seem reasonable given the data in my possession. However, I am not an expert in single cell genomics to evaluate the originality of these conclusions compared to the literature.

Suggested improvements: experiments, data for possible revision

No suggestions.

References: appropriate credit to previous work?

SPATEO seems to be a directly related software and a competitor, I would have expected a citation and a comparison to this work from the beginning of the paper.

Clarity and context: lucidity of abstract/summary, appropriateness of abstract, introduction and conclusions

In general, the paper is clear and well-written. A conclusion would be appreciated (maybe expand a little bit the last paragraph of the main paper and turn it into a conclusion).

I recommend publication for this work. I believe this software unifies many methods and will be of interest to biologists looking to apply modern methods to their datasets with minimal knowledge of optimal transport techniques.

Referee #5 (Remarks to the Author):

This manuscript describes and validates a computation framework called moscot that is capable of computationally assessing very large single cell transcriptome data sets for temporal, spatial and spatiotemporal data sets. The authors demonstrate that moscot computationally outperforms current analysis approaches for large single cell data sets. The authors demonstrate in considerable detail that their computational approach keeps up with the ever-increasing scale and dimensional complexity of single cell data sets. The data are presented clearly and leave the reader convinced that this now tool is indeed an improvement to our toolbox that performs as expected. The amount of validation work that has gone into this manuscript is commendable and very impressive. The resulting computational framework is likely to represent a significant advancement to our ability to analyze large data sets.

Whether the considerable escalation in our ability to generate and analyze ever-larger single cell data sets has led to new biological insights at the same clip is not clear. The concentration of Kupfer cells around the CV in liver has been demonstrated via conventional CD68 stain.

With regard to the lineage relationships between islet cells in pancreas development, the analysis is focused on the lineage relationship of epsilon and delta cells. Epsilon cells make up less than 0.5% of islet endocrine cells in human. Delta cells are more prevalent and play some role in pancreas physiology, but are hardly the reason why the diabetes field has generated such a large amount of islet single cell data. The hypothesis generation of NeuroD2 as an important regulator of epsilon cells – largely absent in mouse pancreas - perhaps does not live up to the expectations raised by phrases such as ‘illustrate novel insight in to complex biological systems’. The use of the Ngn3 reporter mouse to enrich for Ngn3 expression has set the authors up to conduct a careful and detailed reconstruction of the lineage history of alpha and beta cells. Have the traditional developmental biology experiments from the 1990’s and 2000’s done such a good job in delineating the differentiation steps that these large new single cell data sets have nothing new to add – confirmatory or otherwise – to our understanding of alpha and beta cell development?

In this context, it is notably not discussed that between E14.5 and E15.5 there is a transition of alpha to beta cells that ceases between E15.5 and E16.5 (Fig. 5f). Could this be reminiscent of the non-zero rates of alpha to beta cell transdifferentiation that have been reported in healthy mice in the late stages of pancreas development.

The discussion speaks of epsilon cells, which suggests them to be a differentiated fifth endocrine islet cell type. This is not true for mice. Indeed, a recent paper by the same author group reporting a meta-analysis of 300,000 mouse islet cells did not detect epsilon cells as a distinct cluster. In human islets, there is a small but detectable population of ghrelin cells. This species difference was overlooked in the experimental design choice to validate the observation – made in mouse - that NeuroD2 is important in epsilon cell differentiation using a human ES cell system.

The implication that epsilon cells mature without going through a Fev positive state should be tested using the Fev-Cre line to demonstrate that islet cells at key perinatal ages are broadly Fev lineage positive, with the exception of ghrelin positive cells which at that early age are still detectable. This would represent an independent confirmation of the predicted lineage history of epsilon cells by moscot.

References 75 and 76 are co-cited several times and referred to ‘Yu et al., and ‘recent literature’ Ref. 76 is neither recent nor from Yu et al., More importantly, that paper (#76) is not a single cell transcriptome paper, but rather a careful direct accounting of the lineage relationship of ghrelin expressing cells using a relatively efficient (60%) ghrelin Cre driver. No lineage positive delta cells were observed in that study, which is at odds with the suggested lineage history shared by delta and epsilon in this manuscript.

The fact that NeuroD2 is induced by Neurog3 was established in PMID: 17924961. Notably, the deletion of NeuroD2 in that paper does not result in any discernable abnormalities in mouse pancreas development, which is a distinctly different from the suggestion made in this manuscript regarding human ES cell derived endocrine cells (Fig. 5h).

In contrast, deletion of NeuroD1 leads to a reduction in beta cells and prevents the increase in epsilon cells that results from Nkx2.2 deletion. Given that Neurod1/2 are similar and likely partially redundant, how are their observations different from the results described in PMID: 17988662 that already established a link between Neurod1/2 transcription factors and epsilon lineage decisions?

The data in Fig 5h should be presented as % GHRL+ relative to all cells in culture, and the transcript quantification in Fig. 5i should be normalized to the average control value to preserve the spread – and therefore the ability to assess for statistical significance under the presumption of equal variance.

The use of the word 'evolve' is best reserved for situations of actual evolution. 'Differentiate' or similar is more appropriate when speaking of the developmental transition between stages.

Referee #1

1. We want to thank the authors for carefully addressing almost all the points we raised in our previous review. In particular, they show 1) that their new framework, called moscot, which has a more computationally efficient implementation of WOT is also able to reproduce its results 2) they document the optimal transport consistency in time from figures from the literature, but also with new experiments with moscot. 3) they point out their figure 3b that shows how more spatial homogeneity improves the performance of moscot, but also that even in heterogeneous samples where moscot performs worse, it still outperforms the competitors (albeit by a little). However, this doesn't completely address the point about the consistency in assuming that biology introduces batch effects always via optimal transport. It is in fact very possible that batch effects introduced from different experimental protocols influence some cell types differently than others, or even that they cause flows (e.g. with non zero curl) which optimal transport could never capture. The authors should include the originally proposed benchmarking between three batches A,B,C and report the difference $P_{\{A,B\}}*P_{\{B,C\}}*P_{\{C,A\}}-Identity$. It is a useful benchmarking to have since it will showcase the upper limit of how good optimal transport is for batch effects.

We thank the referee for the comment. We agree with the referee that it is very possible that batch effects introduced from different experimental protocols influence some cell types differently than others. We thus set out to conduct the suggested analysis to evaluate the similarity between the identity mapping and the concatenation of three transport maps. To this end, we computed the transport matrices for 3 pairs of time point in the MOSTA dataset (mouse embryogenesis measured with stereo-seq technology), as suggested by the referee, and assessed the fraction of entries larger than the diagonal across rows for the resulting transport map of $P_{\{A,B\}}*P_{\{B,C\}}*P_{\{C,A\}}$. We would like to highlight that we perform entropy-regularized OT, and hence it is impossible to obtain the identity mapping since the smoothing effect introduced by the Sinkhorn algorithm discourages sparse structure, with many small but non-zero entries resulting in a more "diffuse" mapping, see e.g. (Peyré and Cuturi 2018). Yet, it is indeed desirable to obtain a stochastic map which is likely to map a cell onto itself. Hence, for each cell x we report the fraction of cells that are at least as likely to be mapped onto the cell x as the likelihood of the identity mapping, and take the average of this value across all cells (Demetci et al. 2022) (Response Fig. 1a). We also visualize the resulting transport map in Response Fig. 1.b. We observe that the result is robust with respect to the epsilon parameters used in practice, and clearly better than the outer (independent) coupling. In the ideal case, the resulting transport map would have the highest value on the diagonal. While the figure suggests that some diagonal structure is preserved, one should not overinterpret this result as proximity in the row/column number recapitulates proximity in the spatial y-axis but not in gene expression space. In fact, cells are ordered by their spatial locations (Response Fig. 1.c), and while for this particular dataset cells that co-localize in space are partially also similar in gene expression, the association is only partial. Note that this limitation can be overcome by using distributional metrics as was done in the first round of revision, in response to Referee 1, major point 2. Another way to make

the result more interpretable is aggregation to cell type level, which we did in response to Referee 1, major point 2, first revision, where we perform a consistency analysis for the spatiotemporal problem, but discretized at the cluster level.

Response Fig. 1 | Analysis of the deviation to the identity of the resulting transport matrix as $P_{\{A,B\}} \cdot P_{\{B,C\}} \cdot P_{\{C,A\}}$ for the time points 9.5-10.5-11.5 of the mouse embryogenesis spatial transcriptomics dataset. a. Fraction of entries larger or equal than the diagonal in the resulting transport matrix of $P_{\{A,B\}} \cdot P_{\{B,C\}} \cdot P_{\{C,A\}}$, at increasing values of epsilon. For large epsilon values like 10, due to a strong effect of the entropy regularization term, the resulting transport matrix further diverges from the practically desired, but theoretically invalid identity map. **b.** Resulting transport matrix for epsilon=0.05. **c.** Spatial scatterplot of the mouse embryo at time point E9.5 colored by index, visualizing the ordering of the index for the transport matrix in b).

We also show the robustness of moscot with respect to batch effects on the benchmark of spliced/unspliced RNA as the data was generated in different batches (Response Fig. 9). Moreover, the drosophila dataset has strong batch effects (Supplementary Fig. 24), and moscot still yields biologically meaningful results.

- Moreover, We feel that it is important to emphasize on the complexity of trajectory inference in the discussion over the contradictory lineage hypotheses the authors had come up with between previous cellrank paper and this paper (i.e. from Ngn3+ to epsilon and delta cell states), acknowledging that the conclusions regarding lineages are highly dependent on the integrated level of data modalities and the methods used for embedding construction and pseudotime inference. This could be more highlighted along the direction of this statement “While this hypothesis requires further experimental validation, our results demonstrate the potential of moscot to generate novel insights into complex biological systems” in the first paragraph of discussion.

We agree with the referee that it is important to highlight the limitations of our method. In particular, additionally to what we wrote in the discussion, we discuss further restrictions of moscot’s prediction about the identified lack of expression in hypothesized ancestors of epsilon cells, see e.g. response to referee 5, point 6 and response to referee 2, point 3. We also agree with the referee that different trajectory inference methods predict different lineage relationships. Yet, results displayed in Response Fig. 2 (included as Supplementary Fig. 50) show that some methods predict biologically implausible results. For example, there is strong evidence that the majority of *Fev*⁺ alpha cells develop into alpha cells, and the

majority of *Fev*⁺ beta cells develop into beta cells (Bastidas-Ponce et al. 2019; Byrnes et al. 2018). With respect to this, CytoTrace yields poor results, and veloVI's prediction for *Fev*⁺ alpha cells is poor while its predictions for *Fev*⁺ beta cells is correct. We now help the reader by labeling the trajectory inference methods with a green, orange, or red dot, depending on their fate probabilities computed by CellRank 2 (Weiler et al. 2023) on the well understood alpha and beta lineages. In effect, we consider the fate probabilities of alpha, beta, *Fev*⁺ alpha, and *Fev*⁺ beta cells, with alpha, beta, delta, and epsilon being possible lineages. We consider a transition to be correct if the highest fate probability is the biologically most likely one (i.e. alpha -> alpha, *Fev*⁺ alpha -> alpha, beta -> beta, *Fev*⁺ beta -> beta). Finally, we assign a method a green dot if all of the four transitions are correct, an orange dot if three out of four transitions are correct, and otherwise a red dot (Methods, 2.1.4):

To assess the reliability of the trajectory inference methods, we assess their performance by focusing on the well studied fate probabilities of the alpha and beta lineage. In effect, we consider the fate probabilities of alpha, beta, *Fev*⁺ alpha, and *Fev*⁺ beta cells, with alpha, beta, delta, and epsilon being possible lineages. We consider a transition to be correct if the highest fate probability is the biologically most likely one (i.e. alpha \rightarrow alpha, *Fev*⁺ alpha \rightarrow alpha, beta \rightarrow beta, *Fev*⁺ beta \rightarrow beta). Finally, we assign a method a green dot if all of the four transitions are correct, an orange dot if three out of four transitions are correct, and otherwise a red dot.

This results in labels of high quality only for DPT, MultiVelo, and moscot.

Regarding the dependency on the data modality, we would like to refer the referee to the response to the next comment.

3. In Suppl. Fig. 48, on testing other pseudotime methods, it would be better to specify which tools output a lineage structure consistent with what you report from moscot; In particular, to clearly compare with previous cellrank based hypothesis and conclude whether the difference mainly comes from integrating more modalities (transcriptome vs. transcriptome + epigenome) or the different models.

We thank the referee for this remark. We agree that we should help the reader interpret the results with respect to similarity to moscot's predictions. Therefore, we compute the Pearson Correlation coefficient between the aggregated fate probabilities, and add them to the figure, see Response Fig. 2. We added a description to Methods, 2.1.4:

To assess the consistency of a model's predictions with moscot's predictions, we compute the Pearson correlation coefficient between aggregated fate probabilities as output by CellRank.

With respect to the influence of the modality, we are limited to comparing methods which can be run on both gene expression and ATAC modalities independently. Thus, among the considered methods, it is only possible to run CytoTrace, connectivity, diffusion pseudotime, and moscot. Supplementary Figure 28 shows that moscot's predictions are robust, also with respect to the modality. The predictions of the connectivity and CytoTrace kernel are poor (see labeling). We thus perform further analysis on the underlying modality with diffusion pseudotime (DPT). We run DPT on three different graphs, corresponding to a graph

constructed from i) RNA only, ii) ATAC only, and iii) RNA and ATAC combined.

Response Fig. 4 highlights that the results are i) of high quality (assessed on alpha and beta lineages, see above), and ii) are highly similar (measured with the Pearson correlation coefficient of aggregated fate probabilities). This suggests that the difference in predictions originate mainly from the model rather than from the number of modalities incorporated.

Response Fig. 2 | **Analysis of the pancreatic endocrinogenesis dataset with different trajectory inference methods** → **Included as Suppl. Fig. 50**

Trajectory analysis of the pancreatic endocrinogenesis dataset with commonly used TI methods (Methods). We use CellRank(Lange et al. 2022; Weiler et al. 2023) to compute fate probabilities to the four endocrine cell states alpha, beta, delta and epsilon for each single cell followed by aggregation to cell type level (Methods) based on **a.** diffusion pseudotime (Haghverdi et al. 2016), **b.** scVelo (Bergen et al. 2020), **c.** VeloVI (Gayoso et al. 2024), **d.** MultiVelo (Li et al. 2022), **e.** CytoTrace (Gulati et al. 2020), and **f)** connectivity of the graph(Lange et al. 2022). We use moscot's RealTimeKernel to predict compute aggregated fate probabilities (**g**), highlighting these are outputs from CellRank and hence are different from cell type transitions computed directly with moscot. The quality of the inferred trajectories are assessed based on the alpha and beta lineage (Methods), while the Pearson correlation quantifies the similarity with moscot's predictions.

Response Fig. 3 | **Fate probabilities of the pancreatic endocrinogenesis dataset predicted with different trajectory inference methods** → Included as Suppl. Fig. 51

Trajectory analysis of the pancreatic endocrinogenesis dataset with commonly used TI methods (Methods). We use CellRank(Lange et al. 2022; Weiler et al. 2023) to compute fate probabilities to the four endocrine cell states alpha, beta, delta and epsilon (Methods) based on **a.** diffusion pseudotime (Haghverdi et al. 2016), **b.** scVelo (Bergen et al. 2020), **c.** VeloVI (Gayoso et al. 2024), **d.** MultiVelo (Li et al. 2022), **e.** CytoTrace (Gulati et al. 2020), **g.** connectivity of the graph (Lange et al. 2022), and **f.** moscot. The quality label is inherited from Supplementary Fig. 50.

Response Fig. 4 | Fate predictions of diffusion pseudotime are stable with respect to underlying modality → Included as Suppl. Fig. 53

a) Aggregated fate probabilities of diffusion pseudotime (Haghverdi et al. 2016) computed with CellRank. The quality is assessed based on the alpha and beta lineage (Methods), and correlations are computed between aggregated fate probabilities (Methods). **b)** Fate probabilities computed with CellRank based on diffusion pseudotime predictions incorporating different modalities.

4. We also would like to make the below suggestions: Showing the trajectory directionality diagram (using cellrank kernel projection method) for moscot like shown for other methods in Supplementary fig. 48.

As pointed out in (Weiler et al. 2023), the use of stream embedding plots for the RealTimeKernel is not advised. The authors write in the Methods section of their preprint:

“Previously, the most popular approach for visualizing RNA velocity has been the projection of the high-dimensional vector field onto a low-dimensional latent space representation¹³. With CellRank 2, we generalize this concept to any kernel based on a k-nearest neighbor graph, i.e., the PseudotimeKernel, CytoTRACEKernel, and VelocityKernel.”

Briefly, streamline embeddings project a high-dimensional transition matrix onto a 2D embedding space. Implicitly, they assume that the high-dimensional transition matrix has been computed based on a similar k-NN graph as the 2D embedding. While that is the case for most CellRank 2 kernels, it is not the case for the RealTimeKernel, which receives OT-based coupling matrices from moscot. Accordingly, when plotting the stream embedding for the RealTimeKernel with CellRank 2, the software outputs the warning:

```
WARNING: RealTimeKernel[n=11645, threshold='auto', self_transitions='all'] is not a kNN based kernel. The embedding projection works best for kNN-based kernels
```

Indeed, Response Fig. 5 shows that the resulting stream embedding plot does not yield meaningful results. Instead, we recommend plotting the fate probabilities and considering the aggregated fate probabilities as now added in Response Fig. 2 and Response Fig. 3. We would like to emphasize that in the other parts of the pancreas dataset analysis we solely rely on moscot’s downstream functions. For the sake of comparability we use CellRank 2’s interface with moscot for this use case.

Response Fig. 5 | **CellRank 2 recommends not using the velocity projection plot for the RealTimeKernel**

As described in CellRank 2 (Weiler et al. 2023), it is not recommended to plot the stream embedding of the RealTimeKernel.

5. Running the other pseudotime methods on the Bastidas-Ponce et al. dataset to see if the inferred branchings remain consistent with moscot results.

We ran *scVelo* (Bergen et al. 2020), *veloVI* (Gayoso et al. 2024), *DPT* (Haghverdi et al. 2016), *CytoTrace* (Gulati et al. 2020) and *dynamics based on connectivity* (Lange et al. 2022) on time points E12.5, E13.5, E14.5, and E15.5, and computed the fate probabilities with CellRank 2 (Weiler et al. 2023), see Response Fig. 6. We apply the same labeling system to indicate the reliability of the results of each method, see response to comment 2. Due to the poor performance of the methods, we do not explicitly study the consistency with moscot's results.

Response Fig. 6 | **Predictions of fate probabilities for time point E12.5, E13.5, E14.5, and E15.5 on the pancreatic endocrinogenesis dataset published in** (Bastidas-Ponce et al. 2019)

Results of scVelo (Bergen et al. 2020), veloVI (Gayoso et al. 2024), DPT (Haghverdi et al. 2016), CytoTrace (Gulati et al. 2020) and dynamics based on connectivity (Lange et al. 2022), and computation of fate probabilities with CellRank 2 (Weiler et al. 2023). The data was filtered to endocrine cells and its progenitors. Labeling is described in response 2.

6. Changing Supp. fig 31 with better labeling and structure to clearly indicate which states are ancestral and which states are descendant

We thank the referee for the suggestion to make the figure easier to read. With the updated figure and caption in Response Fig. 7, we hope to clarify that we consider cells at a time point and visualize the distribution of cell types of their ancestors.

Response Fig. 7 | **Ancestors of refined cell types obtained by moscot.time → Included as Suppl. Fig. 33**

(a) Cells at E15.5 aggregated to cell types and their ancestors in E14.5 aggregated to cell types and (b) cells at E16.5 aggregated to cell types and their ancestors in E15.5 aggregated to cell types. Only ancestries with a probability of at least 0.05 are visualized.

Referee #2:

I appreciate that the authors' efforts addressing my questions. The revised version added new figures and make a new biology points that NEUROD2 is an important factor for Epsilon differentiation and validated using previous generated KO NEUROD2 human iPSCs. However, I have some questions regarding the newly added data.

1. P-value/ statistical significance is missing in Figure 5h & I and Suppl. Fig. 43 NEUROD2 KO experiments.

We thank the referee for this important remark. P-values have been added to the mentioned graphs, the computational details of which were added to Methods 2.1.6, and a summary of which we added in Response Fig. 8 showing P-values obtained from one-way ANOVA tests of 0.0004 and 0.001 between control and NEUROD2 knockout clones for immunostaining, while the P-value between control and knockout samples are 0.0041 and 0.0008.

Response Fig. 8 | **Summary figure of functional experiments including statistical significance analysis**

a) Adapted from Figure 5, panel h), i): **h.** [...] quantification of GHRL-expressing cells in the control and NEUROD2 knockout (C37, C89) SC-islets at stage 6, day 14 (S6.14, Methods). White arrowheads indicate GHRL+ cells. Scale bar, 50 μ m. $n = 4$ independent experiments. **i.** qPCR analysis of expression levels of GHRL at S6.14. Data are represented as mean and standard deviation (Methods). **b)** Adapted from Supplementary Figure 47: **a.** Insulin mean intensity and the number of SST-positive cells measured with immunostaining (Methods) for control, clone 37 and clone 89⁹⁸. **b.** Relative mRNA expression of *Ins2*, *SSt*, *HHEX*, and *GCG* measured with qPCR. We report mean and standard deviation (Methods).

2. What is the dynamic lineage expression pattern of NEUROD2 during mouse development in your data? The timing of the NEUROD2 expression is important for understanding its function on lineage specification for delta and the epsilon cells.

We agree with the referee that the dynamic lineage expression pattern of *Neurod2* is relevant for assessing its role during endocrinogenesis. We would like to direct the referee to Supplementary Fig. 40, where *Neurod2* expression is i) quantified per cell type and ii) visualized on a UMAP embedding. In particular, we observe a transient expression of *Neurod2* mainly in *Ngn3* high, *Ngn3* high cycling, epsilon progenitors, *Fev*+ delta 0 and *Fev*+ delta 1 cells. We further visualize these results per time points in Response Fig. 16 (included as Supplementary Figure 45), where we also added the expression pattern of *Neurod2* in time points E12.5 and E13.5 using the dataset published in (Bastidas-Ponce et al. 2019), which is consistent with what we observe in our dataset. We refer to the supplementary figure showing the dynamic lineage expression from the manuscript:

The dynamic lineage expression of *Neurod2* is notable in the epsilon progenitor and *Fev*+ delta populations (Supplementary Fig. 45)

We further discuss the expression pattern of *Neurod1* and *Neurod2* in response 9 to referee 5. Given our hypothesis of epsilon cell formation, we now explicitly motivate the naming of this population while highlighting this being due to a prediction:

Moreover, the majority of the epsilon population is mainly derived from a population which we refer to as epsilon progenitors, which themselves we predict to originate from the *Ngn3*^{high} endocrine progenitors (Supplementary Fig. 33).

3. NEUROD2 has been described “to be the second most relevant transcription factor for both the *Fev*+ delta and the epsilon progenitor population” here. However, figure 5h shows NEUROD2 KO can only affect epsilon but not delta differentiation. How to explain the discrepancy?

We thank the referee for this question. Indeed, as shown in Figure 5g and Supplementary Figure 37 the gene expression profiles and ATAC profiles of *Fev*+ delta cells and epsilon progenitor cells are very similar. Thus, the predicted ancestry populations are very similar, which can also visually be seen from Supplementary Figure 31. As the ancestry population predicted by moscot is very similar for *Fev*+ delta cells and epsilon progenitors, it is likely that the set of features which is highly correlated with the pull-back distribution is similar for

both cell types. Thus, the discrepancy stems from the similarity of cellular states of epsilon progenitor cells and *Fev*⁺ delta cells in gene expression and ATAC profile while being regulated from a (partially) different set of transcription factors. In fact, it is important to highlight that moscot gives candidates for driver genes, thus we explicitly replaced the term “driver genes” by “potential driver genes”, i.e. the text now reads

[...] To learn more about the regulatory mechanisms driving delta and epsilon cell fate we used moscot.time to find potential driver genes.

4. Are the cells between control vs C37, C89 derived from same iPSC lines?

Yes, both clones were obtained by targeting a single clone of the heterozygous C-peptide-mCherry reporter hiPSC line (HMGUi001-A-8).

Referee #3:

I greatly appreciate the authors' efforts to address important issues I raised properly. However, I found the following issues that remain to be addressed:

1. The authors seem to completely ignore several suggestions I made in my referee comments, including for example, referee comments 4, I mentioned the possibility to use unspliced /unspliced RNA from the same cell as a way to validate the ground-truth mapping of Moscot. However, this was ignored (I agree the multi-modality mapping is not needed given the focus of this work).

We apologize for not having considered this suggestion so far (which we believed out of focus), and thank the referee for insisting - it turned out to be an interesting task: We now included a benchmark of the performance of moscot on aligning cells based on spliced and unspliced counts. We utilize the pancreas dataset which is published with this work. Response Fig. 9 shows that this is indeed a non-trivial task as the two distributions completely separate on the UMAP, and the expression of genes highly varies across spliced and unspliced gene expression counts. Yet, moscot is able to align the cells, with only a small drop in performance when using the low-rank approximation. At the same time, the influence of the rank is stable until it reaches a very low value. We perform this analysis on different subsets of the dataset to study the influence of the number of cells included in the dataset as suggested by the referee, include it in the manuscript (Supplementary Figure 10), and refer to it in the main text as

Moreover, we found the performance of moscot mapping spliced to unspliced RNA counts to be robust with respect to its rank (Supplementary Fig. 10).

Response Fig. 9 | **moscot's performance on aligning cells from spliced/unspliced counts is robust with respect to low-rank approximations** → Included as Suppl. Fig.

a) UMAP embeddings of different subsets (top: E16.5, middle: E15.5, bottom: E14.5+E15.5+E16.5, i.e. full dataset) of the pancreas dataset published in this manuscript. Cells are split into spliced and unspliced counts, resulting in two distinct populations of cells in which a known one-to-one match exists. Normalized gene expression of relevant marker genes in the mouse pancreas is visualized to demonstrate the difference of gene expression of spliced and unspliced RNA counts, highlighting this to be a non-trivial task. **b)** Fraction of cells closer (or equally close) to their true match (FOSCTTM(Demetci et al. 2022)) score. Given a single cell, the FOSCTTM score reports the fraction of cells which are predicted to be equally or more likely to be matched with the given cell than the true match of the given cell. A value of 0 denotes a perfect match, while a value of 1.0 is as good as the outer (independent) coupling. We report the mean across all cells.

2. The author used Scanpy's code to benchmark the calculated apoptosis and proliferation rates instead of "comparison with simpler methods as outlined in the original WOT or the PRESCIENT". My guess is that Scanpy implemented the rate calculations from these tools. Even though this is true, it is worthwhile to explicitly mention that.

We decided to use Scanpy to compute cellular growth rates as it is very commonly used in the community for this task (Cockburn et al. 2022; Wang et al. 2021; Bastidas-Ponce et al. 2019). However, the routine implemented in Scanpy is very similar to WOT and we used the same gene set (Tirosh et al. 2016) as suggested in the original WOT publication (Schiebinger et al. 2019). Both approaches compare the expression of genes in the supplied gene set with the expression of other genes, and aggregate across genes to compute a single value per cell.

The method implemented in Scanpy follows the original suggestion in Seurat v1 (Satija et al. 2015): it averages over genes in the supplied gene set and normalizes with the average expression of a reference set of genes. Similarly, WOT z-normalizes the expression values of genes in the gene set and averages over z-normalized expression values to obtain a score per cell. PRESCIENT (Yeo, Saksena, and Gifford 2021) follows the WOT approach, but uses a different gene set derived from the KEGG_CELL_CYCLE term.

We added a sentence to the Methods (Section 2.1.1, paragraph "Correlating predicted growth rates with gene-set based growth rates") to better explain the growth-rate score computation:

To validate predicted growth rates, we correlated them with cell-cycle scores computed based on marker genes using scanpy through `scanpy.tl.score_genes`. **Briefly, the Scanpy (Wolf, Angerer, and Theis 2018) implementation of gene scoring follows the original suggestion in Seurat v1 (Satija et al. 2015): it averages over genes in the supplied gene set, normalized by the average expression of a reference set of genes.** For this comparison, we initialized marginals uniformly so that our algorithm was not aware of growth rates and we could use this information for validation.

3. Some of the claims / arguments from the author are not sound and convincing, including: for referee comment 5 about the usefulness of single cell level mapping, the authors responded by saying that 1. Moscote and TOME achieve similar results 2.

Moscot is more robust comparing to TOME for batch effects. These are fine results, but they don't answer my questions about the need, superiority of single cell level mapping over meta-cell or cell group level mapping.

In our previous response to the referee's comment, we compared moscot with TOME (a cluster-level mapping approach) across three datasets and demonstrated that (1) moscot.time is more robust to batch effects, (2) moscot.time's growth rates correlate better with scanpy-derived growth rates and (3) moscot.spatiotemporal attains better mapping accuracy. However, we indeed did not explicitly compare running moscot on the single-cell vs. metacell level to evaluate the need and superiority of single-cell level mappings, thank you for this suggestion. We now include the additional analysis in our manuscript in a new Supplementary Figure 9, and in this letter in Response Fig. 10.

As an example, we used E9.5 of the mouse embryogenesis data (C. Qiu et al. 2022) of Fig. 2 and computed metacells using the popular Metacell-2 algorithm (Ben-Kiki et al. 2022). We found that the algorithm could not resolve rare Primordial germ cells (PGCs; 30 cells or 0.03% of the population); these were split across several metacells and mixed with other cell types (Response Fig. 10a,b). Consequently, any mapping based on these metacells would not be able to provide insights on the ancestors or descendants of PGCs.

We also compared moscot's mapping accuracy on the single-cell vs. metacell level for the E10.5-11.5 pair of time points. While global mapping accuracy in terms of the germ layer and curated cell type transition scores of Fig. 2c was only slightly worse on the metacell level (Response Fig. 10c), we found that metacell mapping yielded much lower correlations with known driver genes for Pancreatic epithelium (Response Fig. 10d), another rare population in our dataset (134 cells at E11.5 or 0.03% of the population).

From these examples, we conclude that single cell-level resolution can be important to resolve rare cell states. We added the following text to the manuscript to summarize the additional analysis:

As an alternative to single-cell mapping, we also explored metacell aggregation (Ben-Kiki et al. 2022) prior to moscot mapping. While performance was similar in terms of germ layer (i) and cell type (ii) scores at E10.5-11.5, metacells could not uniquely resolve rare Primordial germ cells (PGCs) at E9.5 and metacell mapping yielded lower driver-gene correlations for Pancreatic epithelium (Supplementary Fig. 9 and Methods).

Response Fig. 10 | **Metacells do not resolve PGCs and metacell mapping degrades driver gene correlation for Pancreatic epithelium** → Included as Suppl. Fig. 9

a. UMAP of E9.5 cells, visualizing individual cells (small dots) and metacells (large dots) computed using Metacell-2 (Ben-Kiki et al. 2022) (Methods). Colors indicate PGCs and cell types that co-occur in metacells with PGCs. The zoom-in highlights PGCs, which are not captured by any metacell. **b.** Bar chart over cell-type composition for the six metacells at E9.5 containing most PGCs. No metacell received the “PGC”-label because they are dominated by other cell types. **c,d.** Comparing moscot mapping at E10.5-11.5 on the single cell versus metacell levels in terms of the curated transition and germ layer scores (**c**) and correlation between Pancreatic epithelium ancestor probabilities and known driver gene expression (**d**; Methods).

- for referee comments 9, the authors considered the first two suggestions are problematic because "the result is more influenced by the density of cells rather than

uncertainty in many scenarios". However, the authors had already analyzed the whole-body drosophila data, they should use this dataset to demonstrate some of these points to confirm their claims as this dataset has no sampling issue and estimation of cell density is ideal.

We thank the referee for the comments and apologize for the confusion. We would like to clarify that our answer to the referee's comment 9 refers to the density in gene expression space, and not in spatial coordinate space, when stating "the result is more influenced by the density of cells rather than uncertainty in many scenarios". We apologize for the phrasing "suggestions being problematic", and now demonstrate in Response Fig. 11 that biologically meaningful plasticity cannot directly be inferred from the transport matrix (i.e. answering to comment 9 by the referee in the first round of revision: "A wide mapping spectrum of cells"). We measure the "mapping spectrum" with the entropy of the map for a single cell x (with a

slight abuse of terminology) $H(Y|X = x) = - \sum_{y \in Y} p(y|x) \log(p(y|x))$ where p denotes the density of the transport plan (i.e. the entries of the transport matrix), x denotes a cell in the source distribution, and Y the set of cells in the target distribution. This gives a notion to what extent a cell has a single match with a high probability (low entropy) as opposed to multiple matches with lower probability (high entropy). While this notion of uncertainty is of interest when trying to find exact matches of single cells, e.g. when translating modalities, a meaningful plasticity of a cell (plasticity in terms of developmental biology) can be better described with an uncertainty metric considering the feature space (as opposed to only consider the transport matrix). The conditional variance for a single cell x (with a slight abuse

of terminology) $Var(Y|X = x) = \sum_{y \in Y} \left(y - \sum_{y' \in Y} y' p(y'|x) \right)^2 p(y|x)$ gives a notion of uncertainty about the trajectory (e.g. in gene expression space), and hence correlates with plasticity. Here, x denotes a cell in the source distribution, Y the set of cells in the target distribution, and p the density of the transport plan (i.e. the entries of the transport matrix).

We demonstrate this in Response Fig. 11 for both pairs of time points E14.5/E15.5 and E15.5/E16.5 in the pancreas dataset. The conditional entropy is high whenever we have many cells from the same cell type (e.g. Response Fig. 7a shows there is high conditional entropy in beta cells, although the plasticity of beta cells is very low as it is a differentiated cell type). We quantify this by computing the Spearman correlation coefficient between the conditional entropy and the mean distance to the 30 nearest neighbors, which is a measure of inverse density. We obtain a Spearman correlation coefficient of -0.55 and -0.65 for E14.5 and E15.5, respectively. This highlights that there is a strong negative correlation between conditional entropy and mean distance to the nearest neighbors, which means there is a strong positive correlation between uncertainty and density. This result is what we were referring to when speaking about "the result is influenced by the density of cells" in the previous response letter. The Spearman correlation between mean neighborhood distance and conditional variance (in PCA space) is 0.45 (E14.5) and 0.17 (E15.5), while giving more meaningful values of plasticity. For example, the conditional variance in PCA space is very low in fully differentiated alpha and beta cells, which is biologically very plausible. More interestingly, we can compute the conditional variance for any subset of features, and hence investigate the uncertainty with respect to a gene. Response Fig. 7b shows that, according to our model, the uncertainty with respect to *Ngn3* expression is almost zero for all cells

downstream *Ngn3* high cells, which is to be expected as *Ngn3* won't be expressed any more. Similarly, our model predicts that cells in *Ngn3* low are certain with respect to their future *Ins2* expression (due to a constant low expression of *Ins2*), while our model predicts the uncertainty of fully differentiated beta cells to be also relatively low (due to a relatively constant high *Ins2* prediction).

The same line of thought applies to the case of the drosophila embryo. We computed conditional variances for gene expression variation, embedded by PCA. As can be seen in Response Fig. 12a,b no clear pattern emerges. This is to further stress that such score is only meaningful when the phenomenon of interest (gene expression variability as a proxy for tissue differentiation), is associated with the same experimental sampling time. For the drosophila embryo case, we think that the measured time points are too far spaced in time for gene expression variability to reflect any type of plasticity. As highlighted above, while the cell density is ideal with respect to the spatial coordinates, that does not mean that it is ideal with respect to the phenomenon of interest (embryo development). Interestingly, when we compute the conditional variance for two key TFs (*Rbp6* associated with CNS tissue development, and *Mef2* associated with Muscle tissue development, Response Fig. 12c,d), there appear to be an association, but this is not surprising since these genes are specifically expressed in those tissues, and contribute to the gene regulation of tissue differentiation.

We would like to note that we believe we had misinterpreted the initial comment 9 by the referee, which we interpreted as a question whether we can directly infer insights from the transport matrix regarding plasticity. Instead, we now believe the referee was asking whether we could use moscot directly to infer biological factors, e.g. driver genes. We would like to highlight that all of the downstream results in figure 5 are solely based on moscot, and not on CellRank. Results in Figure 4d-g) are based on moscot's interface with CellRank, highlighting the interoperability between CellRank and moscot. Moreover, thanks to the referee's comment, we had added an analysis on the MOSTA dataset which is only based on moscot to identify target genes. I.e., in the last round of revision we wrote in response to comment 9

“Moreover, we added a method to identify target genes, which is, for example, not possible to do with CellRank.”

Response Fig. 11 | **Conditional variances are more meaningful than conditional entropies when considering plasticity of single cells**

a) Conditional entropy of each single cell for moscot.time on the pancreas dataset. Values for cells in E14.5 based on the 14.5/15.5 coupling are displayed on the left hand side, while

values for cells in E15.5 based on the 14.5/15.5 coupling are displayed on the right hand side . **b)** Conditional variance of each single with respect to the PCA representation of gene expression (top) as well as with respect to *Neurog3*, *Ghrl*, and *Ins2*.

Response Fig. 12 | **Conditional variances for gene expression (PCA) and two key Transcription Factors for CNS and Muscle tissue of the drosophila embryo E14-16 and E16-18 measured with stereo-seq technology.**

a) Conditional variance for PCA in gene expression space, proxy for plasticity of cells. **b)** Conditional variance for PCA aggregated by tissue type.

c) Conditional variance for *Rbp6* expression, a key TF associated with CNS tissue development (Suppl. Figure 22), aggregated by tissue type. **d)** Conditional variance for *Mef2* expression, a key TF associated with Muscle tissue development (Suppl. Figure 22), aggregated by tissue type. Hemolymph tissue is missing since it's only measured in time point E16-18.

Referee #4:

Summary of the key results

This paper presents Multi-Omics Single-Cell Optimal Transport (moscot), a computational framework to solve mapping and alignment problems that arise in single-cell genomics. Compared to other methods for single-cell genomics relying on measurements, using a computational framework is not cell-destructive and can capture the whole molecular information. Compared to other computational frameworks for single-cell genomics, moscot supports multimodal data, is more scalable (because it relies on recent optimal transport techniques), and unifies previous applications of optimal transport to single-cell genomics in a single software.

In general, this software should be of great interest to the community, thanks to its scalability in the era of large datasets with temporal and spatial dimensions, and its ability to handle multimodal data, without experimentations which can be cell destructive. In particular, moscot could recover murine differentiation trajectories during embryogenesis across time and space, enriched spatial liver samples with multimodal information, and aligned brain tissue slides. One should be able to apply moscot to a wide range of problems in single cell genomics.

The paper presents the software and convincing use cases. The algorithmic behind moscot relies on recent advances in understanding and implementing optimal transport techniques. These optimal transport techniques have already been applied successfully in various fields of statistics and machine learning, and their application to single cell genomics makes perfect sense.

We thank the referee for the thorough summary and appreciation of our work.

1. I am not sure if uncertainty quantification when mapping and aligning or fitting trajectories is supported by the software. Could the authors clarify please?

We thank the referee for this interesting question.

- On a single-cell level we now consider the conditional entropy and the conditional variance, see response to referee 3, comment 4, and highlight the different use cases for both uncertainty metrics. While we have implemented the conditional entropy (e.g. for the TemporalProblem https://moscot.readthedocs.io/en/latest/genapi/moscot.problems.time.TemporalProblem.compute_entropy.html), we are currently finalizing the implementation of the computation of the conditional variance (link to pull request: <https://github.com/theislab/moscot/pull/712>, the link might not be accessible any more by the time of the referee trying to access it). We also considered sparse feature maps and assessed the uncertainty of a single cell based on the genes which

are relevant for its mapping compared to the set of genes which are relevant for the mappings of the cells in the neighborhood of a single cell, see Supplementary Fig. 41

- On an aggregated view, we consider uncertainty of mappings throughout the manuscript, for example when studying trajectories. For instance, in Figure 5f we consider the stochastic trajectories of cell types.

The conclusions drawn from the different use cases of moscot seem reasonable given the data in my possession. However, I am not an expert in single cell genomics to evaluate the originality of these conclusions compared to the literature.

Suggested improvements: experiments, data for possible revision

No suggestions.

References: appropriate credit to previous work?

2. SPATEO seems to be a directly related software and a competitor, I would have expected a citation and a comparison to this work from the beginning of the paper.

We thank the referee for this remark. We had cited SPATEO already, but in the revision now added a reference to the beginning of the paper

Concurrently to SPATEO (X. Qiu et al. 2022), we introduce the concept of spatiotemporal mapping, which involves the integration of spatial coordinates and gene expression data.

We would like to highlight that we already discussed SPATEO in the Discussion section of our paper. In the manuscript, we write

Parallel to moscot, Qiu et al. developed SPATEO(X. Qiu et al. 2022) which also uses OT to map cells across spatial time courses. However, SPATEO is less scalable: on the same MOSTA Stereo-Seq application(Chen et al. 2022), SPATEO required downsampling to 2,000 cells per embryo(X. Qiu et al. 2022) while moscot mapped entire embryos (500,000 cells).

Clarity and context: lucidity of abstract/summary, appropriateness of abstract, introduction and conclusions

In general, the paper is clear and well-written.

3. A conclusion would be appreciated (maybe expand a little bit the last paragraph of the main paper and turn it into a conclusion).

We would like to point the referee to our Discussion, where we i) summarize our work, ii) discuss SPATEO (see response above), iii) discuss other applications of optimal transport in single-cell genomics, and iv) give an outlook on extensions of moscot incorporating neural optimal transport methods. We have the impression that the Discussion set up this way serves as a conclusion, and we needed to be somewhat succinct given space limitations. Following the referee's suggestion, we renamed the section to 'Discussion & Conclusion'.

I recommend publication for this work. I believe this software unifies many methods and will be of interest to biologists looking to apply modern methods to their datasets with minimal knowledge of optimal transport techniques.

We thank the referee for their appreciation of our work.

Referee #5:

This manuscript describes and validates a computation framework called moscot that is capable of computationally assessing very large single cell transcriptome data sets for temporal, spatial and spatiotemporal data sets. The authors demonstrate that moscot computationally outperforms current analysis approaches for large single cell data sets. The authors demonstrate in considerable detail that their computational approach keeps up with the ever-increasing scale and dimensional complexity of single cell data sets. The data are presented clearly and leave the reader convinced that this new tool is indeed an improvement to our toolbox that performs as expected. The amount of validation work that has gone into this manuscript is commendable and very impressive. The resulting computational framework is likely to represent a significant advancement to our ability to analyze large data sets.

We thank the referee for the positive comments.

1. Whether the considerable escalation in our ability to generate and analyze ever-larger single cell data sets has led to new biological insights at the same clip is not clear. The concentration of Kupfer cells around the CV in liver has been demonstrated via conventional CD68 stain.

We thank the referee for the good point. Indeed, the regionalization of Kupffer cells (KC) around the Portal Veins of the Liver has been described before, and indeed also confirmed by the authors of the original publication of the CITE-seq data. The authors indeed used conventional staining of VSIG4, F4/80, FOLRB, and GLUL combined with Cd51/CD5L for murine KC (Figure 4, (Guilliams et al. 2022)). Nevertheless, we were able to show that with moscot we could reproduce this finding by using a dataset of large-scale image-based transcriptomics and enhancing it by imputing cell types and protein expression. We were then able to confirm the presence of Kupffer cells, and their regionalization around Portal Veins, by both inferred cell type as well as expression of the Folate Receptor Beta, measured in the original CITE-seq data, and imputed onto the Vizgen Merscope dataset thanks to moscot. While it is not a novel biological finding, we believe it showcases well the power of moscot to leverage large scale multimodal dataset to uncover spatial patterns of cellular variation in tissue. We report the passage in the main text below:

we could confirm their enriched presence in areas around PV where liver sinusoids are more prevalent⁴⁷

2. With regard to the lineage relationships between islet cells in pancreas development, the analysis is focused on the lineage relationship of epsilon and delta cells. Epsilon cells make up less than 0.5% of islet endocrine cells in human. Delta cells are more prevalent and play some role in pancreas physiology, but are hardly the reason why

the diabetes field has generated such a large amount of islet single cell data. The hypothesis generation of NeuroD2 as an important regulator of epsilon cells – largely absent in mouse pancreas - perhaps does not live up to the expectations raised by phrases such as ‘illustrate novel insight in to complex biological systems’.

We thank the referee for this remark and agree that insulin-producing beta cells and glucagon-producing alpha cells are the most important cell types for glucoregulation and have hence been studied in detail. For sure somatostatin-producing delta cells play an essential role in inhibiting secretion of alpha and beta cells and play an important role for islet physiology (Huisin et al. 2018). How important ghrelin-producing epsilon cells are for physiology is less clear, but it seems that across >80 million years of evolution from mice to human these cells are still generated during islet neogenesis and may have a function in satiety regulation (Tschöp, Smiley, and Heiman 2000). However, we agree with the referee and tone down our statement and adapt the sentence ‘illustrate novel insight into complex biological systems’ by ‘study complex biological insights’. In particular, we now write

While this hypothesis requires further experimental validation, our results demonstrate the potential of moscot to study complex biological systems.

Moreover, we adapted

Finally, we jointly profile gene expression and chromatin accessibility during mouse pancreatic development and apply moscot to formulate a novel hypothesis about cell trajectories of delta and epsilon cells.

to

Finally, we jointly profile gene expression and chromatin accessibility during mouse pancreatic development and apply moscot to better delineate cell trajectories of delta and epsilon cells.

3. The use of the Ngn3 reporter mouse to enrich for Ngn3 expression has set the authors up to conduct a careful and detailed reconstruction of the lineage history of alpha and beta cells. Have the traditional developmental biology experiments from the 1990’s and 2000’s done such a good job in delineating the differentiation steps that these large new single cell data sets have nothing new to add – confirmatory or otherwise – to our understanding of alpha and beta cell development?

We thank the referee for the statement, and indeed, the developmental biologists have done a great job in delineating the lineage trajectories of all endocrine subtypes in the islets of Langerhans, which our data confirms. However, along these lineage trajectories from endocrine induction to *Fev*⁺ endocrine progenitors to hormone-producing cell types we have now determined a very fine grained and highly resolved dynamic gene expression program showing the molecular machinery regulated over time. This allows us to identify new genes, such as NEUROD2, which we agree has a very interesting transient expression in endocrine progenitors but not the most profound phenotype. In summary, we have provided a detailed, time-resolved analysis of pancreatic endocrinogenesis, providing comprehensive lists of candidates for potential follow-up functional experiments. To tone down the importance of

the NEUROD2 finding and highlight the detailed, comprehensive study of our work, we now write:

We have provided a detailed, time-resolved analysis of pancreatic endocrinogenesis, providing comprehensive lists of candidates for possible follow-up functional experiments. To corroborate the potential of moscot's predictions, we experimentally verified the role of NEUROD2 for epsilon cell differentiation using a stem cell gene knockout and islet cell differentiation system.

4. In this context, it is notably not discussed that between E14.5 and E15.5 there is a transition of alpha to beta cells that ceases between E15.5 and E16.5 (Fig. 5f). Could this be reminiscent of the non-zero rates of alpha to beta cell transdifferentiation that have been reported in healthy mice in the late stages of pancreas development.

Good question; we now set out to explicitly study potential transdifferentiation between alpha and beta cells. Therefore, we computed cells which moscot predicts to potentially transdifferentiate from alpha to beta cells for both E14.5/E15.5 and E15.5/E16.5. While Response Fig. 13 shows that there are indeed cells co-expressing *Gcg* and *Ins2*, we could not find any evidence for transdifferentiation, e.g. via a higher expression of *Pax4* (Zhang et al. 2016) in the subset of alpha cells which moscot predicts to differentiate into beta cells. Instead, we believe that the small proportion of alpha cells which moscot predicts to differentiate into beta cells is due to noise in the cell type clustering procedure and/or transitions which moscot incorrectly predicts, e.g. due to the stochasticity of moscot's prediction of the fate of a single cell. Indeed, this is in line with the previous findings. (Thorel et al. 2010) have shown that only under extreme beta cell loss (>99% ablation) alpha cells transdifferentiate into beta cells. This is consistent with constitutive alpha cell lineage tracing, suggesting that alpha and beta cells differentiate from two independent lineages (Herrera 2000; Quiox et al. 2007). In response to the referee's comment, we added the following sentence to the manuscript:

[...] with slight differences being likely due to noise in cell type annotation, sequencing biases or limitations in the moscot algorithm.

b ■ Identified as potentially transdifferentiating from alpha to beta
■ Identified as not transdifferentiating from alpha to beta

Response Fig. 13 | No clear evidence for transdifferentiation between alpha and beta cells

a) Likelihood of transdifferentiation from alpha to beta cells at time point E14.5 (left, identified via transitions from E14.5 to E15.5), and at time point E15.5 (right, identified via transitions from E15.5 and E16.5). **b)** Comparison of gene expression between binary groups of predicted potentially transdifferentiating cells (orange), and predicted non-transdifferentiating cells (blue), for E14.5 (left) and E15.5 (right). The groups were built by thresholding likelihoods of ancestry at 0.1, resulting in 80/291 and 29/588 potentially transdifferentiating cells in E14.5/E15.5 and E15.5/E16.5, respectively.

5. The discussion speaks of epsilon cells, which suggests them to be a differentiated fifth endocrine islet cell type. This is not true for mice. Indeed, a recent paper by the same author group reporting a meta-analysis of 300,000 mouse islet cells did not detect epsilon cells as a distinct cluster. In human islets, there is a small but detectable population of ghrelin cells. This species difference was overlooked in the experimental design choice to validate the observation – made in mouse - that NeuroD2 is important in epsilon cell differentiation using a human ES cell system.

We apologize for this misunderstanding. We intend to state that epsilon cells are a transient existing cell population in the developing mouse pancreas and islet. Due to its hormone-producing role, we study its lineage formation. To prevent this misunderstanding, we adapted the text

We used moscot.time to compute putative ancestry and descendency relationships and found that mature endocrine cells mostly remained in their cellular state as expected (Supplementary Fig. 28, Supplementary Fig. 29 and Methods).

to

We used moscot.time to compute putative ancestry and descendency relationships and found that embryonic endocrine hormone-producing alpha, beta, and delta cells are predicted to mostly remain in their cellular identity as expected (Supplementary Fig. 28, Supplementary Fig. 29 and Methods).

Regarding the species difference, we would like to highlight that we focused on the developing mouse pancreas whose development we recently showed to be similar in humans and mice (Lickert et al. 2024). While we did not explicitly study the role of *Neurod2* in (Lickert et al. 2024), Response Fig. 14 shows that *Neurod2* is also expressed in a progenitor cluster of epsilon cells.

[REDACTED]

6. The implication that epsilon cells mature without going through a Fev positive state should be tested using the Fev-Cre line to demonstrate that islet cells at key perinatal ages are broadly Fev lineage positive, with the exception of ghrelin positive cells which at that early age are still detectable. This would represent an independent confirmation of the predicted lineage history of epsilon cells by moscot.

We thank the referee for this remark. Unfortunately, it is not possible for us to conduct experiments on the *Fev-Cre* line as, first, we do not know if this mouse model is generated, and second, if it was available, it would take months to import and many more months to analyze the results. Instead, we employ diffusion pseudotime (Haghverdi et al. 2016) to predict the lineage specification of epsilon cells Response Fig. 15, and specify our finding to

Contrary to our recent hypothesis (Bastidas-Ponce et al. 2019) the epsilon progenitor population has a low mean expression of *Fev*, implying a relatively immediate expression of *Ghr1* following *Fev* (Supplementary Fig. 32). While we corroborate this hypothesis with independent computational methods (Supplementary Fig. 35), experimental validation of this claim is necessary.

Response Fig. 15 | *Fev* expression over pseudotime per lineage → Included as Suppl. Fig. 35

a) Normalized expression of *Fev* over pseudotime (Haghverdi et al. 2016) computed with *cellrank.pl.gene_trends* building on CellRank's pseudotime kernel. **b)** Normalized gene

expression of *Fev* and each islet hormone for the respective lineage plotted over pseudotime.

7. References 75 and 76 are co-cited several times and referred to 'Yu et al., and 'recent literature' Ref. 76 is neither recent nor from Yu et al., More importantly, that paper (#76) is not a single cell transcriptome paper, but rather a careful direct accounting of the lineage relationship of ghrelin expressing cells using a relatively efficient (60%) ghrelin Cre driver. No lineage positive delta cells were observed in that study, which is at odds with the suggested lineage history shared by delta and epsilon in this manuscript.

We thank the referee for noticing this mistake, which we apologize for. We assume the referee is alluding to this sentence of the manuscript:

Similarly, while our previous analysis suggested that epsilon cells can be generated from alpha and *Ngn3*⁺ cells, Yu et al. (Yu et al. 2021; Arnes et al. 2012) hypothesized that epsilon cells derive exclusively from *Ngn3*⁺ cells, and can in turn give rise to alpha cells.

We adapted the statement to

Similarly, while our previous analysis hypothesized that epsilon cells can develop from both *Ngn3*⁺ progenitors and glucagon producing alpha cells³⁸, lineage tracing experiments confirmed that ghrelin-producing epsilon cells are not a terminal state and can in turn give rise to alpha and PP cells and rare beta cells⁸⁴.

Moreover, we apologize for the mistake in the paragraph

In particular, we find that epsilon cells partly evolve into alpha cells (Fig. 5f and Supplementary Fig. 30), which has been reported in recent literature (Arnes et al., 2012, Yu et al., 2021)

which we adapted to

In particular, we find that epsilon cells partly evolve into alpha cells (Fig. 5f and Supplementary Fig. 32), which has been reported in the literature (Arnes et al., 2012, Yu et al., 2021).

Regarding the comment that no lineage positive delta cells were observed, we would like to point out that this is in line with our findings, as we predict delta cells to be derived from a *Fev*⁺ delta state, rather than ghrelin-expressing epsilon cells.

8. The fact that NeuroD2 is induced by Neurog3 was established in PMID: 17924961. Notably, the deletion of NeuroD2 in that paper does not result in any discernable abnormalities in mouse pancreas development, which is a distinctly different from the suggestion made in this manuscript regarding human ES cell derived endocrine cells (Fig. 5h).

We thank the referee for pointing us to this relevant work. Indeed, the authors state that “[...] NeuroD2 null mice exhibit normal islet cell differentiation”. Yet, the authors only consider alpha, beta, delta, and pancreatic polypeptide cells as islet cells. Indeed, they state that “The differentiation and distribution of cells expressing insulin, glucagon, somatostatin, and pancreatic polypeptide was similar to that of wild-type mice (Fig. 5, and data not shown).” Thus, their results are in line with the results we find, which we also reported previously (Cota et al. 2023), and are displayed in Fig. 5h, and Supplementary Fig. 43., which we now explicitly mention in the main text citing the work mentioned by the referee:

This analysis suggests a role for NEUROD2 in directing human epsilon cell differentiation. At the same time, our previous (Cota et al. 2023) and current data indicate no function of NEUROD2 in the specification of alpha, beta, and delta cells, which is in line with what has been reported in mouse. (Gasa et al. 2008)

9. In contrast, deletion of NeuroD1 leads to a reduction in beta cells and prevents the increase in epsilon cells that results from Nkx2.2 deletion. Given that Neurod1/2 are similar and likely partially redundant, how are their observations different from the results described in PMID: 17988662 that already established a link between Neurod1/2 transcription factors and epsilon lineage decisions?

We thank the referee for raising this important remark and pointing us towards this relevant work. Response Fig. 16 shows that the expression patterns of *Neurod1* and *Neurod2* are indeed different. In particular, *Neurod2* is only highly expressed in *Ngn3* high, *Ngn3* high cycling, epsilon progenitors and *Fev*⁺ delta, whereas *Neurod1* is highly expressed in all endocrine progenitors as well as all islet cells. Similarly, we observed distinct mRNA expression patterns in iPSCs and human primary islets ((Cota et al. 2023)., Fig. 1c). Moreover, Fig. 2e)-f) suggests that NEUROD1 and NEUROD2 are not redundant as there is no compensation of NEUROD2 by NEUROD1. Finally, Response Fig. 17 shows that the set of predicted target genes of *Neurod1* and *Neurod2* in mouse embryonic endocrine lineages are non-overlapping. All in all, this suggests that the results are non-trivial, and our results are significantly different from the ones reported in PMID: 17988662. We added the following paragraph to highlight the similarities and differences to PMID: 17988662:

Indeed, *Neurod1* has been shown to regulate islet cell differentiation (Chao et al. 2007). Yet, the expression patterns of *Neurod1* and *Neurod2* are clearly distinct during mouse endocrinogenesis (Supplementary Fig. 45) and human iPSC differentiation (Cota et al. 2023), suggesting non-redundant and distinct functions of these TFs in regulating epsilon cell formation.

Response Fig. 16 | **Neurod1 and Neurod2 expression in the course of pancreatic development** → Included as Suppl. Fig. 45

a) Normalized expression of *Neurod1* and *Neurod2* for time point E12.5 and E13.5 in the dataset published by Bastidas-Ponce et al. (Bastidas-Ponce et al. 2019) (top), and normalized expression in our dataset for time points E13.5, E14.5 and E15.5, as well as respective UMAPs colored by cell type. **b)** Mean expression of *Neurod1* and *Neurod2* per cell type across different developmental stages. Only cell types comprising at least 3 cells are kept.

a

Response Fig. 17 | **Target genes of Neurod1 and Neurod2**

a) Target genes of *Neurod1* and *Neurod2* computed with Scenic+ (Bravo González-Blas et al. 2023)

10. The data in Fig 5h should be presented as % GHRL+ relative to all cells in culture, and the transcript quantification in Fig. 5i should be normalized to the average control value to preserve the spread – and therefore the ability to assess for statistical significance under the presumption of equal variance.

We appreciate the referee's suggestion. Regarding the quantification of GHRL+ cells, we have considerable numbers of undifferentiated cells in each independent experiment. Thus, we plotted the data to quantify the number of GHRL+ cells within the differentiated fraction. Having shown that the numbers (Cota et al., 2023) and areas (current study) of INS+ cells are comparable between control and NEUROD2 KO SC-islets, we quantified the number of GHRL+ cells per INS+ cell area to account for the high differentiation variability across independent experiments.

Regarding the qPCR data analysis, we would like to thank the referee for this very constructive suggestion. Indeed, we reanalyzed the data using the approach recommended by the referee that yielded improved results, which are reflected in the new plots provided in the revised version, and which we summarized in Response Fig. 8. We describe the statistical procedure in Methods 2.1.6.

11. The use of the word 'evolve' is best reserved for situations of actual evolution. 'Differentiate' or similar is more appropriate when speaking of the developmental transition between stages.

We thank the referee for this suggestion and replaced by the word 'evolve' by 'differentiate' / 'mature into' / 'develop into'.

References

- Arnes, Luis, Jonathon T. Hill, Stefanie Gross, Mark A. Magnuson, and Lori Sussel. 2012. "Ghrelin Expression in the Mouse Pancreas Defines a Unique Multipotent Progenitor Population." *PLoS One* 7 (12): e52026.
- Bastidas-Ponce, Aimée, Sophie Tritschler, Leander Dony, Katharina Scheibner, Marta Tarquis-Medina, Ciro Salinno, Silvia Schirge, et al. 2019. "Comprehensive Single Cell mRNA Profiling Reveals a Detailed Roadmap for Pancreatic Endocrinogenesis." *Development* 146 (12). <https://doi.org/10.1242/dev.173849>.
- Ben-Kiki, Oren, Akhiad Bercovich, Aviezer Lifshitz, and Amos Tanay. 2022. "Metacell-2: A Divide-and-Conquer Metacell Algorithm for Scalable scRNA-Seq Analysis." *Genome Biology* 23 (1): 100.
- Bergen, Volker, Marius Lange, Stefan Peidli, F. Alexander Wolf, and Fabian J. Theis. 2020. "Generalizing RNA Velocity to Transient Cell States through Dynamical Modeling." *Nature Biotechnology* 38 (12): 1408–14.
- Bravo González-Blas, Carmen, Seppe De Winter, Gert Hulselmans, Nikolai Hecker, Irina Matetovici, Valerie Christiaens, Suresh Poovathingal, Jasper Wouters, Sara Aibar, and Stein Aerts. 2023. "SCENIC+: Single-Cell Multiomic Inference of Enhancers and Gene Regulatory Networks." *Nature Methods* 20 (9): 1355–67.
- Byrnes, Lauren E., Daniel M. Wong, Meena Subramaniam, Nathaniel P. Meyer, Caroline L. Gilchrist, Sarah M. Knox, Aaron D. Tward, Chun J. Ye, and Julie B. Sneddon. 2018. "Lineage Dynamics of Murine Pancreatic Development at Single-Cell Resolution." *Nature Communications* 9 (1): 3922.
- Chao, Christina S., Zoe L. Loomis, Jacqueline E. Lee, and Lori Sussel. 2007. "Genetic Identification of a Novel NeuroD1 Function in the Early Differentiation of Islet Alpha, PP and Epsilon Cells." *Developmental Biology* 312 (2): 523–32.
- Chen, Ao, Sha Liao, Mengnan Cheng, Kailong Ma, Liang Wu, Yiwei Lai, Xiaojie Qiu, et al. 2022. "Spatiotemporal Transcriptomic Atlas of Mouse Organogenesis Using DNA Nanoball-Patterned Arrays." *Cell* 185 (10): 1777–92.e21.
- Cockburn, Katie, Karl Annusver, David G. Gonzalez, Smirthy Ganesan, Dennis P. May, Kailin R. Mesa, Kyogo Kawaguchi, Maria Kasper, and Valentina Greco. 2022. "Gradual Differentiation Uncoupled from Cell Cycle Exit Generates Heterogeneity in the Epidermal Stem Cell Layer." *Nature Cell Biology* 24 (12): 1692–1700.
- Cota, Perla, Lama Saber, Damla Taskin, Changying Jing, Aimée Bastidas-Ponce, Matthew Vanheusden, Alireza Shahryari, et al. 2023. "NEUROD2 Function Is Dispensable for Human Pancreatic β Cell Specification." *Frontiers in Endocrinology* 14 (October): 1286590.
- Demetci, Pinar, Rebecca Santorella, Björn Sandstede, William Stafford Noble, and Ritambhara Singh. 2022. "SCOT: Single-Cell Multi-Omics Alignment with Optimal Transport." *Journal of Computational Biology: A Journal of Computational Molecular Cell Biology* 29 (1): 3–18.
- Gasa, Rosa, Caroline Mrejen, Francis C. Lynn, Peter Skewes-Cox, Lidia Sanchez, Katherine Y. Yang, Chin-Hsing Lin, Ramon Gomis, and Michael S. German. 2008. "Induction of Pancreatic Islet Cell Differentiation by the Neurogenin-neuroD Cascade." *Differentiation; Research in Biological Diversity* 76 (4): 381–91.
- Gayoso, Adam, Philipp Weiler, Mohammad Lotfollahi, Dominik Klein, Justin Hong, Aaron Streets, Fabian J. Theis, and Nir Yosef. 2024. "Deep Generative Modeling of Transcriptional Dynamics for RNA Velocity Analysis in Single Cells." *Nature Methods* 21 (1): 50–59.
- Guilliams, Martin, Johnny Bonnardel, Birthe Haest, Bart Vanderborcht, Camille Wagner, Anneleen Remmerie, Anna Bujko, et al. 2022. "Spatial Proteogenomics Reveals Distinct and Evolutionarily Conserved Hepatic Macrophage Niches." *Cell* 185 (2): 379–96.e38.
- Gulati, Gunsagar S., Shaheen S. Sikandar, Daniel J. Wesche, Anoop Manjunath, Anjan Bharadwaj, Mark J. Berger, Francisco Ilagan, et al. 2020. "Single-Cell Transcriptional

- Diversity Is a Hallmark of Developmental Potential." *Science* 367 (6476): 405–11.
- Haghverdi, Laleh, Maren Büttner, F. Alexander Wolf, Florian Buettner, and Fabian J. Theis. 2016. "Diffusion Pseudotime Robustly Reconstructs Lineage Branching." *Nature Methods* 13 (10): 845–48.
- Herrera, P. L. 2000. "Adult Insulin- and Glucagon-Producing Cells Differentiate from Two Independent Cell Lineages." *Development* 127 (11): 2317–22.
- Huising, Mark O., Talitha van der Meulen, Jessica L. Huang, Mohammad S. Pourhosseinzadeh, and Glyn M. Noguchi. 2018. "The Difference δ -Cells Make in Glucose Control." *Physiology* 33 (6): 403–11.
- Lange, Marius, Volker Bergen, Michal Klein, Manu Setty, Bernhard Reuter, Mostafa Bakhti, Heiko Lickert, et al. 2022. "CellRank for Directed Single-Cell Fate Mapping." *Nature Methods* 19 (2): 159–70.
- Li, Chen, Maria C. Virgilio, Kathleen L. Collins, and Joshua D. Welch. 2022. "Multi-Omic Single-Cell Velocity Models Epigenome–transcriptome Interactions and Improves Cell Fate Prediction." *Nature Biotechnology* 41 (3): 387–98.
- Lickert, Heiko, Kaiyuan Yang, Hannah Spitzer, Michael Sterr, Karin Hrovatin, Xinghao Zhang, Eunike Setyono, et al. 2024. "A Multimodal Cross-Species Comparison of Pancreas Development." <https://europepmc.org> › PPR › PP...<https://europepmc.org> › PPR › PP... <https://europepmc.org/article/PPR/PPR832963>.
- Qiu, Chengxiang, Junyue Cao, Beth K. Martin, Tony Li, Ian C. Welsh, Sanjay Srivatsan, Xingfan Huang, et al. 2022. "Systematic Reconstruction of Cellular Trajectories across Mouse Embryogenesis." *Nature Genetics* 54 (3): 328–41.
- Qiu, Xiaojie, Daniel Y. Zhu, Jiajun Yao, Zehua Jing, Lulu Zuo, Mingyue Wang, Kyung Hoi (Joseph) Min, et al. 2022. "Spateo: Multidimensional Spatiotemporal Modeling of Single-Cell Spatial Transcriptomics." *bioRxiv*. <https://doi.org/10.1101/2022.12.07.519417>.
- Quoix, Nicolas, Rui Cheng-Xue, Yves Guiot, Pedro L. Herrera, Jean-Claude Henquin, and Patrick Gilon. 2007. "The GluCre-ROSA26EYFP Mouse: A New Model for Easy Identification of Living Pancreatic Alpha-Cells." *FEBS Letters* 581 (22): 4235–40.
- Satija, Rahul, Jeffrey A. Farrell, David Gennert, Alexander F. Schier, and Aviv Regev. 2015. "Spatial Reconstruction of Single-Cell Gene Expression Data." *Nature Biotechnology* 33 (5): 495–502.
- Schiebinger, Geoffrey, Jian Shu, Marcin Tabaka, Brian Cleary, Vidya Subramanian, Aryeh Solomon, Joshua Gould, et al. 2019. "Optimal-Transport Analysis of Single-Cell Gene Expression Identifies Developmental Trajectories in Reprogramming." *Cell* 176 (6): 1517.
- Thorel, Fabrizio, Virginie Népote, Isabelle Avril, Kenji Kohno, Renaud Desgraz, Simona Chera, and Pedro L. Herrera. 2010. "Conversion of Adult Pancreatic Alpha-Cells to Beta-Cells after Extreme Beta-Cell Loss." *Nature* 464 (7292): 1149–54.
- Tirosh, Itay, Andrew S. Venteicher, Christine Hebert, Leah E. Escalante, Anoop P. Patel, Keren Yizhak, Jonathan M. Fisher, et al. 2016. "Single-Cell RNA-Seq Supports a Developmental Hierarchy in Human Oligodendroglioma." *Nature* 539 (7628): 309–13.
- Tschöp, M., D. L. Smiley, and M. L. Heiman. 2000. "Ghrelin Induces Adiposity in Rodents." *Nature* 407 (6806): 908–13.
- Wang, Xiaofang, Yanjuan Chen, Zongcheng Li, Bingyan Huang, Ling Xu, Jing Lai, Yuhong Lu, et al. 2021. "Single-Cell RNA-Seq of T Cells in B-ALL Patients Reveals an Exhausted Subset with Remarkable Heterogeneity." *Advancement of Science* 8 (19): e2101447.
- Weiler, Philipp, Marius Lange, Michal Klein, Dana Pe'er, and Fabian J. Theis. 2023. "Unified Fate Mapping in Multiview Single-Cell Data." *bioRxiv*. <https://doi.org/10.1101/2023.07.19.549685>.
- Wolf, F. Alexander, Philipp Angerer, and Fabian J. Theis. 2018. "SCANPY: Large-Scale Single-Cell Gene Expression Data Analysis." *Genome Biology* 19 (1): 15.
- Yeo, Grace Hui Ting, Sachit D. Saksena, and David K. Gifford. 2021. "Generative Modeling of Single-Cell Time Series with PRESCIENT Enables Prediction of Cell Trajectories with

- Interventions." *Nature Communications* 12 (1): 3222.
- Yu, Xin-Xin, Wei-Lin Qiu, Liu Yang, Yan-Chun Wang, Mao-Yang He, Dan Wang, Yu Zhang, et al. 2021. "Sequential Progenitor States Mark the Generation of Pancreatic Endocrine Lineages in Mice and Humans." *Cell Research* 31 (8): 886–903.
- Zhang, Yanqing, Genevieve E. Fava, Hongjun Wang, Franck Mauvais-Jarvis, Vivian A. Fonseca, and Hongju Wu. 2016. "PAX4 Gene Transfer Induces α -to- β Cell Phenotypic Conversion and Confers Therapeutic Benefits for Diabetes Treatment." *Molecular Therapy: The Journal of the American Society of Gene Therapy* 24 (2): 251–60.

Reviewer Reports on the Second Revision:

Referees' comments:

Referee #1 (Remarks to the Author):

We want to thank the authors for carefully addressing a consistency check on their algorithm. More specifically they showed that when integrating different batches, the choice of reference batch does not significantly distort the biological information. As they point out they use entropy regularized OT, so the mapping becomes stochastic, but the way they perform the benchmarking in this context seems appropriate. The authors have completely addressed this point. Authors have also satisfactorily addressed all other points we made regarding comparisons with other pseudotime estimators, and suggestions on improving figures and result presentations. We congratulate the authors on this wonderful work.

Referee #2 (Remarks to the Author):

I appreciate that the authors' efforts addressing my questions. I do not have more questions.

Referee #3 (Remarks to the Author):

The authors have mostly addressed my previous comments. I am having a few additional suggestions to their figures:

1. Many main figures have very small font sizes. For example, fonts in Fig. 3a and Fig. 3g are too small to read.
2. The same issue applies to many supplementary figures: Fig. SI1b, Fig. SI3d, Fig. SI11a, Fig. SI12C, Fig. SI13, Fig. SI15, Fig. SI37, Fig. SI43, Fig. SI44, Fig. SI45, Fig. SI50,
3. Some figures' organization should be improved, e.g., Fig. SI9. There is a huge white space between panel b and d. You can for example, shrink the size of panel b and move it on the right side of panel a under the figure legend. Then shrink the width of panels c, d and put it below panel a, b.
4. Similar problem for Fig. SI10, Fig. SI 22, Fig. SI 32, Fig. SI 37, fig. SI38, fig. SI39, Fig. SI41, Fig. SI46, Fig. SI47.

Referee #5 (Remarks to the Author):

Thank you for your careful and thoughtful responses to the prior comments to this manuscript. I think they have strengthened an already highly interesting manuscript that is based on a technically outstanding approach and dataset.

I have two follow up comments:

1) A major conclusion from this work – where it pertains to the pancreas lineage development and perhaps other areas of biology – is that your work has in large part independently validated the careful work of many others in the field. The lack of major shifts in our understanding of pancreas development is a testament to the overall quality of the work by a generation of developmental biologists before us. While I recognize that authors and editors alike are incentivized by claims of novelty, perhaps a statement to the lack of new biological insight / validation that our understanding of lineage history of pancreas development is largely established is appropriate here. Perhaps this statement from the rebuttal could be refined and worked into the manuscript text, before discussing the added granularity of the insights obtained here:

‘...indeed, the developmental biologists have done a great job in delineating the lineage trajectories of all endocrine subtypes in the islets of Langerhans, which our data confirms.

2) In response to my earlier comment suggesting that Moscon might have picked up alpha to beta cell transitions between E14.5 and E15.5, the authors are quick to attribute this ‘noise in their cell clustering procedure’, which may of course be correct. However, there is no indication that lineage transitions of alpha cell have to be accompanied by enhanced Pax4. The work by Thorel and Herrera shows that lineage transitions do not normally occur in adult pancreas unless under extreme experimental conditions. However, the results described in PMID: 28380380 with the same Gcg-Cre driver that led Herrera to a conclusion of strictly separate lineages for alpha and beta cells indicate that in fact some lineage transitions occur between beta and alpha cells in both directions during development, but no longer thereafter once this developmental window for lineage plasticity has closed. The main argument against the events in PMID: 28380380 reflecting mere artifacts of lineage tracing, is that these lineage-converted cells occur specifically at the periphery: the occasional beta cell picking up an alpha cell lineage trace in stochastic fashion would not have resulted in such cells occurring preferentially at the periphery of the islet. So perhaps, you need not be so quick to dismiss the detection of transitions between alpha and beta cells as ‘noise’ as these events are in line with known lineage plasticity specifically in developing pancreas that is known to not extend to adult mice (or humans). Are these events clinically relevant? Almost certainly not. But the observation of such potential lineage conversions via moscon indicates that your approach is sensitive enough to detect infrequent events that were described by others in a completely different and independent approach.

Author Rebuttals to Second Revision:

Response letter to Nature Manuscript 2023-05-08448

In the following, we present our response to the reviewers comments. We restate their original **comments (black)**, give **point-by-point answers (green)** to the questions, and **copy parts of the text or specific panels (blue)**, which directly correspond to comments or refer to them.

Reviewer #1

We want to thank the authors for carefully addressing a consistency check on their algorithm. More specifically they showed that when integrating different batches, the choice of reference batch does not significantly distort the biological information. As they point out they use entropy regularized OT, so the mapping becomes stochastic, but the way they perform the benchmarking in this context seems appropriate. The authors have completely addressed this point. Authors have also satisfactorily addressed all other points we made regarding comparisons with other pseudotime estimators, and suggestions on improving figures and result presentations. We congratulate the authors on this wonderful work.

We thank the reviewer for their effort in reviewing our paper and for their positive feedback.

Reviewer #2

I appreciate that the authors' efforts addressing my questions. I do not have more questions.

We thank the reviewer for the positive feedback and are glad to have addressed all questions.

Reviewer #3

The authors have mostly addressed my previous comments.

We thank the reviewer for their constructive comments and are glad to have addressed all comments.

I am having a few additional suggestions to their figures:

1. Many main figures have very small font sizes. For example, fonts in Fig. 3a and Fig. 3g are too small to read.

We increased the font in Fig.3a and Fig.3g

2. The same issue applies to many supplementary figures: Fig. SI1b, Fig. SI3d, Fig. SI11a, Fig. SI12C, Fig. SI13, Fig. SI15, Fig. SI37, Fig. SI43, Fig. SI44, Fig. SI45, Fig. SI50,

We did increase the font size wherever possible, in detail (with the same numbering as in the last round of revision):

Fig. SI3d: We increased the font size in the legend

Fig. SI11a: We rearranged the figure and hence were able to increase the font size in panel e

Fig. SI12C: We slightly increased the font size.

Fig.13S: We increased the font sizes.

Fig. 15S: Unfortunately, we did not find a way to improve the presentation.

Fig. SI37: We increased the font size.

Fig. SI43: We increased the font size.

Fig. SI44: We increased the font size.

Fig. SI45: We increased the font size by rearranging the figure.

Fig. SI50: We slightly increased the font size by rearranging the figure

3. Some figures' organization should be improved, e.g., Fig. SI9. There is a huge white space between panel b and d. You can for example, shrink the size of panel b and move it on the right side of panel a under the figure legend. Then shrink the width of panels c, d and put it below panel a, b.

We adapted Fig S.9 according to the reviewer's suggestion.

4. Similar problem for Fig. SI10, Fig. SI 22, Fig. SI 32, Fig. SI 37, fig. SI38, fig. SI39, Fig. SI41, Fig. SI46, Fig. SI47

Fig. SI10: We restructured the figure

Fig. SI22: We tried to improve the layout of the figure

Fig. SI 32: We rearranged the figure

Fig. SI37: We increased the font size (see above), but could find a way to better restructure the figure

Fig. SI 38: We reduced the white space

Fig. SI 38: We restructured the figure

Fig. SI 41: We apologize for not finding a better way to have less white space

Fig. SI 46: We restructured the figure

Fig. SI 47: We reduced the white space, but would like to keep the overall structure for the sake of readability

Reviewer #5

Thank you for your careful and thoughtful responses to the prior comments to this manuscript. I think they have strengthened an already highly interesting manuscript that is based on a technically outstanding approach and dataset.

We thank the reviewer for the kind words.

I have two follow up comments:

1) A major conclusion from this work – where it pertains to the pancreas lineage development and perhaps other areas of biology – is that your work has in large part independently validated the careful work of many others in the field. The lack of major shifts in our understanding of pancreas development is a testament to the overall quality of the work by a generation of developmental biologists before us. While I recognize that authors and editors alike are incentivized by claims of novelty, perhaps a statement to the lack of new biological insight / validation that our understanding of lineage history of pancreas development is largely established is appropriate here. Perhaps this statement from the rebuttal could be refined and worked into the manuscript text, before discussing the added granularity of the insights obtained here:

'...indeed, the developmental biologists have done a great job in delineating the lineage trajectories of all endocrine subtypes in the islets of Langerhans, which our data confirms.

We thank the reviewer for this suggestion and we have included a similar statement in the beginning of the corresponding paragraph. In detail, we replaced

Similarly, while our previous analysis hypothesized that epsilon cells can develop from both *Ngn3+* progenitors and glucagon producing alpha cells, lineage tracing experiments confirmed that ghrelin-producing epsilon cells are not a terminal state and can in turn give rise to alpha and PP cells and rare beta cells.

by

Similarly, while our previous analysis hypothesized that epsilon cells can develop from both *Ngn3+* progenitors and glucagon producing alpha cells, one of multiple insightful and carefully conducted lineage tracing experiments studying endocrine cells in the islets of Langerhans confirmed that ghrelin-producing epsilon cells are not a terminal state and can in turn give rise to alpha and PP cells and rare beta cells.

2) In response to my earlier comment suggesting that Moscon might have picked up alpha to beta cell transitions between E14.5 and E15.5, the authors are quick to attribute this 'noise in their cell clustering procedure', which may of course be correct. However, there is no indication that lineage transitions of alpha cell have to be accompanied by enhanced Pax4. The work by Thorel and Herrera shows that lineage transitions do not normally occur in adult pancreas unless under extreme experimental conditions. However, the results described in PMID: 28380380 with the same Gcg-Cre driver that led Herrera to a conclusion of strictly separate lineages for alpha and beta cells indicate that in fact some lineage transitions occur between beta and alpha cells in both directions during development, but no longer thereafter once this developmental window for lineage plasticity has closed. The main argument against the events in PMID: 28380380 reflecting mere artifacts of lineage tracing, is that these lineage-converted cells occur specifically at the periphery: the occasional beta cell picking up an alpha cell lineage trace in stochastic fashion would not have resulted in such cells occurring preferentially at the periphery of the islet. So perhaps, you need not be so quick to dismiss the detection of transitions between alpha and beta cells as 'noise' as these events are in line with known lineage plasticity specifically in developing pancreas that is known to not extend to adult mice (or humans). Are these

events clinically relevant? Almost certainly not. But the observation of such potential lineage conversions via moscon indicates that your approach is sensitive enough to detect infrequent events that were described by others in a completely different and independent approach.

We thank the reviewer for this suggestion and we have corrected our statement accordingly and mentioned that during embryonic development and early postnatally cellular plasticity still exists, one example are virgin beta cells that form alpha cells in the islet periphery (PMID: 28380380), but that in adult mice these alpha-to-beta transitions are only observed under extreme experimental conditions, i.e. >99% ablation of beta cells.

In particular, we replaced the paragraph

[...] with slight differences being likely due to noise in cell type annotation, sequencing biases or limitations in the moscot algorithm.

by

While slight differences in transition probabilities are likely due to noise in the cell type annotation, sequencing biases, or limitations of the moscot algorithm, certain changes in transition likelihoods might be biological. While we could not find any further evidence for the non-zero transdifferentiation probability between alpha and beta cells in E14.5/E15.5, which ceases for E15.5/E16.5, this cellular behavior has been observed in mice in the late stages of pancreas development (van der Meulen et al. 2017)

In the name of all authors, I would like to thank the reviewers for their constructive criticism and insightful comments during all rounds of revision.

Best regards,

Fabian Theis